# Removal of Iodine-Containing X-ray Contrast Media from Environment: The Challenge of a Total Mineralization

**DOI:** 10.3390/molecules28010341

**Published:** 2023-01-01

**Authors:** Wei Zhang, Florence Fourcade, Abdeltif Amrane, Florence Geneste

**Affiliations:** 1Ecole Nationale Supérieure de Chimie de Rennes, CNRS, ISCR-UMR 6226, Univ Rennes, 35000 Rennes, France; 2CNRS, ISCR-UMR 6226, Univ Rennes, 35000 Rennes, France

**Keywords:** iodinated X-ray contrast media, depollution, byproducts, total mineralization, advanced oxidation processes (AOPs), reductive deiodination

## Abstract

Iodinated X-ray contrast media (ICM) as emerging micropollutants have attracted considerable attention in recent years due to their high detected concentration in water systems. It results in environmental issues partly due to the formation of toxic by-products during the disinfection process in water treatment. Consequently, various approaches have been investigated by researchers in order to achieve ICM total mineralization. This review discusses the different methods that have been used to degrade them, with special attention to the mineralization yield and to the nature of formed by-products. The problem of pollution by ICM is discussed in the first part dedicated to the presence of ICM in the environment and its consequences. In the second part, the processes for ICM treatment including biological treatment, advanced oxidation/reductive processes, and coupled processes are reviewed in detail. The main results and mechanisms involved in each approach are described, and by-products identified during the different treatments are listed. Moreover, based on their efficiency and their cost-effectiveness, the prospects and process developments of ICM treatment are discussed.

## 1. Introduction

The continuous increase in the human population has created a comparable growth in the demand for water resources. In such an urgent situation, protecting our aquatic environment to achieve a comprehensive sustainable development strategy is one of the most essential environmental topics of the past decades. It has been shown that abounding compounds can enter the environment, such as pharmaceuticals [1,2,3], disinfectants, and other contaminants [4,5,6]. One of the most important factors contributing to this phenomenon is hospitals’ effluents as a source of contaminants discharge, such as specific iodinated X-ray contrast media (ICM) [7,8,9], antitumor agents [10,11] as well as antibiotics [12,13].

Since the discovery of X-ray radiation by Wilhelm C. Rogentgen in 1895, the diagnostic imaging procedure has become a routine part of medicine due to the possible external visualization of the internal anatomic structures of a patient without surgery [14]. Suspensions of barium sulfate have been applied for radiographic imaging since 1910. However, the marked radiodensity and poor intrinsic affinity for the gastrointestinal mucosa of this agent led to inaccurate diagnoses. In such a background, water-soluble iodinated X-ray contrast media (ICM) are the most extensively used pharmaceuticals for intravascular administration [15].

ICM are derivatives of 2,4,6-triiodobenzoic acid with polar carboxyl and hydroxyl moieties in their chains. Some of them are ionic, i.e. have one or several free carboxyl groups, and others are amide derivatives. They have all three iodine atoms in their chemical structure, which are responsible for the absorption of X-rays.

As ICM have been widely used for more than 40 years, their production is well-controlled. However, their synthesis is still the subject of high investigations to make it greener [16] and more efficient [17,18]. The recovery of the iodine atoms in industrial wastewater is also well-studied as it can reduce the cost of the overall production process [19,20]. 

Due to their high administered concentrations and their biorecalcitrance, ICM emerged as new environmental pollutants. For this reason, many articles have been focused on processes allowing their removal, deiodination, and degradation. The interest in ICM remediation is highlighted by a recent review on ICM transformation [8,21,22]. Since partial elimination of ICM cannot avoid the formation of toxic by-products [22], we will focus this review paper on the treatments that can be used for the total degradation of ICM. In the first part, we will give a detailed overview of the occurrence of ICM in the environment and on their ecotoxicology. Then we will detail the results of the degradation of ICM according to the different processes that have been tested with special attention paid to their effectiveness to provide total mineralization of ICM and to the toxicity and biodegradability of formed by-products. They include the classical biological treatments, the large family of oxidation processes, and also more selective processes such as reductive deiodination.

## 2. ICM in the Environment

### 2.1. Effluents Polluted by ICM

From the internet site of the French National College of Medical Pharmacology (https://pharmacomedicale.org, accessed on 28 December 2022), 1.5 million contrast media injections were notified in France in 2018 and the most used were iodinated contrast media. Moreover, in the nineties, the annual consumption of ICM worldwide is estimated to have reached a value of 3.5 × 10^6^ kg, since they are administered intravascularly in very high doses (up to 200 g ICM, corresponding to approximately 100 g iodine per application) [23]. In addition, ICM has an average half-life in the human body of around 2 h, and 75% of the administered dose is excreted via urine or feces into the hospital wastewater within 4h and nearly 100% within 24 h [24]. They were identified as a major source of increased absorbable organic halogen (AOX) concentrations in hospital sewage water [25]. In a large part of the region, hospital wastewater is not treated separately and is finally discharged into the local sewage systems together with domestic wastewater.

Even if wastewater treatment plants (WWTP) can eliminate most pharmaceutical chemicals during the treatment process, various treatment systems are required for different contaminants in order to achieve efficient removal. Pollutants with polar characters can easily bypass the water purification process and so reach surface waters with a potential impact on the ecosystem and human health [26].

The n-octanol/water partition coefficient of ICM (Table 1) characterizes them as very hydrophilic.

As a consequence, a large amount of literature has revealed that ICM have been detected in our living waters. For example, Seitz et al. [28] determined ICM concentrations in the Danube river through a monitoring programmer. The study revealed that the maximum concentrations of ICM (over 500 ng L^−1^ for diatrizoic acid and iopamidol) were found in samples taken from downstream of the Neu-Ulm metropolitan area of Germany. A similar situation was found in Galicia, Spain [29] and in the southwestern United States [30]. In the same way, in China, total ICM concentrations around 160 and 97.4 ng L^−1^ were found in the Huangpu River and the Tahu Lake, respectively; higher concentrations (about 430 ng L^−1^) of iodinated tri-halomethane, an ICM oxidation product, were determined in local wastewater plant from these places [31].

In a recent review, Sengar and Vijayanandan summarize data from 51 articles from 2001 on the occurrence of ICM in hospital wastewater, surface water, groundwater, and drinking water in North America, Asia, and Europe [8]. Table 1 summarizes the concentration range found in Europe for diatrizoate, iohexol, iopamidol, iopromide, ioxithalamic acid, and iomeprol. Concentrations from 0.01 to 3800 μg L^−1^ were found in hospital wastewater, with the highest reached concentrations for iohexol, iopamidol, and iomeprol. In surface and groundwater, the concentrations were lower ranging from 10^−3^ to 10 μg L^−1^ and <0.02 μg L^−1^ for drinking water. This review also centralized data at the international level on the ICM concentrations at the entry and exit of wastewater plants and highlighted the variability of their removal performances depending on the ICM and on implemented treatment in the WWP that explained the presence of ICM in the environment.

Considering the high concentrations in hospital wastewater, the MERK’MAL project suggested urine collection in order to reduce the ICM environmental impact on the Ruhr River [32]. The collection was conducted on four medical facilities using daily ICM *via* urine bags. This concept allowed an ICM reduction in the range of 20–30% in the tested wastewater treatment plants.

### 2.2. Ecotoxicology of ICM

Pharmaceuticals are designed to target specific metabolic and molecular pathways in humans and animals, but they often have important side effects too. When introduced into the environment, they may affect the same pathways in animals having identical or similar target organs, tissues, cells, or biomolecules [33]. For ICM acute ecotoxicology, even if it is usually reversible, ICM can lead to contrast-induced nephropathy (CIN), which is associated with longer hospitalization time, the need for dialysis, higher incidence of later cardiovascular events, and higher mortality [34]. However, in short-term toxicity tests with bacteria (*Vibrio fisheri*, *Pseudomonas putida*), algae (*Scenedesmus subspicatus*), crustaceans (*Daphnia magna*), and fish (*Danio rerio*, *Leuciscus idus*), no toxic effects were detected at the highest tested concentration of 10 g L^−1^ according to the study of Steger-Hartmann [35]. Ecotoxicity investigation gave indications of acute effects *in vivo* in organisms after short-term exposure, but only rarely after chronic exposures. Oleksy-Frenzel et al. [36] have shown that ICM may significantly contribute to the burden of absorbable organic halogens (AOX), which can cause a severe underestimation of pollution in municipal wastewater.

Considering disinfection processes in water treatment, as most ICM have an amino group in their molecular structure, the production of some nitrogenous byproducts such as haloacetonitrile; haloacetamides occurs and is problematic due to their toxicity [37]. Moreover, iodinated trihalomethane, another toxic ICM byproduct can be produced from the ICM reaction with disinfecting agent [38]. In the case of Iopamidol, the formation of toxic iodinated trihalomethane can be reduced during the chlorination step after a pre-oxidation with Fe(VI) at pH 9.

In conclusion, the occurrence of ICM in the environment has been largely observed over the world. Although their toxicity is low, their presence during wastewater treatment and disinfection process can lead to the formation of toxic by-products. It is therefore a major concern that has to be solved. Their total mineralization by concentrated wastewater treatment can be a solution.

## 3. Processes for the Treatment of ICM

Owing to diffusion in the environment of ICM and to their persistence after treatment in wastewater treatment plants, the degradation of high concentrated ICM effluents (i.e. in-house sewer, hospital waste, and industrial wastewater) would contribute to the better control of pollution. To obtain high-concentration removal, ICM can also be concentrated beforehand, through adsorption or a membrane process, for instance. To achieve better and tunable contaminant adsorption, a uniform pore structure with a high specific surface area, large pore volume, chemical inertness, controllable structure, and thermal stability are essential properties to select an optimum adsorbent. Activated carbon, alumina, silica, zeolite, composite materials, and metal-organic frameworks have already shown good adsorption capacity for the removal of pollutants [39,40,41,42]. Low-cost and widely available adsorbents such as magnetically modified agro-industrial wastes, spent tea leaves, and marine microalga have also shown promising results for the treatment of pollutants [43,44,45]. Nevertheless, few works have been reported on the adsorption of ICM. For this purpose, superparamagnetic iron oxide nanoparticles coated with methacrylic acid, Al(OH)_3_, or SiO_2_ have been tested for adsorptive removal of diatrizoate (100 μg L^−1^). The maximum removal efficiency was around 60% with 0.05 g adsorbent in 0.5 L of influent at pH 8 and after 7h of incubation [46]. Another way to concentrate organic compounds can be membrane separation. In their study concerning the removal of pharmaceuticals in hospital wastewater, Lan et al. studied the relevance of micropollutants concentration by nanofiltration before their degradation by electrochemical oxidation [47]. 

### 3.1. Biological Processes for Removing ICM

Among existing processes that have been tested to treat ICM wastewater effluents, biodegradation is considered an environmentally friendly and cost-effective approach.

#### 3.1.1. Aerobic Biological Treatment Processes

Aerobic microorganisms can degrade some organic pollutants. They grow fast and can mineralize organic compounds that are used as a carbon and/or an energy source [48]. Several studies on the biotransformation of low ICM concentration have been reported in the literature in this regard [49,50,51]. Iopromide in the range of 0.10 to 0.27 µg L^−1^ can be removed at a high percentage by biotransformation (Table 2, entry 1) [49].

The removal efficiency reached 97% in the presence of nitrifying bacteria in activated sludge with the formation of a dehydroxylated iopromide as a metabolite. When the nitrifying bacteria was inhibited, the removal yield decreased to 86% and the metabolite was a carboxylate derivative [49]. Kalsch et al. also noticed that after 54h of incubation in fresh activated sludge, 85% of iopromide (1.85 nmol L^−1 14^C) were biotransformed but metabolites were not identified (Table 2, entry 2) [51]. The same experiment was performed with diatrizoate but the degradation with activated sludge was negligible. The biodegradability of diatrizoate was also tested with a Zahn-Wellens test and a simulating biological sewage treatment. According to the first test, diatrizoate was biotransformed into 2,4,6-triiodo-3,5-diamino-benzoic acid but the latter was not eliminated (Table 2, entry 3) [50]. 

Kormos et al. [52] also selected typical ICM (diatrizoate, iohexol, iomeprol, and iopamidol) and investigated their biotransformation in aerobic soil. After a slow process of up to half year, they observed the biotransformation of iohexol, iomeprol, and iopamidol at neutral pH, whereas diatrizoate was persistent (Table 2, entry 4). The biorecalcitrance of diatrizoate has also been observed with biotransformation performed in a subsurface flow-constructed wetland pond with floating plants (Table 2, entry 6). Common microbial reactions were observed with ICM as exemplified by the biodegradation pathways of iohexol and iopamidol in Figure 1. Although the effectiveness of the bioreactions is dependent on the matrix composition, some reaction types such as the oxidation of the primary or secondary alcohols, a decarboxylation, and the cleavage of the N-C bond and the acetyl group could be identified (Figure 1).

The coupling of the activated sludge process and membrane separation was also attempted at Rensselaer Polytechnic Institute and commercialized in the 70′ s and 80′ s [56]. The bioreactor and membrane modules have a specific function:(i)Biological degradation of organic pollution is carried out in the bioreactor by adapted microorganisms.(ii)After treatment, the microorganisms are separated from the treated wastewater by the membrane module.

A pilot-scale membrane bioreactor (MBR) was installed and operated for one year at a Swiss hospital to study the efficiency of ICM micropollutant elimination [55]. The highest elimination efficiency within six ICM was for iopromide (31%), while the elimination was negligible for the others (Table 2, entry 7).

#### 3.1.2. Anaerobic Biological Treatments

The anaerobic biological treatment has also received some attention, although biodegradation studies with emerging organic contaminants, such as pharmaceuticals and personal care products, have been mainly performed under aerobic conditions.

Anaerobic conditions can be interesting to treat biorecalcitrant compounds such as diatrizoate. Indeed, Redeker et al. [53] have found that in anaerobic batch experiments with soil and sediment, diatrizoate at 100 µg L^−1^ could be removed, leading to seven by-products (Table 2, entry 5). The determination of the biodegradation pathways revealed that deacetylations are present in addition to deiodinations. However, one by-product, 3,5-diaminobenzoic acid, was stable under anaerobic conditions. For iopromide, the anaerobic transformation during batch experiments in water-sediment systems was also set up in a glovebox under argon atmosphere [57]. After 20 days of incubation, iopromide, initially at 2.6 µmol L^−1^, nearly completely disappeared and was totally dehalogenated by successive reductive deiodinations. In addition, the anilide moieties in the chain were hydrolyzed. The predominant transformation of iopromide was sequential reductive deiodination and five transformation by-products were formed.

Table 2 summarizes the recent studies of ICM treatment by aerobic and anaerobic biological processes. For some ICM, especially diatrizoate, the removal efficiency was negligible in conventional conditions. For iopromide improvement in specific conditions such as activated sludge containing ammonia oxidizing bacteria could be achieved but still need to be explored. The implementation of aerobic biological treatment for the removal of high ICM concentrations does not seem to be efficient since only some ICM at low concentration can be biotransformed. An anaerobic treatment could be more appropriate since dehalogenation can occur. However, the studies have been only performed at low ICM concentrations. Since classical biological treatment cannot lead to total mineralization of ICMs, physico-chemical processes have been widely investigated.

### 3.2. Advanced Oxidation Processes (AOPs) for ICM Removal

Advanced oxidation processes (AOPs) have been extensively studied for the treatment of organic pollutants. AOPs are very efficient degradation treatments for bio-refractory substances, since they involve the generation of hydroxyl radicals (•OH). Indeed, hydroxyl radicals are very reactive species, which attack pollutants with kinetic constants higher than 10^6^–10^9^ L mol^−1^ s^−1^. Figure 2 shows the oxidation potentials of common oxidants used in the depollution processes. Although the oxidation potential of •OH depends on the pH, it is a strong oxidant whose oxidation potential is just below those of fluorine [58,59]. Most AOPs use hydroxyl radicals or sulfate radicals SO_4_^•−^ which are also very strong oxidants (Figure 2).

The ICM degradation by AOPs has been widely investigated in recent years. Ozone and UV-based AOPs, as well as catalytic, electrochemical, and physical AOPs, including novel approaches for the activation of radicals, have been examined to treat ICM. In the following part, we summarized the literature on the degradation of ICM by AOPs, underlining their performances according to the nature of ICM, the proposed degradation mechanisms, and the possible shortcomings of each method. 

#### 3.2.1. Ozone-Based AOPs

Ozone-based AOPs involve the formation of hydroxyl radicals and could be summarized into different categories including ozonation, and combinations of O_3_/H_2_O_2_ (peroxone-process) and O_3_/UV. The O_3_/UV will be discussed in Section 3.2.2 as a UV-based AOPs.

##### Ozonation

Ozone has a wide range of action on micropollutants removal and water disinfection due to its interesting features involving molecular ozone oxidation and the possibility of generating hydroxyl radicals [60,61]. However, different studies focusing on ICM degradation by ozone have shown a high resistance of ICM to ozonation, with only partial degradation. It has been shown that O_3_ does not directly oxidize ICM, but a 50% degradation yield of ICM can be achieved for a high O_3_ concentration of 5 mg L^−1^, owing to the formation of hydroxyl radicals [62]. Even at this concentration, no degradation was obtained for diatrizoate, which seems to be less reactive to •OH. A series of investigations has been dedicated to the determination of the oxidation by-products. For instance, Seitz et al. [63] applied different liquid chromatography methods to characterize the by-products after ozonation. Aldehyde and carbonyl-containing compounds were identified as stable by-products in the case of iomeprol. Owing to the partial reactivity of ICM to ozonation, the addition of a co-reactant or catalyst to enhance the mineralization level of ICM during ozonation is demonstrated thereafter. Indeed, in order to improve the removal efficiency of iohexol, Yan et al. implemented catalytic ozonation [64]. Goethite and a composite magnetite goethite material proved their interest as a catalyst. An increase of the removal rate of 16.5% and 21.2% was obtained with goethite and the composite, respectively, compared with sole ozonation.

##### Peroxone-Process

The combination of ozone with hydrogen peroxide to enhance the removal of ICM is based on the ability of hydrogen peroxide to promote the formation of •OH compared with ozone alone [65,66]. The simplified mechanism is as follows:(1)$$O_{3}{+H}_{2}O_{2}\;\to {\;•\mathrm{OH}+\mathrm{HO}}_{2}{•+O}_{2}$$
(2)$$O_{3}{+\mathrm{HO}}_{2}•\;\to {\;•\mathrm{OH}+2O}_{2}$$

The aqueous degradation of four iodinated ICM compounds (diatrizoate, iomeprol, iopromide, and iopamidol) by ozone and ozone combined with hydrogen peroxide were investigated at a laboratory scale [66]. An enhancement of the radical mechanism by the addition of hydrogen peroxide was observed during ozonation, leading to the complete removal of the nonionic compounds, and more than 80% removal of diatrizoate, at relatively low oxidant mass ratios (H_2_O_2_/O_3_ < 0.25 with O_3_ concentration of 16 mg L^−1^ with 30 min ozonation). Radical oxidation based on ozone led to the release of inorganic iodine. As discussed above (see Ozonation section), the lower reactivity of diatrizoate under ozone treatment was confirmed by another study [65]. In that study, the ionic diatrizoate exhibited removal efficiencies of around only 24%, whereas the non-ionic ICMs were removed to a degree higher than 80% with 10 mg L^−1^ ozone (contact time: 18 min, H_2_O_2_/O_3_ = 1). Taking into account the low removal efficiencies of ICM under O_3_ and H_2_O_2_/O_3_ conditions, other AOPs have been explored to obtain a higher removal efficiency. 

The removal of diatrizoate was also studied by an electro-peroxone process [67]. Ferrite-modified carbon nanotubes were applied as catalyst-based gas diffusion cathodes. Although high degradation yields of up to 70% were obtained, no information on mineralization was given. 

#### 3.2.2. UV-Based AOPs

It is well known that ultraviolet (UV) light may degrade various pharmaceuticals to varying degrees. The carbon-iodine bond has been demonstrated to be photosensitive and readily cleaved [68]. However, UV photolysis alone investigated to treat ICM such as diatrizoate [69], iohexol [70], and iopamidol [71] led to lower removal efficiencies compared to UV-based AOPs. The combination of UV photolysis and hydroxyl radical reactions includes UV/H_2_O_2_, UV/Peroxydisulfate, UV/chlorine, and UV/Fe^3+^, UV/H_2_O_2_/Fe^3+^. Among them UV/Fe^3+^, UV/H_2_O_2_/Fe^3+^ will be described below in the Fenton part. The general mechanism of UV-based AOPs can be simply summarized by the generation of the fast reacting •OH by UV radiation.

##### UV/H_2_O_2_

UV/H_2_O_2_ as a typical approach for disinfection and organic contaminants degradation in water matrices has been widely investigated [72]. The addition of H_2_O_2_ can improve the removal efficiency due to the formation of •OH. ICM removal during UV/H_2_O_2_ is achieved by two major reaction pathways: direct photolysis by UV irradiation at 254 nm and oxidation by the non-selective hydroxyl radicals formed in situ [73].
(3)$$H_{2}O_{2}+\mathrm{hv}\;\to \;2\;•\mathrm{OH}$$

The degradation of ICM was studied in UV-based photolysis processes (at 254 nm with UV alone and UV/H_2_O_2_). Around 40% of 0.5 μmol L^−1^ diatrizoate was degraded through direct photolysis at UV fluence of 160 mJ cm^−2^. However, the degradation efficiency of diatrizoate was only slightly accelerated while adding H_2_O_2_, even if the concentration of H_2_O_2_ never exceeded the “economic” range (i.e., excess H_2_O_2_ would consume •OH and compete with the degradation of ICM [74]) [75]. An improvement in the degradation yield was nevertheless observed for other ICM, such as iohexol [76]. Although no appreciable mineralization occurred during UV/H_2_O_2_ treatment of iopromide, the by-products formed after UV/H_2_O_2_ treatment were biodegradable [77], which indicated that coupling with a biological treatment would be possible for this ICM.

##### Chlorine as Co-Reactant with UV

UV/chlorine is considered an interesting alternative to the UV/H_2_O_2_ process, since it generates a series of radicals such as •OH, •Cl, •Cl_2_^−^ and •OCl formed by photolysis of hypochlorous acid (HOCl) and hypochlorite ion (OCl^−^) as follows [69]:
(4)$$\mathrm{HOCl}\;+\;\mathrm{hv}\to •\mathrm{OH}+•\mathrm{Cl}\;Ф=1.45\;\mathrm{mol}\;{\mathrm{Einstein}}^{-1}$$
(5)$${\mathrm{OCl}}^{-}+\mathrm{hv}\to {•O}^{-}+\mathrm{Cl}\;Ф=0.97\;\mathrm{mol}\;{\mathrm{Einstein}}^{-1}$$
(6)$$•\mathrm{OH}+\mathrm{HOCl}\;\to •\mathrm{OCl}+H_{2}O\;k_{1}=2.0\;\times \;10^{9}\;M^{-1}\;s^{-1}$$
(7)$${•\mathrm{OH}+\mathrm{OCl}}^{-}\;\to {•\mathrm{OCl}+\mathrm{OH}}^{-}\;k_{2}=8.8\;\times \;10^{9}\;M^{-1}\;s^{-1}$$
(8)$$•\mathrm{Cl}+\mathrm{HOCl}\;\to {•\mathrm{OCl}+\mathrm{Cl}}^{-}{+H}^{+}\;k_{3}=3.0\;\times \;10^{9}\;M^{-1}\;s^{-1}$$
(9)$${•\mathrm{Cl}+\mathrm{OCl}}^{-}\to {•\mathrm{OCl}+\mathrm{Cl}}^{-}\;k_{4}=8.2\;\times \;10^{9}\;M^{-1}\;s^{-1}$$
(10)$${•O}^{-}{+H}_{2}O\to {•\mathrm{OH}+\mathrm{OH}}^{-}\;k_{5}=1.8\;\times \;10^{6}\;M^{-1}\;s^{-1}$$
(11)$${•\mathrm{Cl}+\mathrm{Cl}}^{-}\to {•\mathrm{Cl}}_{2}^{-}\;k_{6}=6.5\;\times \;10^{9}\;M^{-1}\;s^{-1}$$

Some studies have explored the degradation kinetic of widely used ICM, such as diatrizoate [69,78], iohexol [70], iopamidol [71], and iopromide [79]. Although diatrizoate was not degraded by chlorine, around 90% removal yield was obtained by UV irradiation in ten minutes and it was completely removed after only three minutes UV/chlorination process [69]. The high removal efficiency of diatrizoate has also been reported by other authors [78,80]. However, the formation of carcinogenic chlorinated species, such as trihalomethanes and chloroacetonitrile (C_2_HCl_2_N) has been observed, requiring a post-treatment to decrease the toxicity of formed by-products.

##### Photocatalysis

The mineralization of ICM such as diatrizoate [81,82], iomeprol [83,84], iopamidol [85], and iopromide [84] has been studied by photocatalysis. Photocatalysts such as TiO_2_, ZnO, Fe_2_O_3_, CdS, and ZnS have been used owing to their low cost, small bandgap energy, high chemical stability, and low water solubility. Among them, TiO_2_ is the most widely used semiconductor photocatalyst [86]. The mechanism of degradation of organic pollutants by photocatalysis is well-known. Ultraviolet (UV) light with energy exceeding the TiO_2_ semiconductor bandgap energy (Ebg = 3.2 eV for TiO_2_ anatase for example, corresponding to λ < 385 nm) is used [81].

In a relatively early study, Tusnelda and Fritz [84] showed that the photocatalytic degradation of both iomeprol and iopromide can be achieved using different TiO_2_ powder in aqueous solutions. However, the results indicated that a fast transformation but not complete mineralization or deiodination of the iodinated organic compounds occurs. In the study of Fu-Xiang et al [85], UPLC-ESI-MS with the total ion chromatograms (TIC) mode and the selected ion monitoring (SIM) mode were applied for analyzing the photodegradation intermediates of iopamidol. The postulated structures for the photodegradation products of iopamidol during UV irradiation were summarized (Figure 3). The photocatalytic oxidation of diatrizoate led to a low mineralization of the molecule, but a stoichiometric amount of iodide ions could be released [81].

Table 3 summarizes the most recent examples on the treatment of ICM by UV-based AOPs.

#### 3.2.3. Fenton Process

##### Classical Fenton Process

The Fenton process was first discovered by Henry J. Fenton, who reported that H_2_O_2_ can be activated by iron salts to oxidize tartaric acid [89]. After more than a century of evolution and development, it has been widely studied in wastewater treatment due to its relatively low cost. The oxidation of organic pollutants leads to the formation of intermediate species, which can be further oxidized up to CO_2_, H_2_O, and inorganic salts. The overall process can be schematically described by the following paths:(12)$${\mathrm{Pollutants}+H}_{2}O_{2}\;\overset{{\mathrm{Fe}}^{2+}}{\to }\;\mathrm{intermediates}$$
(13)$${\mathrm{intermediates}+H}_{2}O_{2}\;\overset{{\mathrm{Fe}}^{2+}}{\to }{\;\mathrm{CO}}_{2}{+H}_{2}O+\mathrm{inorganic}\;\mathrm{salts}$$

The advantages of the Fenton process relatively to other oxidation techniques are the simplicity of equipment and the ease of implementation [90].

In recent literature, the oxidation of diatrizoate using the Fenton reactant was performed at 20 °C varying the initial concentrations of ferrous ions and hydrogen peroxide [91]. The removal of diatrizoate (3 mg L^−1^) obtained after 5 min reaction was around 41.5% at pH 3 using 0.5 mol L^−1^ H_2_O_2_ and 0.5 mmol L^−1^ Fe^2+^. A higher removal percentage (72.0%) of 5 mg L^−1^ diatrizoate was achieved when 0.1 mmol L^−1^ Fe^2+^ and 1.5 mmol L^−1^ H_2_O_2_ were used. Some authors noticed that the percentage of diatrizoate removal was more influenced by the initial hydrogen peroxide concentration than the initial Fe^2+^ concentration [87]. In addition, Fe^0^ was investigated as a catalyst to activate H_2_O_2_ for the elimination of iopamidol according to the following reactions: (14)$${\mathrm{Fe}}^{0}{+H}_{2}O_{2}{+2H}^{+}\to {\;\mathrm{Fe}}^{2+}{+2H}_{2}O$$
(15)$${\mathrm{Fe}}^{2+}{+H}_{2}O_{2}\to {\;\mathrm{Fe}}^{3+}{+•\mathrm{OH}+\mathrm{OH}}^{-}$$

A 100% removal efficiency of 2 mg L^−1^ iopamidol was achieved with zero-valent nanoscale iron (1 g L^−1^) in the presence of 1.0 mmol L^−1^ H_2_O_2_, whereas only 9.5% degradation was obtained in the absence of Fe^0^ after 120 min [92].

Although the Fenton process has been widely studied and was found to be relatively efficient for ICM treatments, it still has some identified drawbacks such as a limited optimum pH range (around pH 3), a large volume of iron sludge produced, and difficulties in recycling the homogeneous catalyst (Fe^2+^) [93]. In this regard, to achieve the generation of •OH from H_2_O_2_ at mild pH, exploring new practically-acceptable and economically-viable Fenton catalysts are being an interesting area. The following part will detail the modified Fenton process to treat ICM.

##### Modified Fenton

Fe-free Fenton-like system 

As illustrated in Figure 4, elements with multiple redox states such as chromium, cerium, copper, cobalt, manganese, and ruthenium decompose H_2_O_2_ into •OH through conventional Fenton-like pathways. 

Although some studies have examined the effects of the above catalysts on the degradation of organic pollutants, they have been rarely investigated for ICM removal. Iopamidol has been treated by zero-valent aluminum-activated H_2_O_2_. Compared to zero-valent-iron (E^0^ = −0.430 V/SHE), zero-valent-aluminum (E^0^ = −1.662 V/SHE) is a stronger reducing agent. Iopamidol (2 mg L^−1^) removal efficiencies reached 95% under pH=3 conditions with H_2_O_2_ (0.25 mmol L^−1^) and zero-valent aluminum (0.5 mmol L^−1^). However, no iopamidol removal occurred without zero-valent aluminum in the same conditions [95].

Photo-Fenton

The photo-Fenton process is a combination of the Fenton process and UV-vis radiation (λ < 600 nm) that enhances the formation of •OH by two additional reactions [96]:(i) Photo-reduction of Fe^3+^ to Fe^2+^ ions
(16)$${\mathrm{Fe}(\mathrm{OH})}^{2+}+h\nu \;\to {\mathrm{Fe}}^{2+}+•\mathrm{OH};\;\lambda <580\;\mathrm{nm}$$

2.(ii) Peroxide photolysis via shorter wavelengths


(17)
$${\;H}_{2}O_{2}+h\nu \;\to \;2\;•\mathrm{OH};\;\lambda \;<\;310\;\mathrm{nm}$$


UV-Fenton has been developed to degrade ICM. For example, a TOC removal efficiency of 67% for diatrizoate was obtained after 4h of photo-Fenton treatment using iron-activated carbon as a catalyst [97]. Another study performed by Polo et al. [93] showed that whereas direct photolysis was not efficient to degrade diatrizoate, 100% degradation was achieved with solar radiation/Fenton or Fenton-like [93]. The acute toxicity of the degradation byproducts generated by different processes (UV, UV/H_2_O_2_, UV/Fenton, UV/Fenton-like, UV/K_2_S_2_O_8_) was measured with *Vibrio fischeri* bacteria test. The results underlined the production of degradation byproducts that were more toxic than the original product diatrizoate.

Electro-Fenton

In the electro-Fenton process (EF), H_2_O_2_ is generated by 2-electron reduction of dissolved O_2_ (Equation (19)), and ferrous iron used as a homogeneous catalyst is electrochemically regenerated from Fe^3+^ formed during the Fenton reaction (Equation (20)).
(18)$$O_{2}+2H^{+}+2e^{-}\to H_{2}O_{2}$$
(19)$${\mathrm{Fe}}^{3+}+e^{-}\to {\mathrm{Fe}}^{2+}$$

The elimination of diatrizoate by EF using graphite felt cathode (18 cm × 5 cm × 0.5 cm) was recently studied and optimized parameters (1000 mA current intensity, 0.2 mmol L^−1^ Fe^2+^ and 1 L min^−1^ compressed air bubbled through the solution) allowed the complete mineralization of diatrizoate [97]. To decrease the energy consumption related to the electro-Fenton process, coupling with a biological treatment has also been explored for the treatment of a complex pharmaceutical mixture containing diatrizoate [98]. Adjusting the current density allowed a sufficient improvement of the biodegradability of the effluent, leading to total mineralization after a biological treatment with acclimation of the biomass. This result is very promising for the cost-effective treatment of real effluents concentrated in pharmaceuticals.

#### 3.2.4. Electrochemical AOPs

Electrochemical methods constitute a distinct group of AOPs methods because they involve reactions induced by electric current passing across the interface between an electrochemical-controlled working electrode, or a system of electrodes, and the solution. The mechanism of degradation can involve a direct interaction of the target compound with the electrode surface leading to electrons released or consumed at the interface. Another mechanism may result in the formation of oxidants or reductants, which react with the target contaminant in the bulk solution but not necessarily at the electrode/solution interface; such modes of electrochemical treatment can be termed indirect [99]. We focus this part on electrooxidation processes since electro-Fenton treatment has already been developed in Section 3.2.3. and electrochemical processes are investigated in Section 3.3. 

Anodic oxidation can involve: a direct electron transfer to the electrode surface M, heterogeneous reactive oxygen species (ROS) produced by the oxidation of water, including physiosorbed •OH at the anode surface M(•OH), (Equation (20)), oxidants such as H_2_O_2_ produced from M(•OH) dimerization (Equation (21)) and O_3_ (Equation (22)), and other oxidant agents electrochemically produced from ions existing in the bulk [100].
(20)$${M+H}_{2}O\;\to \;M\left(•\mathrm{OH}\right){+H}^{+}{+e}^{-}$$
(21)$$M\left(•\mathrm{OH}\right)\to {\;2\;M+H}_{2}O_{2}$$
(22)$${3\;H}_{2}O\;\to {\;O}_{3}{+6\;H}^{+}{+6\;e}^{-}$$

The ability of anodic oxidation is deeply dependent on the anode material. Several anode materials have been reported for the degradation of ICM during the last twenty years and are discussed thereafter.

Generally, there are two types of anode materials named active anodes with low oxygen overpotential and non-active anodes with high oxygen overpotentials. The higher potential for O_2_ evolution is responsible for the weaker interaction of •OH with anode surface and the higher chemical reactivity towards organic pollutants.

Ruthenium dioxide (RuO_2_), iridium dioxide (IrO_2_), platinum (Pt), graphite, and other sp^2^ carbon-related electrodes normally exhibit potentials of O_2_ evolution lower than 1.8 V/SHE, as can be seen in Table 4. Lead dioxide (PbO_2_), tin dioxide (SnO_2_), boron-doped diamond (BDD), and sub-stoichiometric TiO_2_ electrodes present potentials of O_2_ evolution from 1.7 to 2.6 V/SHE [101]. Among them, BDD as one of the most promising anodes has been applied for the degradation of ICM in recent years.

##### BDD Anode

Recently some studies reported the removal of ICM by electrooxidation on BDD electrodes because of their high chemical, mechanical, and thermal stability as well as their high overvoltage. The most attractive advantage of BDD anodes over other anode materials is their superior capability to degrade persistent organic contaminants, owing to the formation of several radicals, such as •OH, •Cl, SO_4_^•−^ [104].

In Schneider et al study [105], six ICM (iotalamic acid, iopamidol, iohexol, iopromide, iomeprol, diatrizoate) have been degraded on BDD electrode in 0.01 mol L^−1^ Na_2_SO_4_. The results showed that ICM were completely deiodinated, leading to iodate ions as the oxidation product. The dissolved organic carbon decreased with a yield ranging from 30 to 80% among the six ICM, showing partial mineralization. Beyond this approach, recently, electrochemical activation of sulfate ions to sulfate radical species and activated persulfate have been demonstrated at BDD, which enhanced the electrooxidation kinetics of iopromide and diatrizoate [104]. In the case of diatrizoate, 80% of deiodination efficiency was obtained. The efficiency of the electrooxidation process was higher than with iopromide. This result can be explained by the occurrence of the oxidation process on the alkyl side chains since iopromide contains alkyl side chains with steric hindrance. The analysis of by-products showed that the main mechanistic steps in the oxidation of iopromide were H-abstraction and bond cleavage of the alkyl side chains, whereas diatrizoate was transformed through oxidative cleavage of iodine substituent and inter-molecular cyclization.

##### DSA Anode

Except for the BDD anode, Dimensionally Stable Anode (DSA), also called mixed metal oxide electrode, has high conductivity and corrosion resistance for use as anodes in the electrooxidation of ICM. They are generally prepared by thermal deposition of a thin layer (a few micrometers) of metal oxides (RuO_2_, IrO_2_, SnO_2_, PbO_2_, etc.) on a base metal, such as Ti, Zr, and Ta [106]. The electrochemical degradation of six ICM was investigated in a batch experiment performed under constant current conditions using two DSA electrodes (titanium coated with a mixed metal oxide solution of precious metals such as iridium, ruthenium, platinum, rhodium, and tantalum). The results showed that the removal yield was never below 85% when the oxidation was performed with a current density of 64 mA cm^−2^ within 150 min. The analysis of the degradation by-products showed three possible degradation pathways: the reductive deiodination of the aromatic ring, the de-alkylation of aromatic amides to unsubstituted amides, and the de-acylation of *N*-aromatic amides to produce aromatic amines [107].

#### 3.2.5. Other AOPs

Advanced oxidation processes including ultrasound, plasma, and electron beam irradiation remove organic pollutants in all three—gas, liquid, and solid—states and became of interest in recent years. In this part, the use of these types of AOPs for ICM degradation is detailed.

##### Ultrasound

Sonolysis of an aqueous solution leads to the formation and collapse of cavitation bubbles, which generate locally high temperatures and pressures and reactive free radicals upon their collapse. Water vapor under these extreme conditions undergoes a thermal dissociation to form highly reactive radical species, such as •OH. A comparison of ultrasound irradiation in the presence or in the absence of •OH (using *tert*-BuOH scavenger of •OH) was performed for the degradation of ICM. The results showed that around 70% of the degradation of ICM was performed by •OH radical attack, and 30% by thermal decomposition [108].

##### Plasma

Non-thermal plasma in liquid and gas-liquid environments generates in situ oxidizing species, such as hydroxyl radicals, ozone, hydrogen peroxide, and peroxynitrites. The plasma is generally generated in the gas phase in contact with a thin layer of liquid, with a dielectric barrier or corona discharges [109]. The removal of iopromide from an aqueous solution using dielectric barrier discharge has been studied [110]. The removal efficiency reached 98.8%. No decrease in total organic carbon was observed but the by-products of iopromide were easily adsorbed and biodegraded by activated sludge. The degradation of diatrizoate with corona plasma was also conducted [111]. The authors found that hydroxyl radicals were primarily responsible for the decomposition steps. Diatrizoate was the most stable and the slowest to be decomposed among the seven investigated recalcitrant pharmaceutical residues, with a degradation yield of only 50% after 1 h. The low energy efficiency of this kind of process is still a big challenge, which might be improved by its coupling with cost-effective methods if applied to real wastewater treatments.

##### Electron Beam Irradiation

Since the electron beam irradiation process in water can induce not only oxidation but also reduction reactions, this technology can be interesting to treat halogenated compounds. The decomposition of water under electron beam irradiation occurs with the production of ^•^OH, ^•^H, H_2_O_2_, H_3_O^+^, H_2_, and electrons [112]. Electron beam irradiation has been investigated to treat ICM compounds. Velo, et al. [112] applied gamma irradiation for the removal of diatrizoate from different water matrices. The removal efficiency of up to 91% was obtained at an absorbed dose of 1000 Gy (Gray) but the TOC value showed that no mineralization occurred. In the study of Kwon et al. [113], E-beam with an absorbed dose of 19.6 KGy was required to achieve 90% degradation of 100 μmol L^−1^ iopromide. According to the authors, in the mechanism, the role of •OH is low since a high degradation yield could be obtained even in the presence of •OH scavenger. A kinetic study underlined the important role of hydrated electron (e^−^_aq_) in the degradation of iopromide.

##### AOPs Involving the Activation of Sulfate Radicals

New methods allowing the activation of persulfate to generate SO_4_^•−^ have emerged recently such as those with iron and transition metals to treat ICM. Without activation, the persulfate anion reacts with ICM only at a negligible degree because of its low oxidation potential. Different activators of persulfate anions were studied in the literature for ICM degradation, as presented below.

Iron-activated persulfate

Iron has been widely studied as a persulfate activator in ICM degradation since it is relatively non-toxic, environmentally friendly, and cost-effective. Ferrous iron (Fe^2+^) can react with persulfate to form the sulfate radical (Equation (23)) [114].
(23)$$S_{2}O_{8}^{2-}{+\mathrm{Fe}}^{2+}\to {\mathrm{Fe}}^{3+}{+\mathrm{SO}}_{4}^{2-}{+•\mathrm{SO}}_{4}^{-}\;\;k\;=2.0\;M^{-1}s^{-1},\;22\;{}^{\circ}C\;$$
(24)$$•{\mathrm{SO}}_{4}^{-}+H_{2}O\;\to {\;\mathrm{SO}}_{4}^{2-}+H^{+}+{\mathrm{OH}}^{•}$$

Zero valent iron (Fe^0^) can also produce Fe^2+^ through oxidation in the presence or absence of oxygen according to the following equations:(25)$${\mathrm{Fe}}^{0}+0{.5\;O}_{2}{+H}_{2}O\to {2\;\mathrm{Fe}}^{2+}{+4\;\mathrm{OH}}^{-}$$
(26)$${\mathrm{Fe}}^{0}{+2\;H}_{2}O\to {\mathrm{Fe}}^{2+}{+2\;\mathrm{OH}}^{-}{+H}_{2}$$

Persulfate could also directly react with Fe^0^ and release Fe^2+^ as follow:(27)$${\mathrm{Fe}}^{0}{+S}_{2}O_{8}^{2-}\to {\mathrm{Fe}}^{2+}{+2\;\mathrm{SO}}_{4}^{2-}$$

The degradation of diatrizoate [115], iohexol [78], and iopamidol [116] has been examined in Fe^2+^-activated persulfate systems. Results showed that for 10 mmol L^−1^ persulfate with an n(persulfate)/n(Fe^2+^) ratio of 1:10 and an initial pH of 3.0, the diatrizoate abatement was around 30% in 2h treatment. With only persulfate or Fe^2+^, an abatement close to 10% was found in the same conditions. A study with SO_4_^•−^ and •OH scavengers showed that the degradation of diatrizoate was mainly attributed to •OH produced by the reaction of sulfate radical with water (Equation (25)) [115]. Similar conclusions were also considered for iopamidol degradation [116]. For iohexol (10 μmol L^−1^), 80% degradation was achieved with 0.4 mmol L^−1^ persulfate and Fe^2+^ in 5 min, whereas it was not possible with Fe^2+^ or persulfate alone. The effect of water matrices was also investigated, showing that a low concentration of chloride ions promoted iohexol degradation, whereas natural organic matter slightly inhibited it due to the scavenging capacities of radicals [78].

Co^2+^ activated persulfate

Dionysiou and co-workers [117] investigated nine transition metals for the generation of sulfate radicals through the decomposition of peroxymonosulfate (PMS), and found that cobalt ion (Co^2+^) was the most effective homogenous catalyst for activating sulfate radicals, according to the following reaction:(28)$${\mathrm{Co}}^{2+}{+\mathrm{HSO}}_{5}^{-}\to {\mathrm{Co}}^{3+}{+•\mathrm{SO}}_{4}^{-}{+\mathrm{OH}}^{-}$$

The degradation of iomeprol and iohexol by sulfate radical generated through Co(II)-mediated activation of peroxymonosulfate had been concomitantly investigated [118]. The low TOC removal efficiency and the identification of iodinated by-products elucidated that iomeprol and iohexol were mostly degraded via deiodination and transformation of side chains without total mineralization. The presence of humic acid usually found in waterbodies remarkably inhibited the oxidation of iohexol, while the effect of Cl^−^ was negligible.

Metal oxide-activated persulfate

According to the experimental results of Ji et al. [119], the mechanism of activation of PMS by CuO may involve the following reactions:(29)$${\mathrm{Cu}}^{2+}{+\mathrm{HSO}}_{5}^{-}\to {\mathrm{Cu}}^{+}{+\mathrm{SO}}_{5}^{-}{+H}^{+}$$
(30)$${\mathrm{Cu}}^{+}{+\mathrm{HSO}}_{5}^{-}\to {\mathrm{Cu}}^{2+}{+(•\mathrm{SO}}_{4}^{-}{+•\mathrm{OH}}^{-}{)\;\mathrm{or}\;(\mathrm{SO}}_{4}^{2-}+•\mathrm{OH})$$
(31)$${•\mathrm{SO}}_{4}^{-}+H_{2}O\to {\mathrm{SO}}_{4}^{2-}+•\mathrm{OH}+H^{+}$$

Cupric oxide (CuO) and manganese dioxide (MnO_2_) were investigated by Hu et al. [120] as catalysts to activate PMS to degrade iopamidol. Both CuO and MnO_2_ catalyzed the total degradation of iopamidol in 60 min, but the degradation rate with CuO/PMS was 3.7 times higher than with MnO_2_/PMS. The majority of iodine released from iopamidol was oxidized to iodate (IO_3_^−^) and a small fraction of the initial total organic iodine was transformed to iodoform (CHI_3_). More recently, a series of CuFeO_4_ catalysts showed high catalytic activity to activate PMS for the removal of iohexol [121]. A 66% mineralization yield of iohexol was obtained, about 30% higher than for CuO/PMS and α-Fe_2_O_3_/PMS systems. Twelve intermediates were identified by UHPLC-QTOF/MS, showing that the degradation mechanism involved alcohol oxidation, decarboxylation, amide hydrolysis, deiodination, and amino oxidation.

UV/Peroxydisulfate

Another nonselective powerful oxidant is SO_4_•^−^ (E^0^ = 2.6 V/ESH) [59,88,122], which can be activated from peroxydisulfate (PDS, S_2_O_8_^2−^) by different treatments such as UV, heat, and transition metals. Different studies used UV/PDS to degrade ICM. Duan et al. [75] found that the degradation rates of diatrizoate was significantly enhanced in UV/S_2_O_8_^2−^ system compared to UV/H_2_O_2_ (the removal efficiency increased from 40% to 100%), which suggested the potential vulnerability of diatrizoate towards UV/S_2_O_8_^2−^ treatment. The authors explained this higher efficiency from the higher radical quantum yield of SO_4_•^−^ and the accelerated electron transfer for SO_4_•^−^ reaction by the released I^−^, used as a mediator. In addition, SO_4_•^−^ can generate •OH according to Equation (24) [74].

Figure 5 shows the proposed oxidation mechanisms of diatrizoate by SO_4_•^−^ [59], involving deiodination and decarboxylation. The higher activity of SO_4_•^−^ for the degradation of diatrizoate has been confirmed by another study [59]. The irradiation of diatrizoate in the presence of H_2_O_2_ led to a removal yield of 17%, whereas 42% of diatrizoate was degraded when using UV/PDS after 1 h [59]. Nonionic ICM, iopromide, can be completely removed in 30 min and almost complete mineralization has been noticed after 80 min of treatment [88]. Its analog iohexol can also be efficiently mineralized to CO_2_ by UV/S_2_O_8_^2−^, however, the authors also reported the formation of toxic iodoform CHI_3_ during the treatment [74].

A successful UV-A and UV-C photoassisted persulfate oxidation of iopamidol (initially at 2.5 ppm) were conducted in a real, tertiary-treated municipal wastewater [123]. In presence of 0.2 mM of persulfate, 48% of mineralization was obtained in 120 min at pH 7. 

In their study, Dong et al. applied UV-C laser irradiation for the removal of three ICM (iohexol, iopamidol, and diatrizoate) initially at 10 μM at pH 7 [124]. An initial persulfate concentration of 1 mM was sufficient to obtain 60% ICM degradation in 40 s. Sulfate radical was found to be the main active species for ICM degradation. To explain their choice of UV irradiation for persulfate activation, the authors argued the absence of secondary pollution linked to the presence of metal or metal oxide sometimes used in the generation of sulfate radicals.

### 3.3. Reductive Deiodination for Removing ICM

The recalcitrance of persistent halogenated organic compounds is usually partly attributed to carbon-halogen bonds and dehalogenation can improve their biodegradability [125,126,127]. The reductive deiodination of ICM might be an effective method to convert them into less toxic compounds before biological treatment. It would have several advantages: (1)The degradation of ICM by AOP_S_ can lead to the formation of toxic intermediates due to the absence of selectivity of this class of processes, whereas electro-reductive dehalogenation is selective.(2)Reductive deiodination allows the easy recovery of iodide ions, which is very promising to achieve a cost-effective treatment on the production sites, owing to the wide consumption of ICM over the world.

#### 3.3.1. Electroreductive Deiodination

In recent years, halogenated compounds have been successfully dechlorinated with different types of electrode material [125,127,128,129,130,131,132,133,134]. Electroreduction is considered a green technique since it does not require the addition of a chemical reducer. The cleavage of the carbon-iodine bond is known to require less energy compared with the cleavage of carbon-bromine and carbon-chlorine bonds [135].

The electro-deiodination of ICM strongly depends on the electrode material and on the molecular structure of ICM. Iopamidol and diatrizoate were electro-reduced at potentials lower than −0.45 V/SCE (i.e., −1.1 V vs. SHE) at a gold rotating ring-disc electrode [136]. Electrodeiodination of iomeprol was achieved at −1 V vs AgCl/KCl_aq_ with a nickel foam electrode [137]. Diatrizoate could also be reduced on bare graphite felt and Pd nanoparticle-loaded graphite felt with applied potentials ranging from −1.1 V to −1.7 V vs SHE [19,20]. Molecular complexes have also shown interesting catalytic activities toward the reduction of ICM, such as organometallic complexes based on corrinoids. Cobalt corrinoids contain three different oxidation states (+I, +II, and +III), which allow them to function as an electron shuttle between a reducing agent and ICM. Cyanocobalamin (vitamin B_12_) and dicyanocobinamide have shown good catalytic properties toward the reductive deiodination of three ICM (iopromide, iopamidol, and diatrizoate) in the presence of titanium citrate as a reducing agent [138,139]. Whereas the control tests without corrinoid showed a low degradation of ICM, complete deiodination was achieved in the presence of vitamin B_12_ and dicyanocobinamide. Iohexol has been also totally deiodinated by electrochemical reduction on graphite felt, improved when Vitamin B12 was used as a redox catalyst [140]. All these processes can achieve a total removal efficiency under optimal operating conditions and three deiodinated by-products (one, two, and three cut iodine-carbon bonds) were detected. However, the coulombic efficiencies decreased when very negative potentials were applied due to competition with hydrogen evolution. Electrodes with high hydrogen evolution overpotential can also be used to enhance the biodegradability of by-products resulting from the electroreduction of ICM [141]. It allows the electro-reduction of both carbon-iodide bounds and amido groups, leading to fewer biorecalcitrant by-products.

#### 3.3.2. Metallic Reductants

Since it is a highly reactive, cost-effective, and environmentally friendly material, zero-valent-iron (ZVI) has been extensively employed to remove various contaminants from groundwater by releasing electrons to the target compound. The dehalogenation by ZVI can be achieved according to the two following mechanisms [142]:

- Direct electron transfer from zero-valent-iron at the surface of the metal:
(32)$${\mathrm{Fe}}^{0}{+\mathrm{RX}+H}^{+}\to {\mathrm{Fe}}^{2+}{+\mathrm{RH}+X}^{-}$$

- Catalytic hydrogenation by atomic hydrogen H_ads_ formed either by reduction of H_2_O or acidic iron corrosion
(33)$${\mathrm{Fe}}^{0}{+2H}^{+}\to {\mathrm{Fe}}^{2+}+2H_{\mathrm{ads}}$$
(34)$${2H}_{\mathrm{ads}}+\mathrm{RX}\;\to {\;\mathrm{RH}+H}^{+}{+X}^{-}$$

To determine the ability of ZVI to deiodinate ICM, several studies have been performed. At pH 3 with 40 g L^−1^ granular iron and with a particle size of 0.125–3.000 mm, 100 mg L^−1^ iopromide and diatrizoate were reduced, leading to 95% and 60% degradation yield after 8 hours of treatment, respectively. The proposed by-product structure of iopromide after ZVI treatment was a completely deiodinated compound with shortened side chain compared with the parent compound [142]. Another study shows that iopamidol (10 μmol L^−1^) was not completely degraded (68%) after 72h with 5 g L^−1^ ZVI at pH 7.0, although the adsorption of iopamidol to ZVI was negligible [143]. However, rapid and complete removal of iopromide (30 mg L^−1^) was achieved with 0.3 g L^−1^ Fe after only 15 minutes at pH = 2. The concentration of released I^−^ compared with the iopromide concentration indicated that the deiodination was not complete [144]. It seems that the deiodination of iopromide by ZVI is a stepwise and not a simultaneous process and a longer time would contribute to its total deiodination. It has also been reported that the addition of humic acid prevented the passivation of ZVI surfaces, enhancing the reductive removal of diatrizoate [145].

Recently, it has been shown that the cheap and green thiourea dioxide reductant coupled with trace Cu(II) allowed the degradation of diatrizoate [146]. The mechanism involved a Cu(I) intermediate that reduced diatrizoate into several by-products including deiodinated derivatives. The in situ formation of Cu(I) has been also achieved with the combination of copper ion and sulfite (Cu(II)/S(IV)) under anaerobic conditions and led to the degradation of diatrizoate, mainly by deiodination [147].

#### 3.3.3. Biogenic Metal Nanoparticles as New Catalysts for the Hydrogenation of ICM

Bacterial reduction of Pd (II) and Pt (II) is a green and cost-effective method to generate palladium and platinum nanoparticles. These biogenic particles have emerged as catalysts for the hydrogenation reaction of ICM. Indeed, 20 mg L^−1^ diatrizoate dissolved in a 10 mg L^−1^ bio-Pd nanoparticles suspension was completely hydrodeiodinated at pH = 4 within 4 h [148]. Electroreduction of diatrizoate (2 mg L^−1^) was also carried out with bio-Pd nanoparticles immobilized in a graphite cathode of a microbial electrolysis cell, where H_2_ was produced. Diatrizoate was completely deiodinated within 24 h at −0.4 V/SHE and the resulting compound, 3,5-diacetamidobenzoate, could be biotransformed by nitrifying sludge [149]. Catalytic deiodination of ICM by biogenic metal nanoparticles can be then a reliable alternative due to its effectiveness and selectivity.

#### 3.3.4. Coupled Processes for ICM Removal

Many ICM can be reduced by different reductive processes to form deiodinated products. However, the total removal of iodine atoms from ICM structure may not be sufficient to make them easily biodegradable. Sequential treatments involving a reductive deiodination step followed by another physico-chemical treatment may allow a good degradation of ICM in a cost-effective manner. For example, Radjenovic et al. [19] studied a coupled process between the electro-reduction of diatrizoate to 3,5-diacetamidobenzoic acid on graphite felt doped with palladium and electrooxidation on BDD electrode. Interestingly, 3,5-diacetamidobenzoic acid was more readily mineralized than diatrizoate and no stable iodinated intermediates were formed. The reduction-oxidation combined approach has also been used for removing diatrizoate with nano-sized zero-valent iron (nZVI) under aerobic conditions [150]. First, electrons were transferred from nZVI to diatrizoate, leading to its total deiodination. Then nZVI was oxidized by oxygen dissolved in solution to form ferrous ions and hydrogen peroxide under acidic conditions, giving rise to the generation of •OH by the Fenton reaction. The resulting free radicals attacked the amide groups on diatrizoate, leading to its deacetylation.

The coupling of two selective electrochemical treatments has been successfully applied to the removal of diatrizoate [20]. Whereas a biological treatment with activated sludge over a period of 21 days led to a low mineralization yield of diatrizoate (5%), this value increased to 41% after electro-reduction on graphite felt and to 60% after an electrochemical treatment combining electro-reduction with a subsequent electrochemical oxidation step performed at 1.3 V/SCE. 

### 3.4. By-Products Identification during the Degradation of ICM

By-products formed after AOP_S_ and reductive treatments of ICM have been determined by different approaches mainly based on LC-MS. The proposed structures of by-products are summarized in Table 5, Table 6, Table 7, Table 8 and Table 9 for diatrizoate, iohexol, iomeprol, iopamidol and iopromide. The structures of the intermediates were different even for similar advanced oxidation processes, underlining the non-selectivity of AOP_S_. Although electrochemical reduction usually led selectively to the deiodinated derivatives, it is interesting to note that other by-products can be formed in some cases. For this reason, the structure of by-products formed during non-selective reductive processes are also given.

## 4. Conclusions

In this bibliographic review, an overview of ICM and their removal was presented. As these compounds are administered in a high dose and excreted rapidly from the body, high concentrations can be found in effluents such as hospital wastewater.

Research concerning biodegradation showed only the biotransformation of such compounds even if low concentrations were considered. Due to the fact that these organic compounds are not mineralized in biological processes, their release can cause diffuse pollution in the environment.

Their removal by advanced oxidation processes was largely studied but mineralization was also hard to achieve. Some ICM, such as diatrizoate, seemed to be recalcitrant to hydroxyl radical attack. In this case, oxidation by means of sulfate radical was more efficient. Radical attacks are non-selective and can entail the formation of toxic intermediate especially when the mineralization is partial. However, recent results have shown that AOPs such as electro-Fenton, operating in optimal conditions can lead to total mineralization of ICMs such as diatrizoate. Coupling with a biological treatment is also possible to reduce the cost of the treatment. These results are very promising for the cost-effective treatment of real effluents.

As the presence of a halogen atom in a molecule participates in its biorecalcitrance, ICM dehalogenation could also improve the biodegradability of the molecule and then help further mineralization by biological process. Moreover, the iodide ions recovery could be an advantage from an economic point of view, especially for the treatment of industrial wastewater on ICM site production. However, it has been shown that ICM-deiodinated products cannot be totally mineralized by a classical treatment with activated sludge. For the efficient implementation of a coupling process involving a reductive method and a biological treatment, the reduction of other chemical groups, such as amido groups, is required, which is more challenging and energy-consuming.

## Figures and Tables

**Figure 1 molecules-28-00341-f001:**
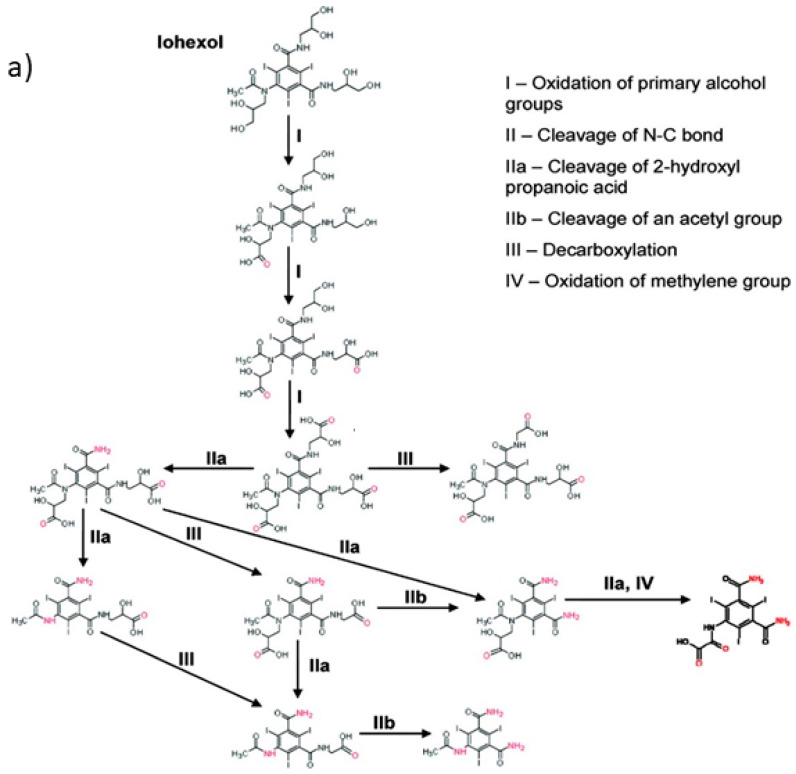
Proposed microbial transformation pathway (TP: transformation products) of (**a**) iohexol and (**b**) iopamidol [52] Reprinted with permission from ref [52]. Copyright 2010 American Chemical Society.

**Figure 2 molecules-28-00341-f002:**
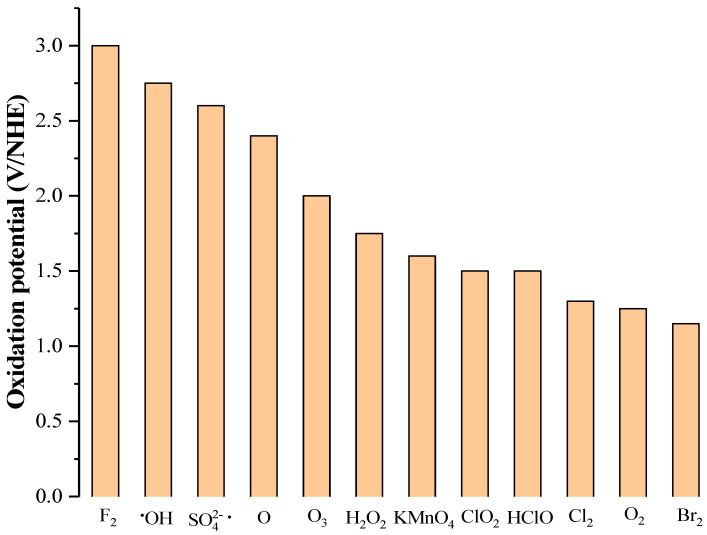
Standard oxidation potentials of common oxidants [58,59].

**Figure 3 molecules-28-00341-f003:**
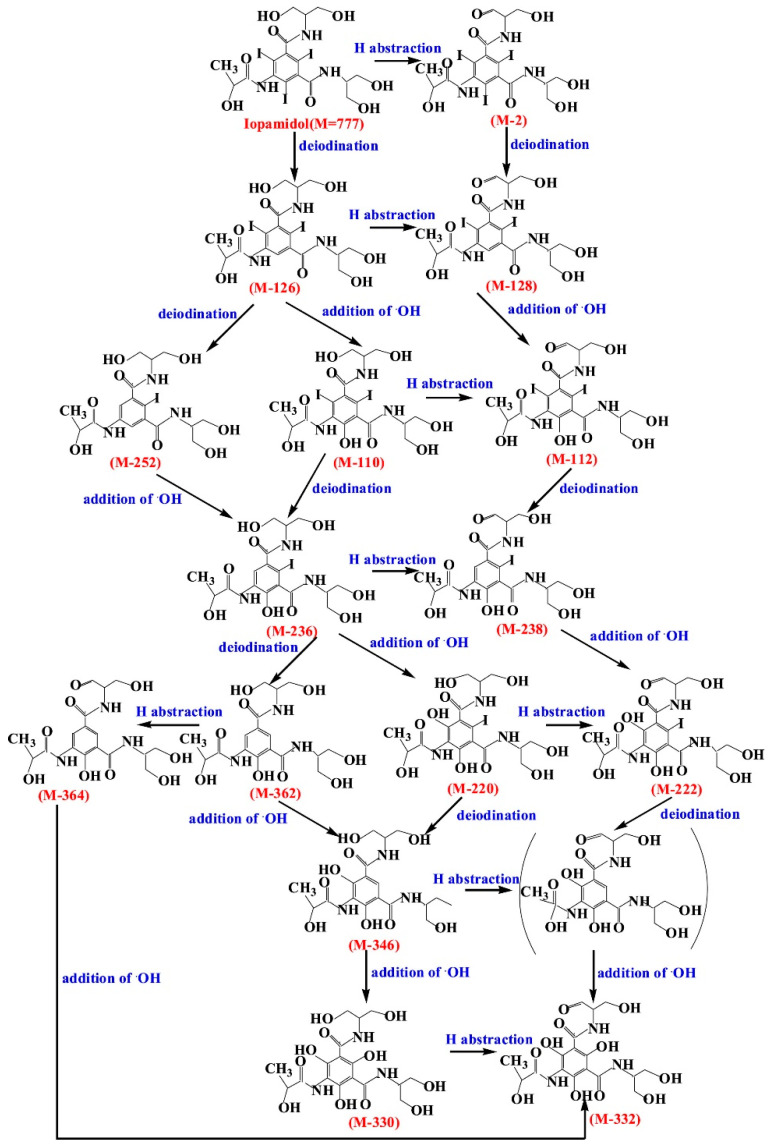
Proposed destruction pathways of iopamidol during UV irradiation [85]. Reprinted from ref [85], Copyright 2014, with permission from Elsevier.

**Figure 4 molecules-28-00341-f004:**
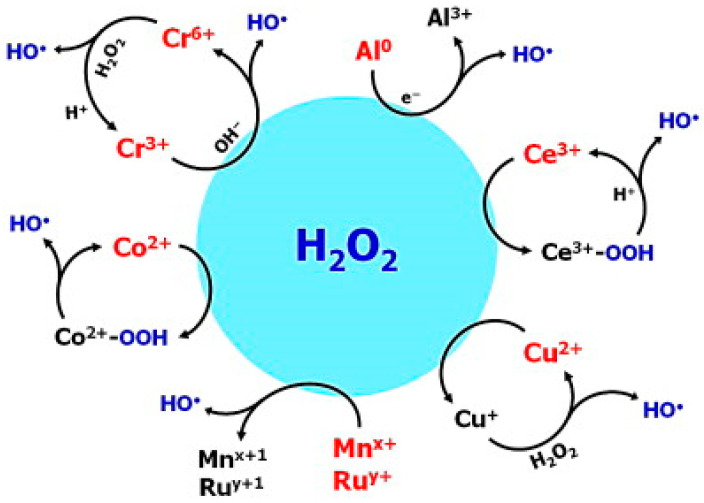
Schematic illustration of H_2_O_2_ activation mechanisms using different nonferrous Fenton-type reagents [94]. Reprinted from ref [94], Copyright 2014, with permission from Elsevier.

**Figure 5 molecules-28-00341-f005:**
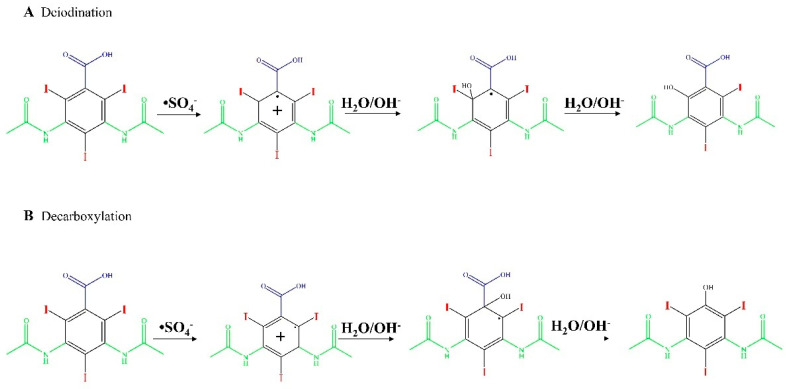
Proposed pathways of oxidation of diatrizoate by SO_4_•^−^ attack [59].

**Table 1 molecules-28-00341-t001:** Chemical structure and physicochemical properties of ICM.

	Diatrizoic Acid	Iohexol	Iopamidol	Iopromide	Ioxithalamic Acid	Iomeprol
Structure	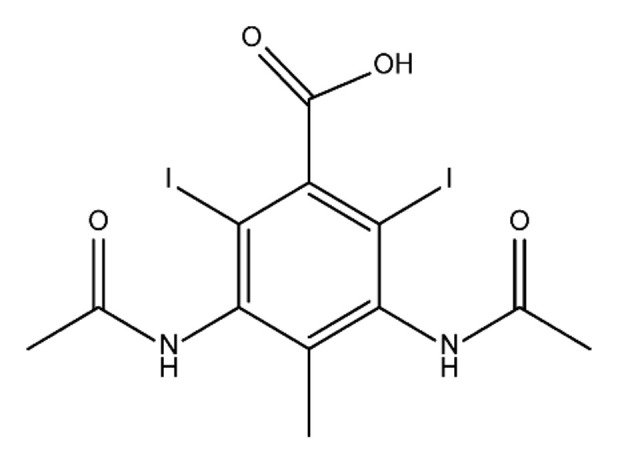	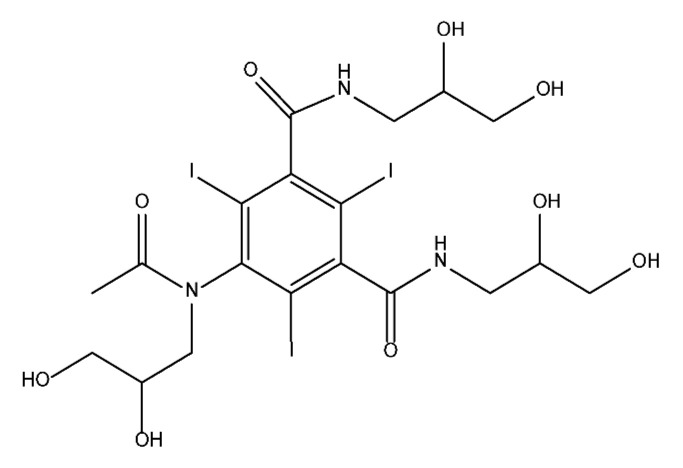	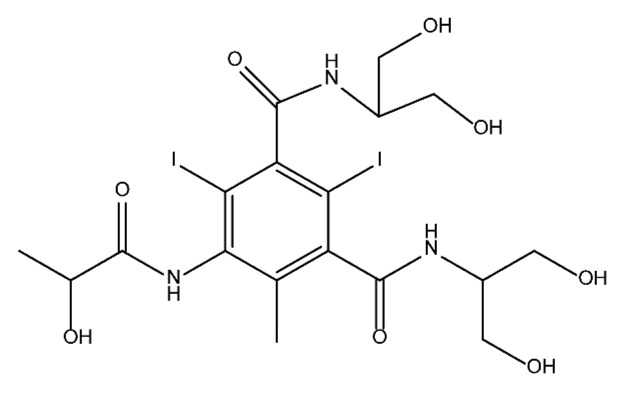	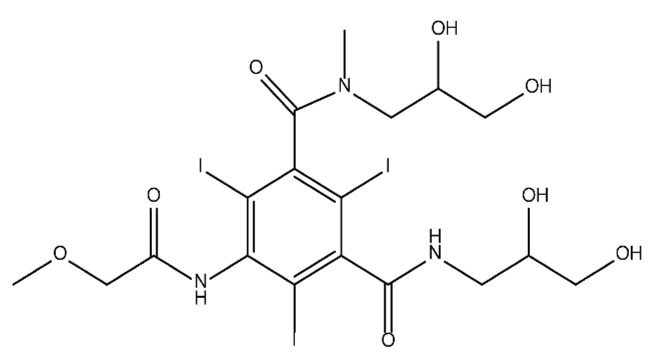	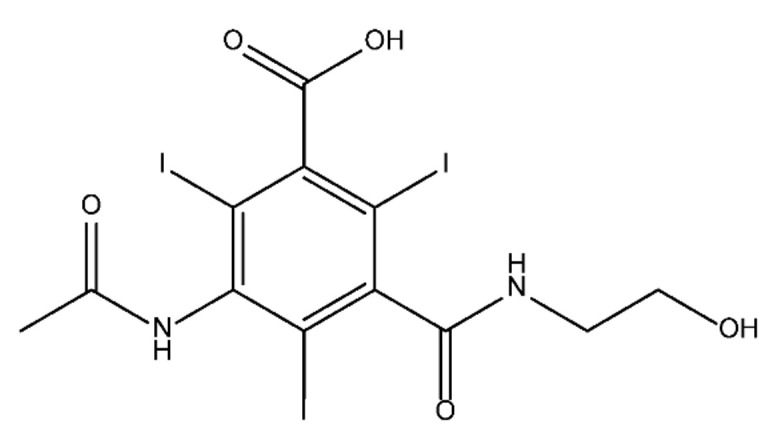	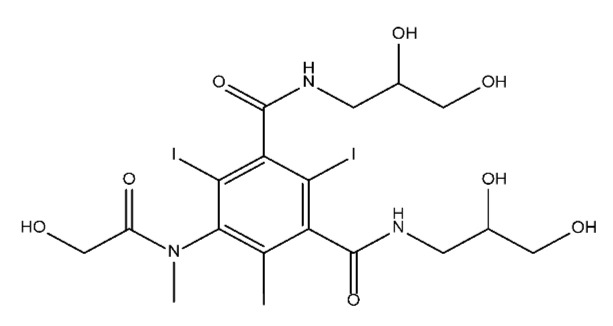
Log K_ow_ ^a^	1.8	−3	−2.4	−2.1	1.2	−2.3
Ionicity	Ionic	Nonionic	Nonionic	Nonionic	Ionic	Nonionic
pKa	2.17	11.73	11.00	11.09	2.13	11.73
Concentrations ^b^ (μg L^−1^)						
Hospital wastewater	17.1–61	0.07–3810	0.03–2599	0.008–3.2	15–550	0.05–2400
Surface water	0.032–4.55	0.01–1.326	0.008–3.2	0.01–13	0.01–0.438	0.023–6.1
Ground water	0.02–9.6	0.003–0.187	0.006–0.47	0.003–0.687	0.204	0.003–1.655
Drinking water	0.0009–1.2	0.001–0.034	0.02–0.27	0.0005–0.084	-	0.0013–0.034

^a^ [8] ^b^ Results in Europe from 2001 [27].

**Table 2 molecules-28-00341-t002:** Examples of treatment of ICM by biological processes.

Entry	Contaminant	Microorganism	Type of Reactor	Concentration	pH	T (°C)	Degradation Time	Conclusions	Refs.
1	Iopromide	Nitrifying Activated Sludge	Batch (flask containing 5 L biomass with liquor-suspended solids)	0.10–0.27 g μL^−1^	7.5	-	96 h	In a full-scale municipal wastewater treatment plant (WWTP), iopromide (0.10 to 0.27 μg L^−1^) was removed at an efficiency of 61% and has a higher solid retention time (49 days) than conventionally activated sludge (6 days)	[49]
2	Diatrizoate Iopromide	Activated sludge	250 mL gas scrubbing cylinders + 100 mL activated sludge	1.5 nmol L^−1^ 1.85 nmol L^−1^	6.0-7.5	Room	54 h	Degradation of DTR was negligible. Around 85% of iopromide was transformed into two metabolites.	[51]
3	Diatrizoate	Activated Sludge 200 mg L^−1^ from municipal sewage treatment plant	Guidelines for Testing of Chemicals: 302 B Zahn-Wellens Test and 301 A Confirmatory Test, adopted by the Council on 17 July 1992	0.14 mg L^−1^	6.0-7.5	21-25	30 days	Diatrizoate was biotransformed into 2,4,6-triiodo-3,5-diamino-benzoic acid in the modified Zahn-Wellens test	[50]
4	Diatrizoate Iohexol Iomeprol Iopamidol	Aerobic soil-water and river sediment	Batch system (Braunschweig soil, Braunschweig soil+LUFA 2.2 soil, river sediment)	1 mg L^−1^	Neutral	20–22	0–104 days	Diatrizoate was not biotransformed Three other nonionic ICM were transformed into several biotransformation products.	[52]
5	Diatrizoate	Agricultural field soil sediment from a sulfate reducing zone of a polishing pond groundwater	Batch (500 mL amber glass in glovebox, acetate and Fe(III) were added)	100 μg L^−1^	-	22	20 days	Seven by-products resulted from successive deiodinations and deacetylations. 3,5-diaminobenzoic acid was stable under anaerobic conditions.	[53]
6	Diatrizoate Iomeprol	Subsurface flow constructed wetland pond with floating plants	Test tubes made of quartz glass (32 mL) were positioned horizontally on a rack in the pond	3.4 ± 1.0 6.3 ± 5.8 μg L^−1^	Initial 8.8	17–19	18 days 22 days 10d	Both removal efficiency range from 43–66% in summer decreased in winter. Photodegradation was found to be an important process for the removal of diatrizoate and iopromide.	[54]
7	Iopromide Diatrizoate Iohexol Iomeprol Iopamidol Ioxitalamic	Organic sludge	Pilot-Scale MBR fed with an average influent of 1.2 m^3^ of wastewater per day Submerged ultrafiltration flat sheet membrane plates	2600 μg L^−1^	7.8	29	1 year	The highest elimination efficiency within this pharmaceutical class was detected for iopromide (31%), while the elimination was negligible for the other ICM.	[55]

**Table 3 molecules-28-00341-t003:** Examples on the treatment of ICM by UV based AOPs.

Contaminant	Type of AOPS	Radical	Concentration	pH	Time	UV Intensity μW cm^−2^	Rate Constant First-Order S^−1^ Second-Order M^−1^ S^−1^	Main Results	Refs.
Diatrizoate	H_2_O_2_	•OH	25 mg L^−1^ DTR, 5 mg L^−1^ H_2_O_2_	-	24 h	-	-	22.40% degradation of initial DTR	[87]
	UV		25 mg L^−1^ DTR	6.5	60 min	-	1.02 × 10^5^ s^−1^	80.60% degradation of initial DTR	
	UV/ H_2_O_2_	•OH	25 mg L^−1^ DTR, 0.5 mM H_2_O_2_	6.5		-	1.48 × 10^5^ s^−1^	91.3% degradation of initial DTR	
	UV/ K_2_S_2_O_8_	SO_4_^•−^	25 mg L^−1^ DTR, 10 mM K_2_S_2_O_8_	6.5		-	1.27 × 10^5^ s^−1^	78.3% degradation of initial DTR	
Diatrizoate	UV		0.5 μM DTR	6.0–7.5	36 min	-	-	40% degradation of initial DTR	[75]
	UV/ H_2_O_2_	•OH	0.5 μM DTR, 1.0 mM H_2_O_2_	7.4	-	-	-	50% degradation of initial DTR	
	UV/ K_2_S_2_O_8_	SO_4_^•−^	0.5 μM DTR, 1.0 mM K_2_S_2_O_8_	7.4	-	-	3.7 × 10^9^ M^−1^ s^−1^	100% degradation of initial DTR	
Diatrizoate	UV/ K_2_S_2_O_8_	SO_4_^•−^	30 mg L^−1^ DTR, 12 mM K_2_S_2_O_8_	6.5	1 h	-	1.9 × 10^9^ M^−1^ s^−1^	42% oxidation pathways: deiodination-hydroxylation, decarboxylation- hydroxylation, side chain cleavage	[59]
Iopromide	UV/ K_2_S_2_O_8_	SO_4_^•−^	0.126 mM Iopromide, 2 mM K_2_S_2_O_8_	3-8	30 min	9 x 10^−6^	1-2 × 10^4^ M^−1^ s^−1^	Complete removal and almost complete mineralization in 80 min	[88]
Iohexol	UV/ H_2_O_2_	•OH	10 μM iohexol 100, 200 500 μM Na_2_S_2_O_8_	5-7		225	5.73 × 10^8^ s^−1^	Anions inhibitory effects: Cl^−^ > HCO_3_^−^ >> SO_4_^•−^ UV/ SO_4_^•−^ can effectively mineralize iohexol to CO_2_ but promoted the generation of toxic iodoform (CHI_3_)	[74]
	UV/ K_2_S_2_O_8_	SO_4_^•−^		7-9			3.91 × 10^10^ M^−1^ s^−1^		
Diatrizoate	UV/Chlorine	Reactive chlorine species	10 μM DTR, 100 μM HOCl	7	180 s	245	3.05 x 10^−2^ s^−1^	95% degradation of initial DTR, the pseudo-first-order rate constant during UV/chlorination was 9.3 times higher than that during UV photolysis. Byproduct dichloroacetonitrile (C_2_HCl_2_N), which is carcinogenic	[69]
Diatrizoate	UV/Chlorine	Reactive chlorine species	10 μM DTR, 25 μM Chlorine 25 μM chloramine 4 mM phosphate buffer	7	5 min	-	8.45 × 10^−3^ s^−1^	UV/chlorine degraded DTR more efficiently than UV/chloramine process. Degradation inhibited in natural waters. Formation of chloroform, dichloroacetonitrile, and iodoform. Both processes were restricted by water matrix	[78]
	UV/Chloramine	•OH				-	4.19 × 10^−3^ s^−1^		
Iohexol	UV/Chlorine	•OH	10 μM iohexol 10 mM phosphate buffer 100 μM HOCl	7	5 min	3020	3.8 × 10^9^ M^−1^ s^−1^	Formation of iodinated trihalomethanes (I-THMs) during post-chlorination	[70]
Iopamidol	UV-LED/chlorine	Reactive chlorine species •OH	[iopamidol]0 = 10 μM	7.0	-	120.8 76.3 99.6	-	The dual-wavelength system significantly accelerated iopamidol degradation during both direct photolysis and the UV-LED/chlorine process (*p* < 0.05) due to the promotion of photon excitation process	[71]

**Table 4 molecules-28-00341-t004:** O_2_ evolution potential of various anode materials [102,103].

Anode Material	Potential for O_2_ Evolution (V/SCE)
RuO_2_	1.4–1.7
IrO_2_	1.5–1.8
Pt	1.6–1.9
Graphite	1.7
PbO_2_	1.8–2.0
SnO_2_	1.9–2.2
BDD	2.2–2.6

**Table 5 molecules-28-00341-t005:** Identified by-products of diatrizoate in recent literature.

Pathways	Methods	by-Products							Ref.
Deiodination	Catalytic-Hydrodeiodination Gamma-Radiolysis	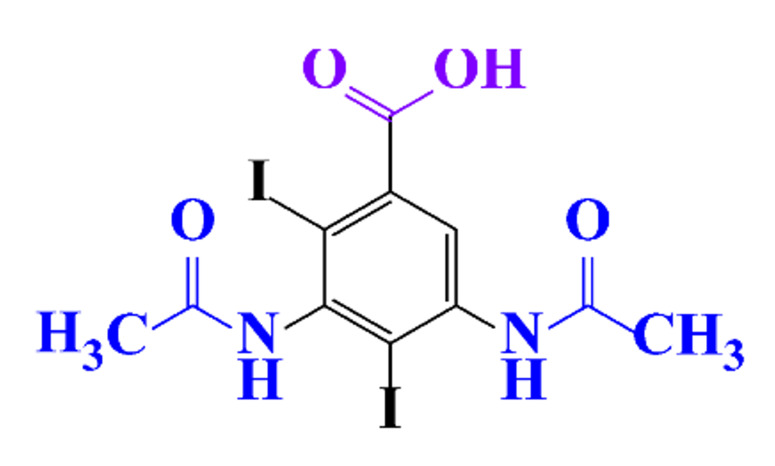	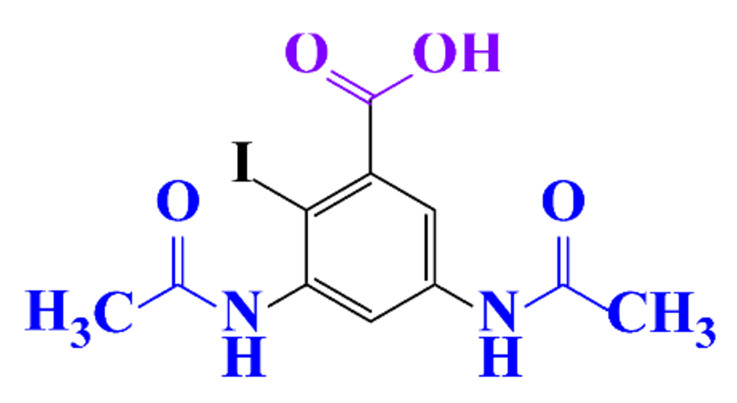	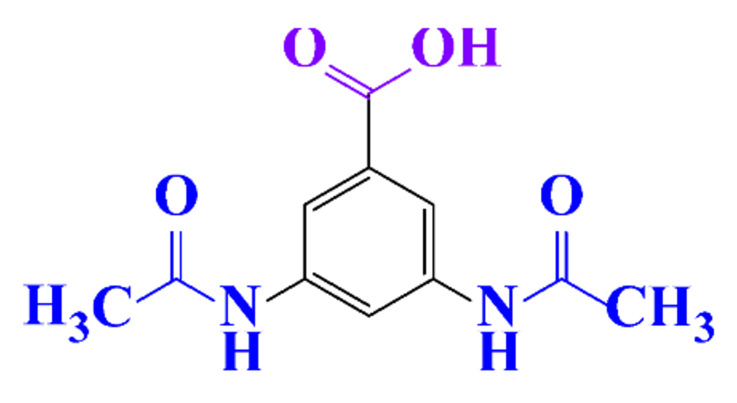	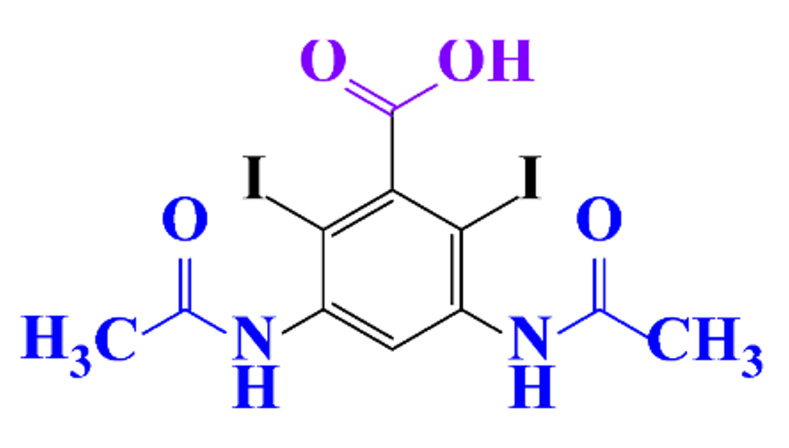	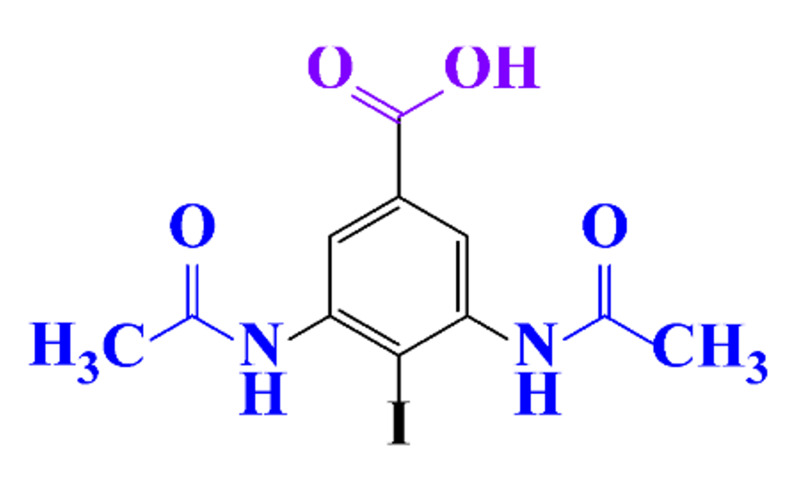	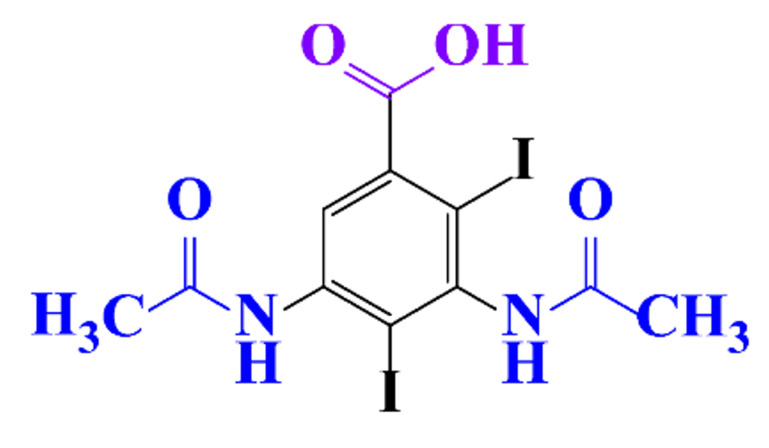		[151] [152] [153]
Deacetylation	UV-H_2_O_2_ Gamma-Radiolysis	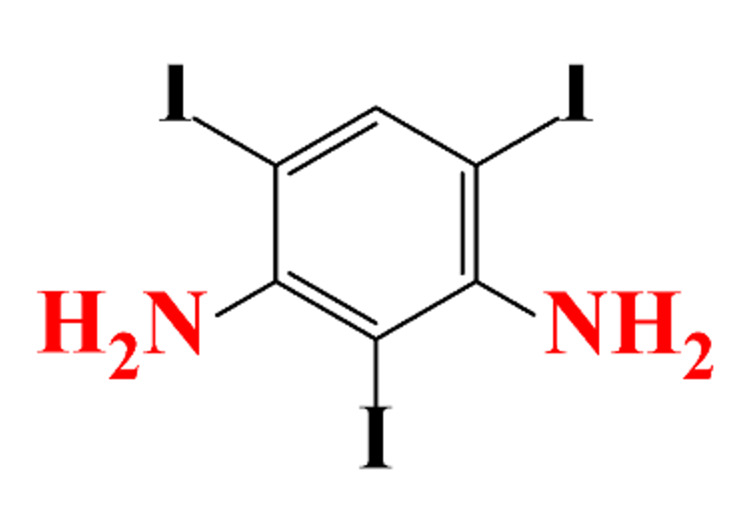	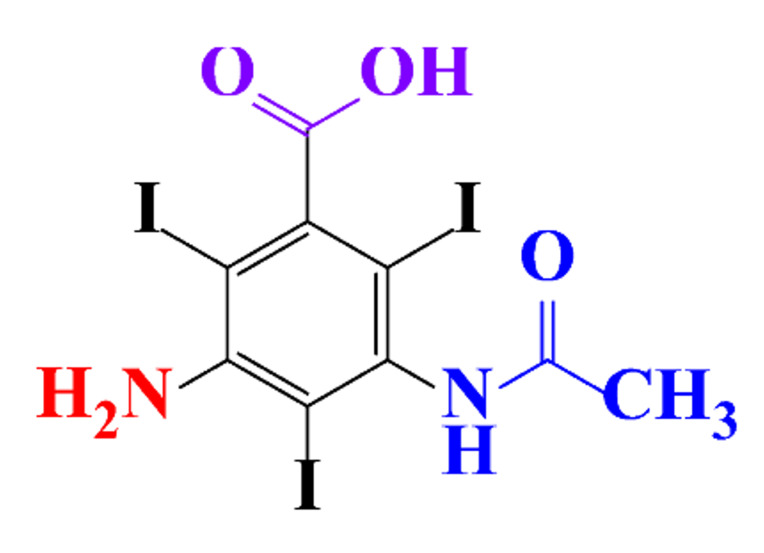	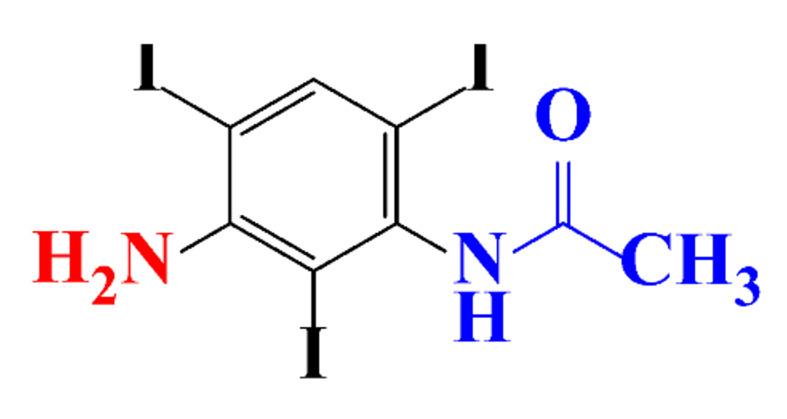	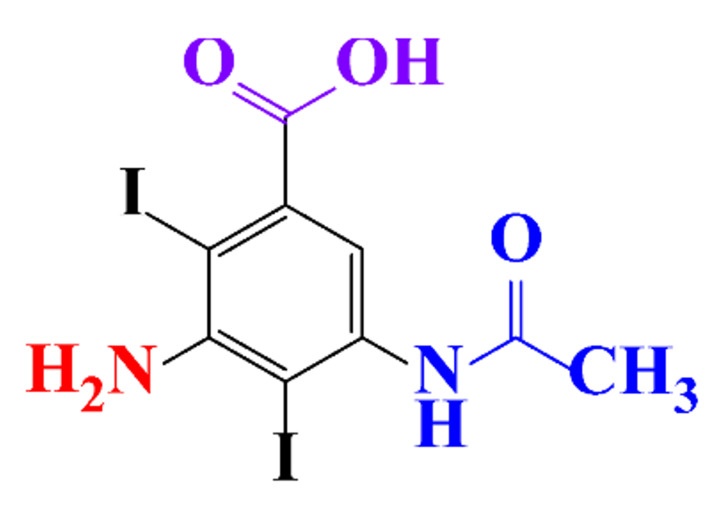	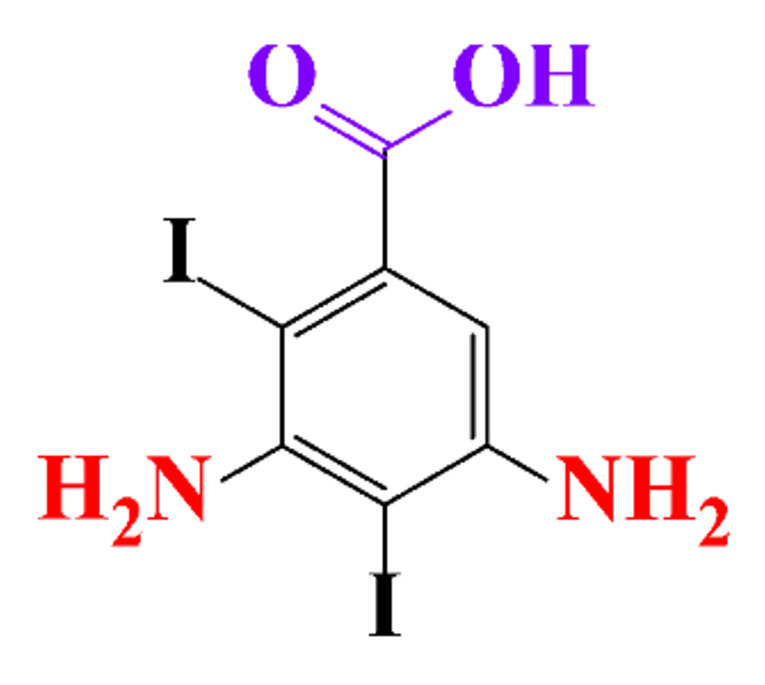			[93] [151]
Hydroxylation	UV-SO^•^_4_^−^ Electro-oxidation UV-H_2_O_2_ Electro-Fenton Gamma-Radiolysis	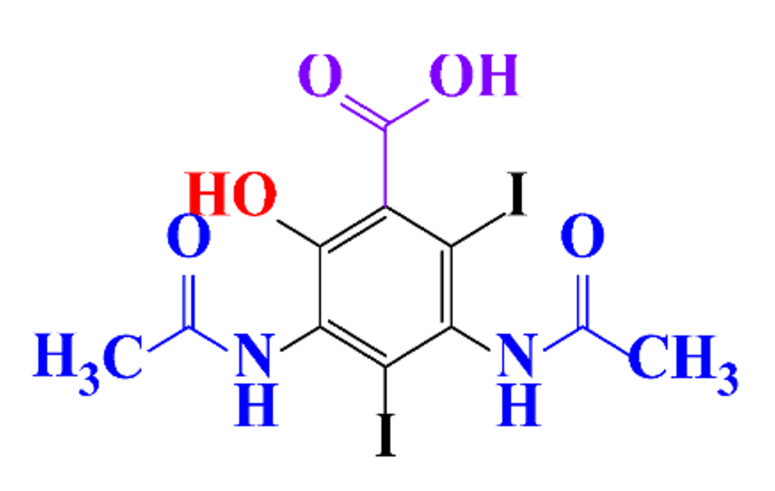	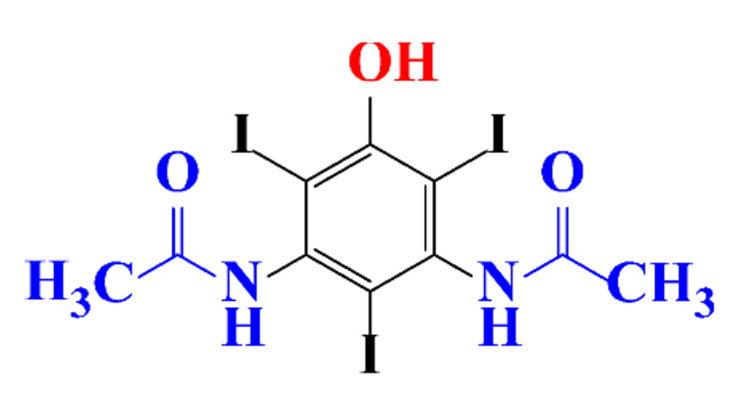	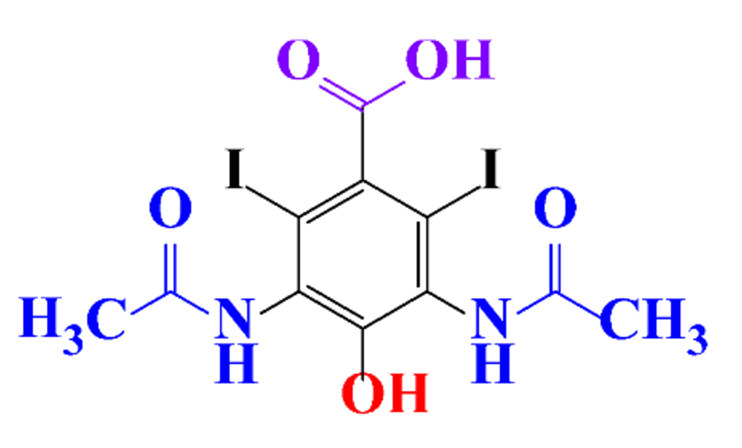	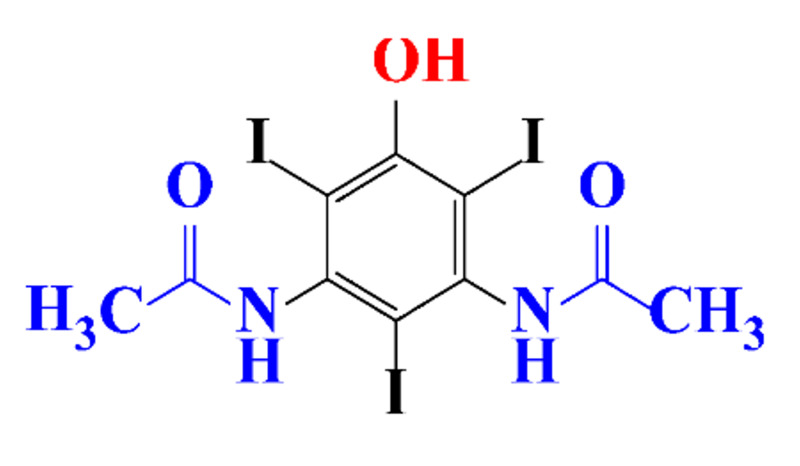	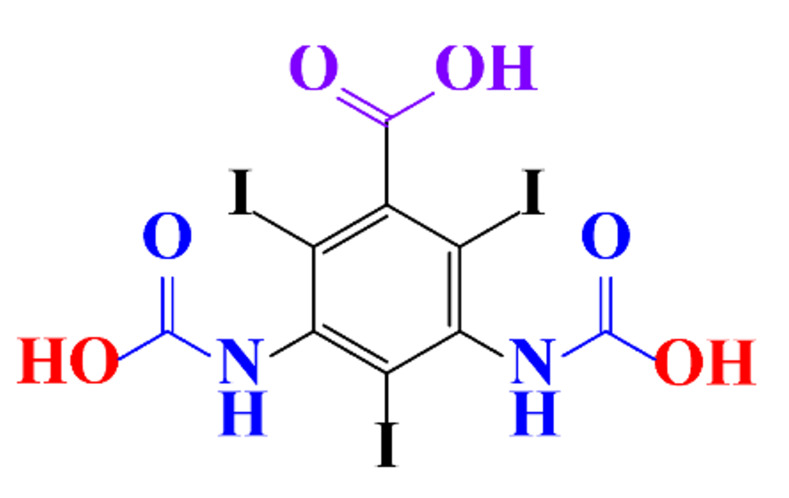	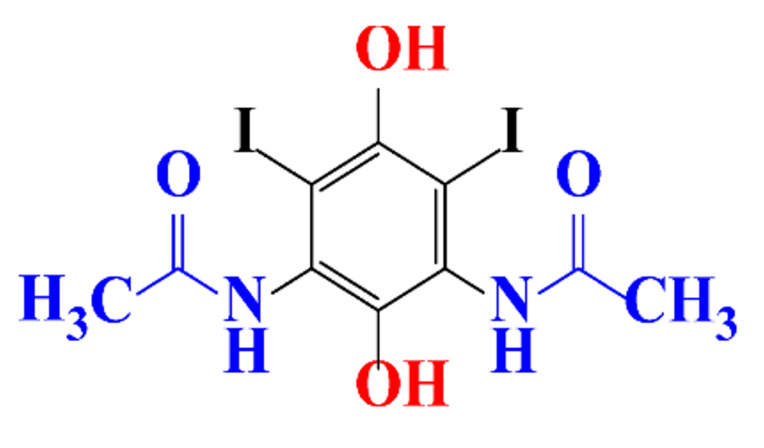	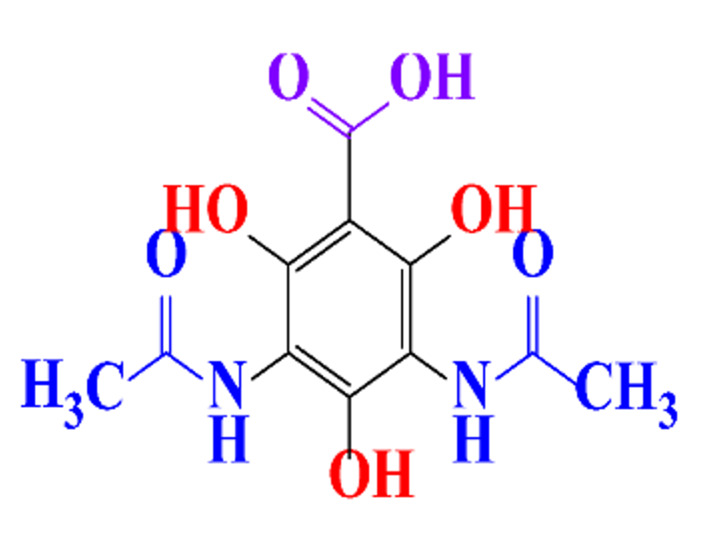	[82] [104] [153] [93] [97] [151]
		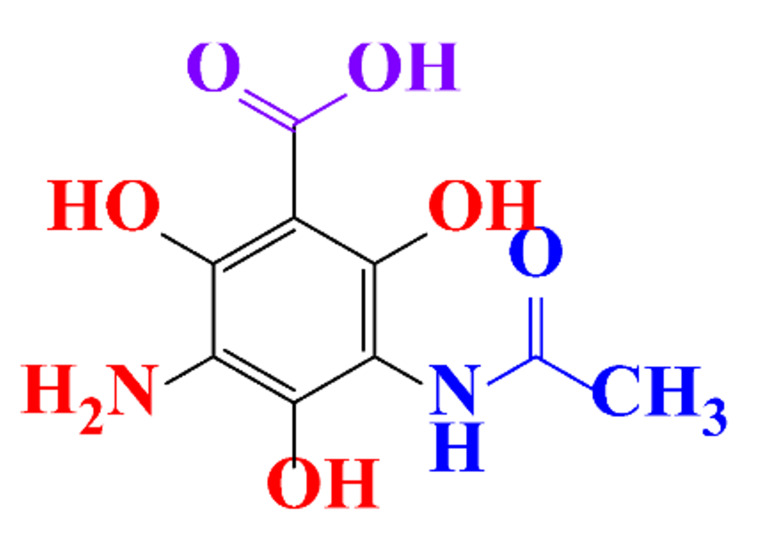	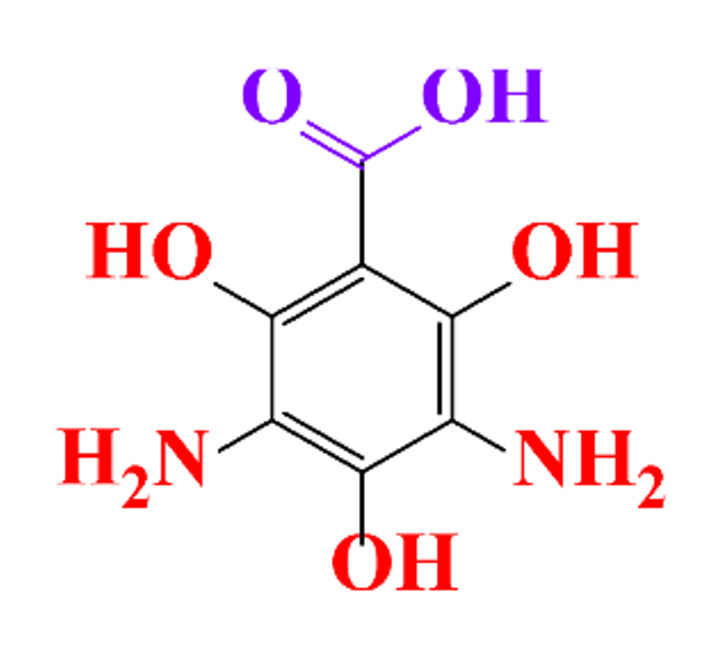	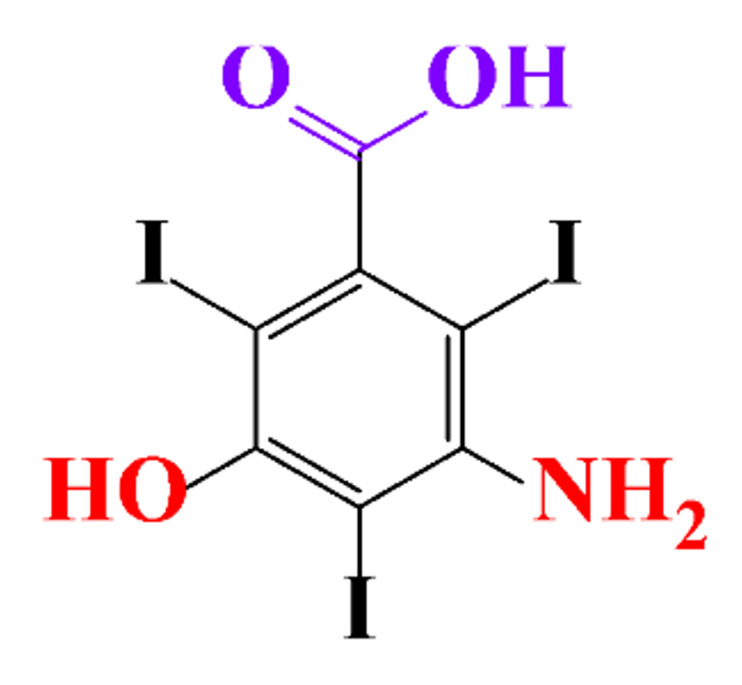	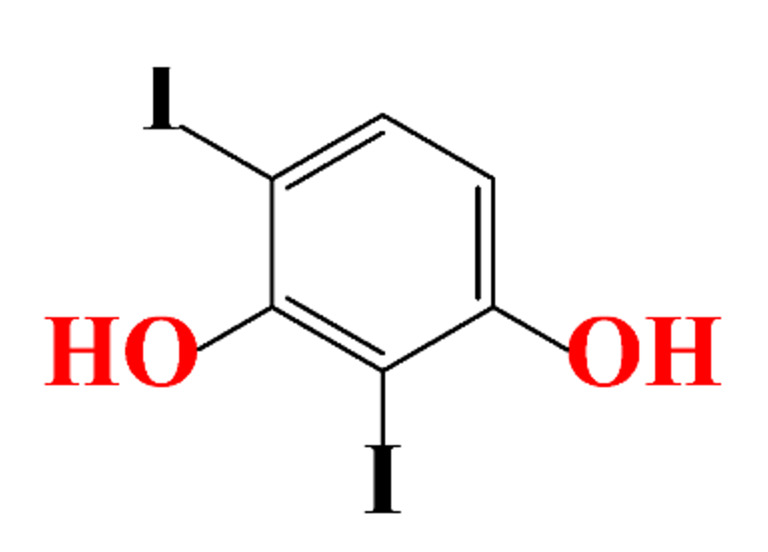	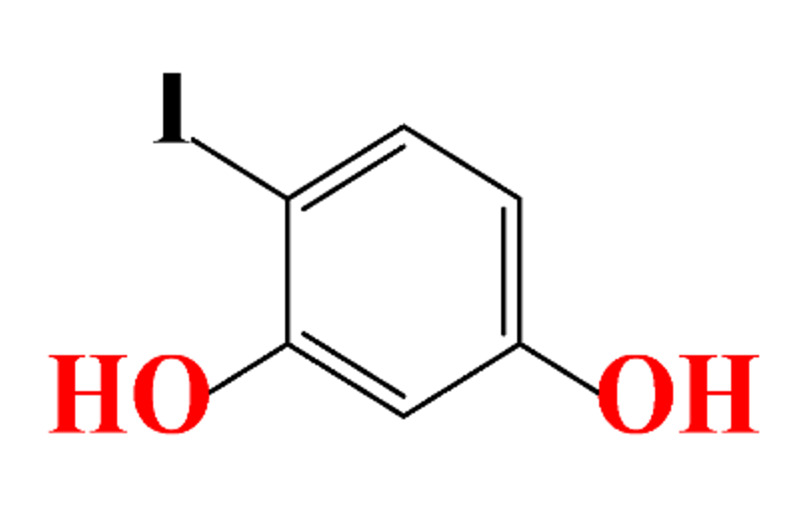	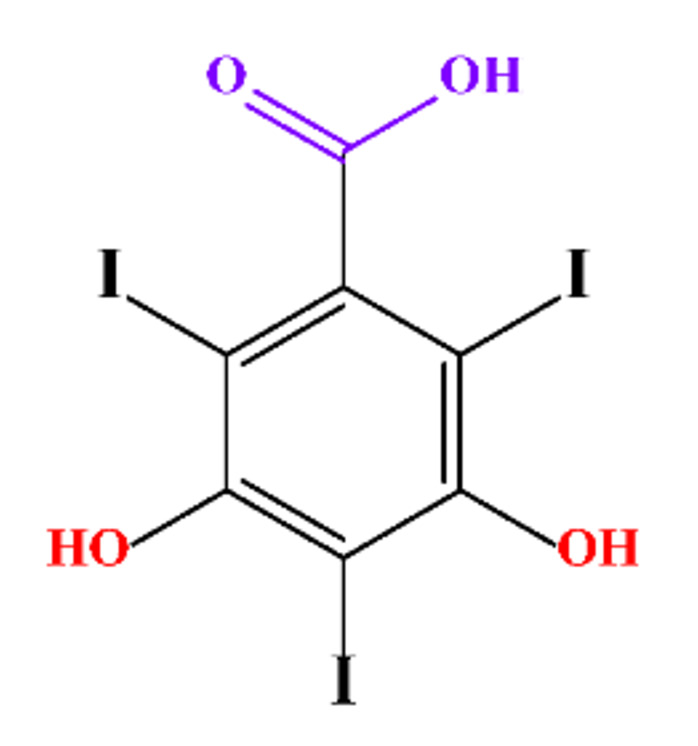		
Cyclization	UV-SO^•^_4_^−^ Electro-oxidation	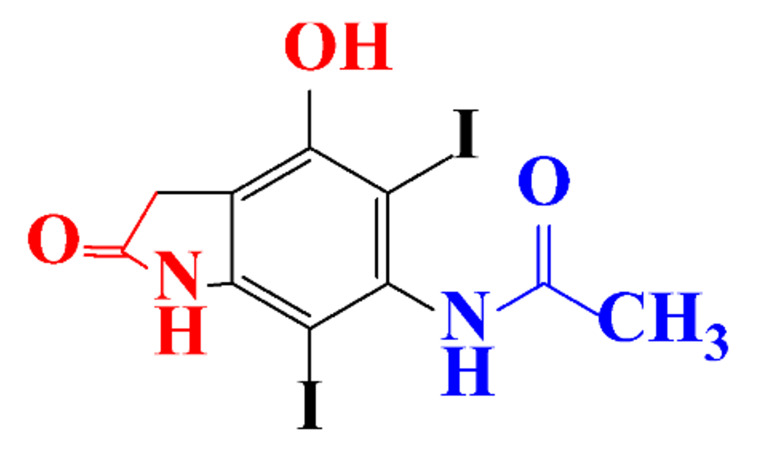	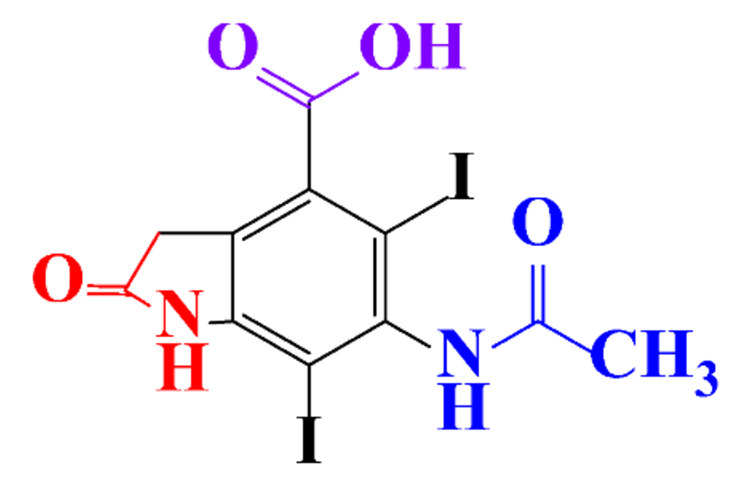	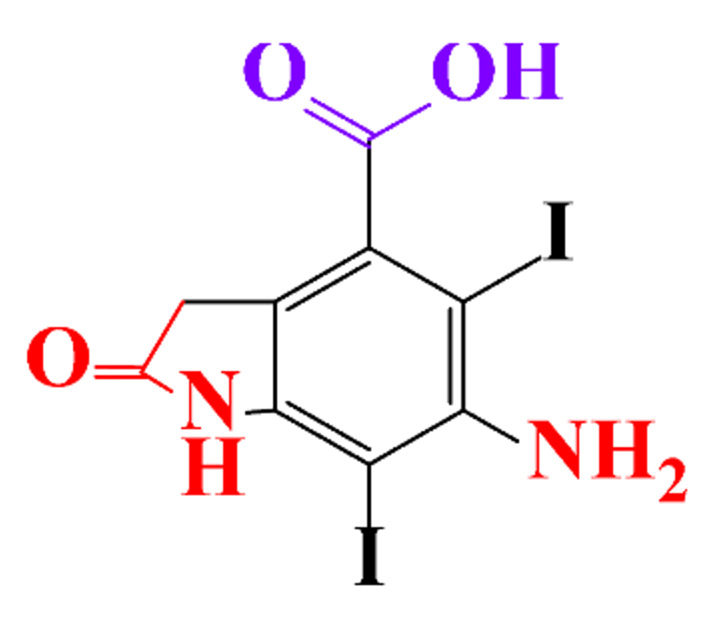	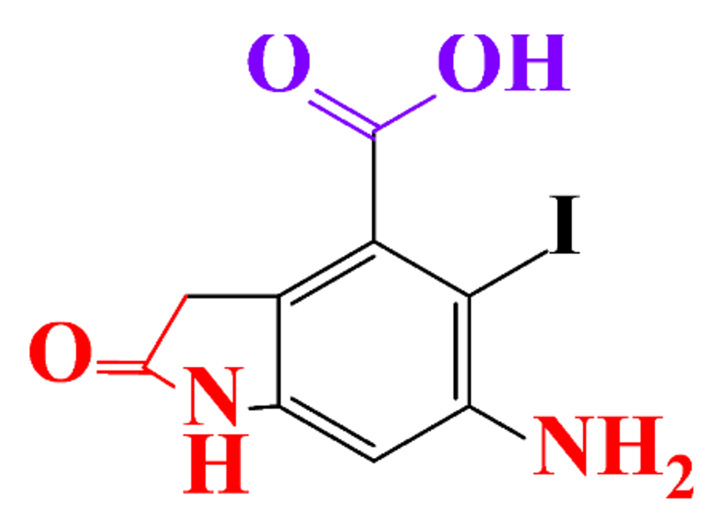				[82] [104]
Electroreduction	Lead electrode 20 mA cm^−2^	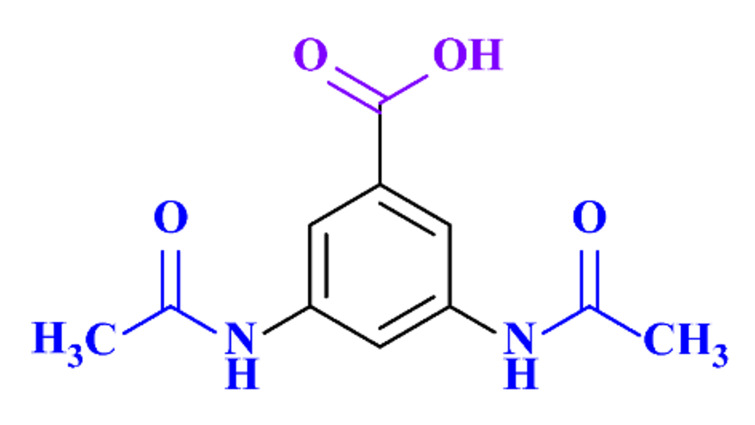	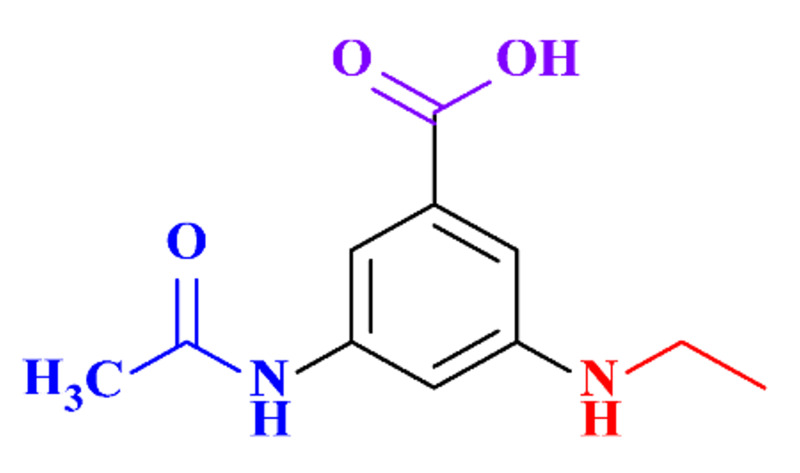	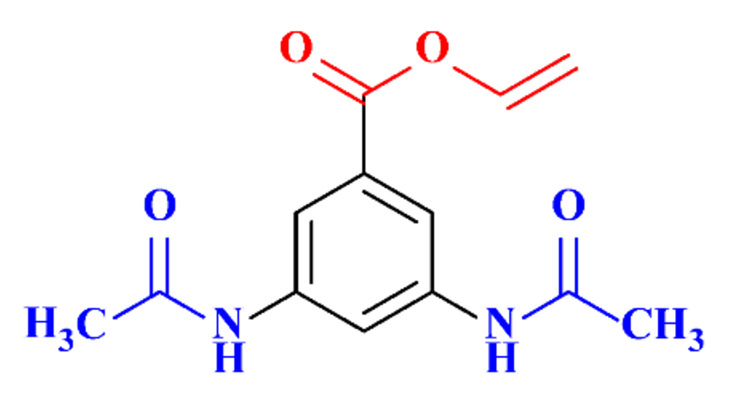					[141]
Cyclic ketones	UV-SO^•^_4_^−^ Gamma-Radiolysis	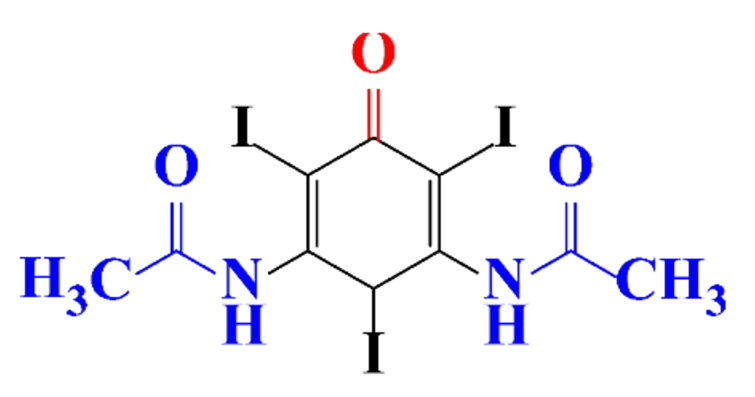	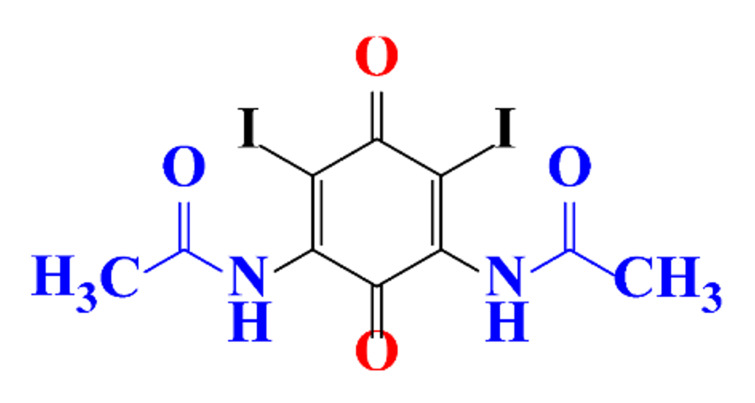	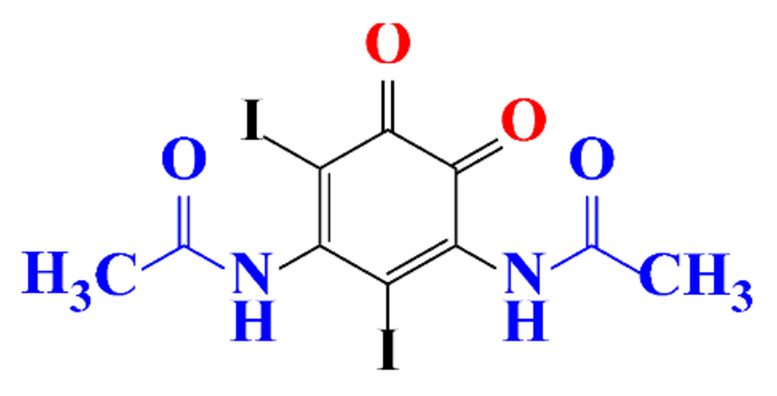					[59] [151]

**Table 6 molecules-28-00341-t006:** Identified by-products of iohexol in recent literature.

Pathways	Methods	by-Products							Ref.
Deiodination	Cu cathode-SO^•^_4_^−^ UV-AOPs Co^2+^-SO^•^_4_^−^	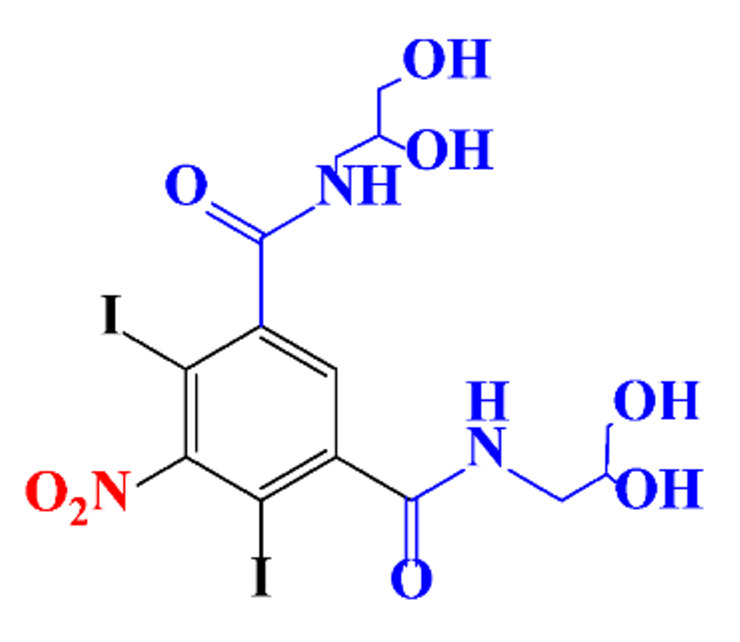	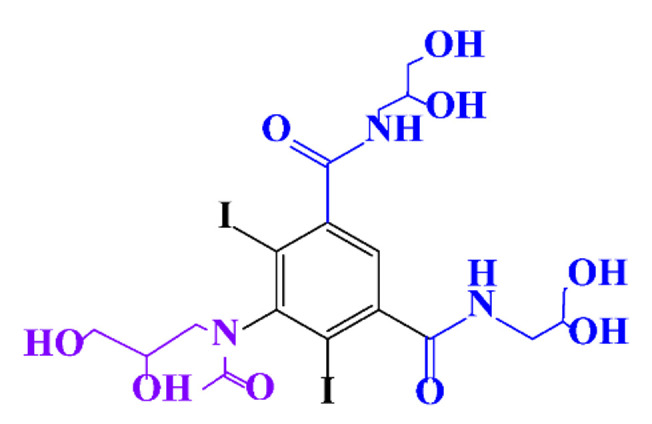	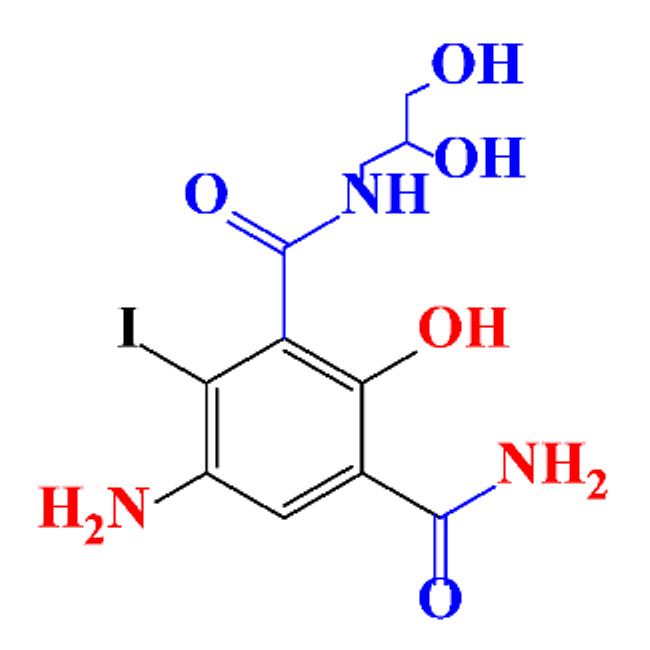	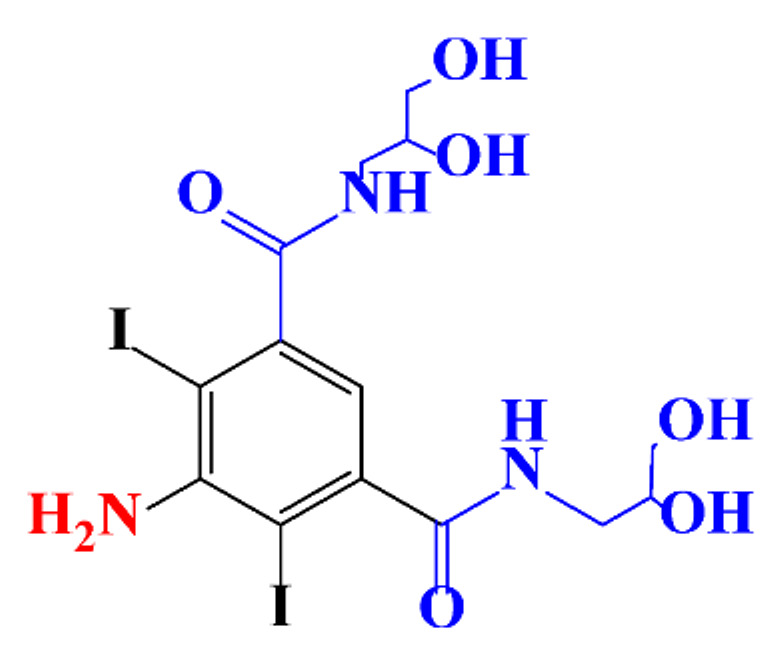	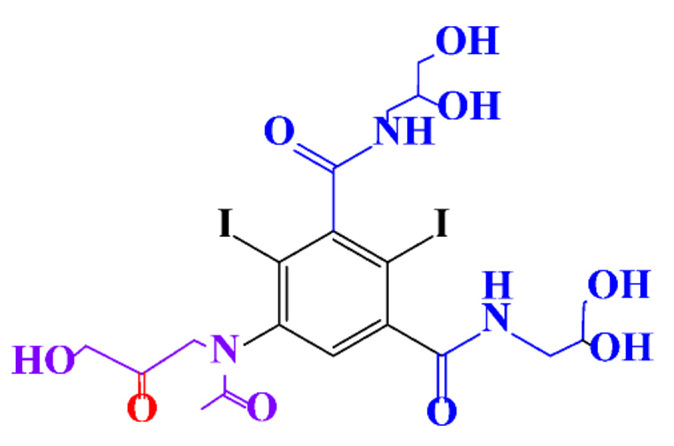	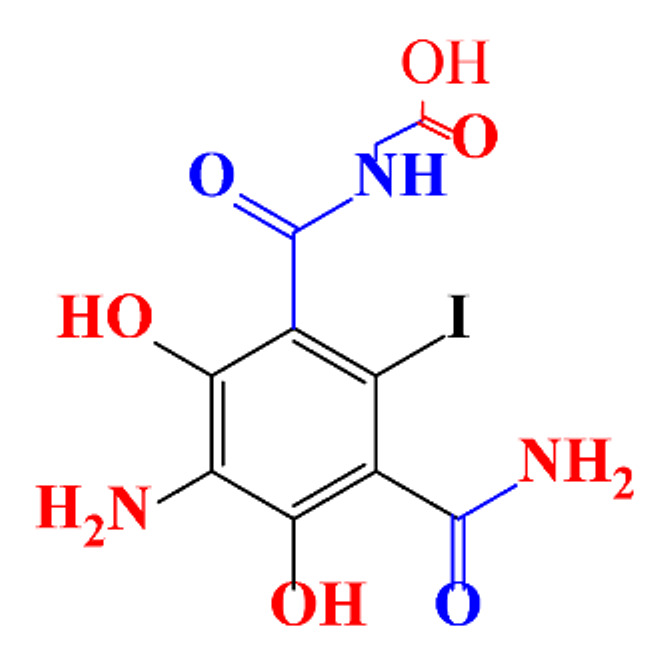		[154] [76] [155]
Amide-Hydrolysis	Cu cathode-SO^•^_4_^−^ UV-AOPs Co^2+^-SO^•^_4_^−^	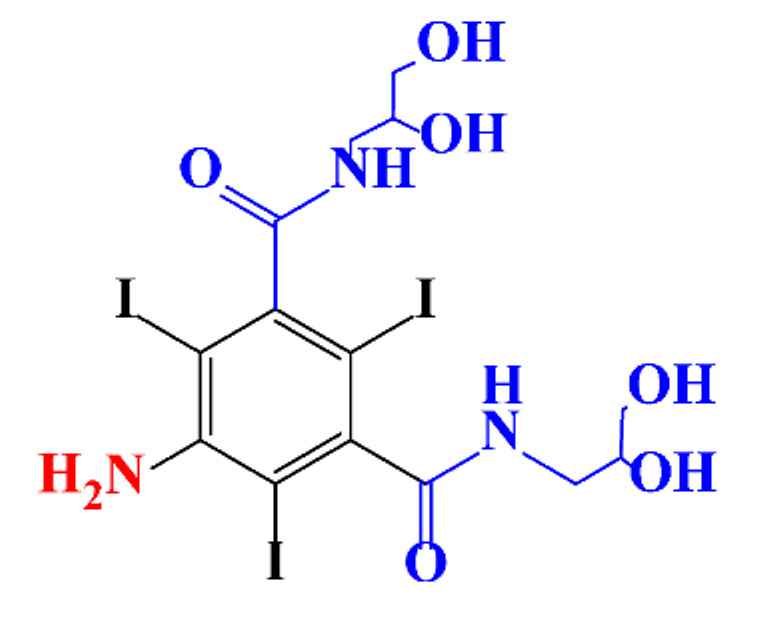	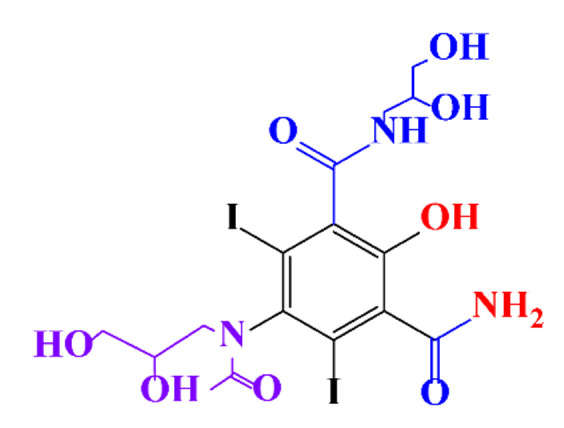	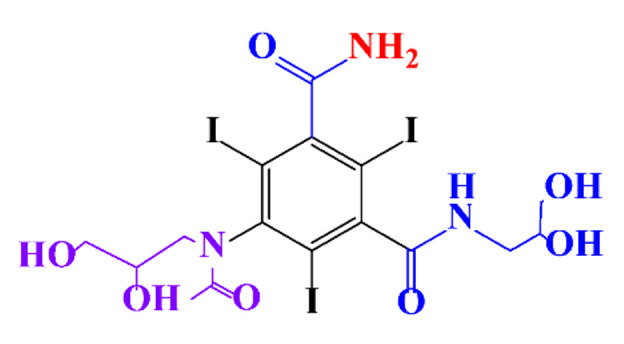	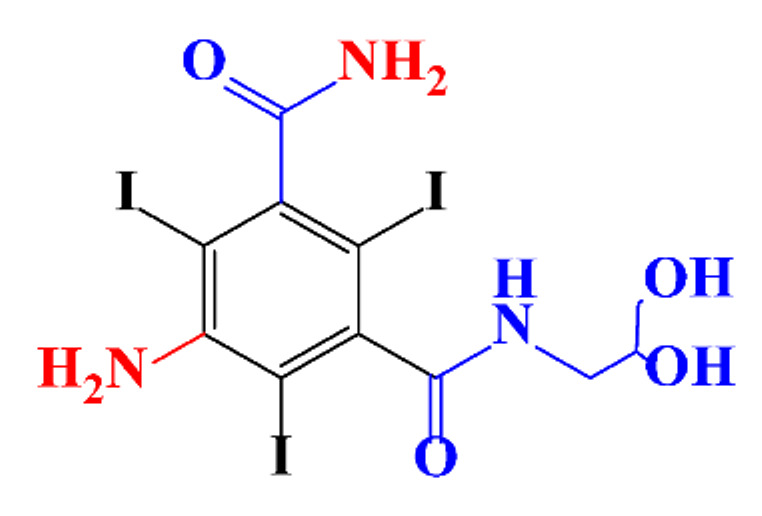	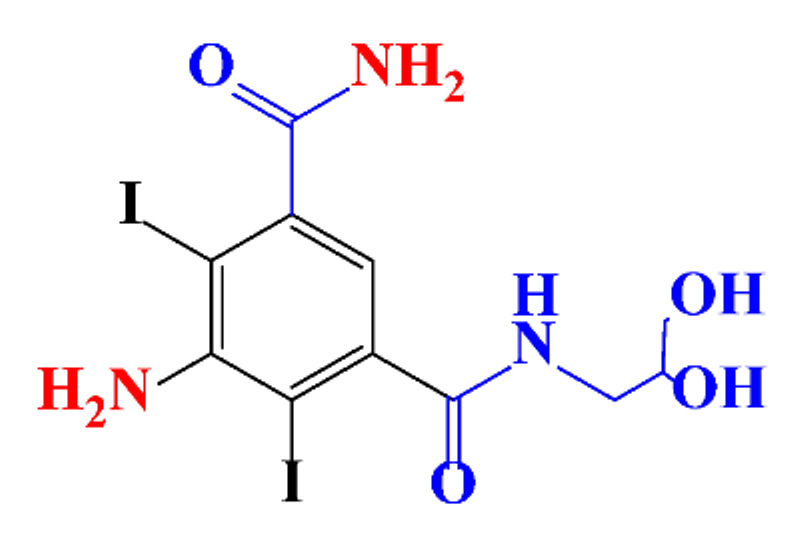	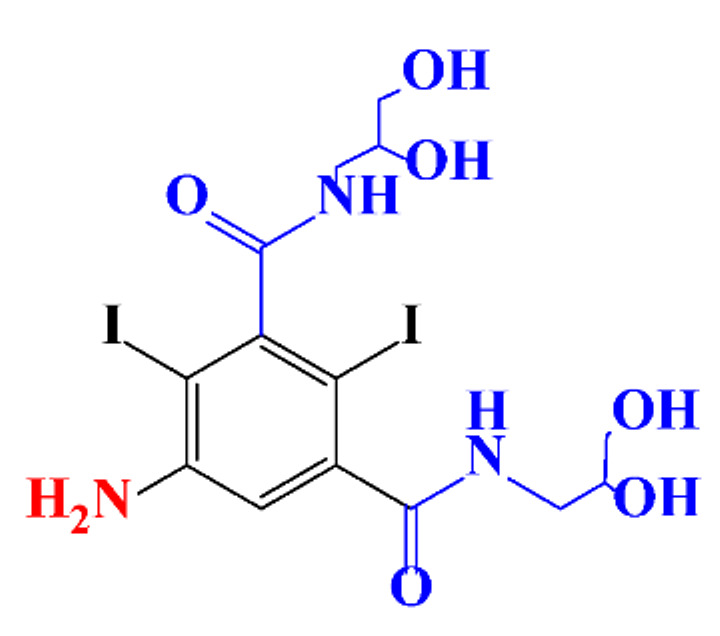	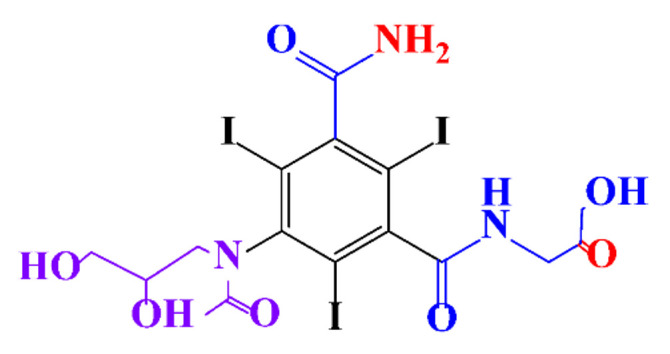	[154] [76] [79]
Hydrogen- Abstraction	Cu cathode-SO^•^_4_^−^ UV-AOPs Co^2+^-SO^•^_4_^−^	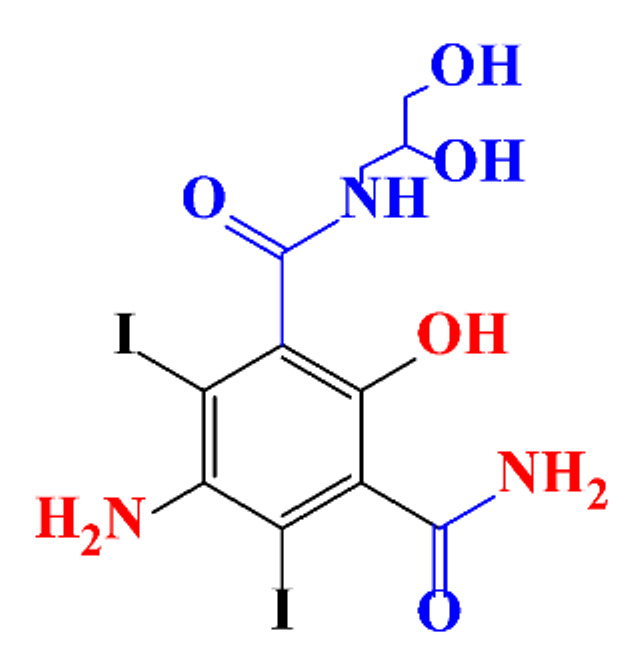	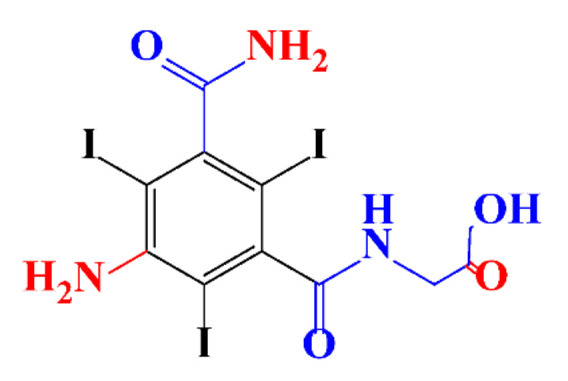	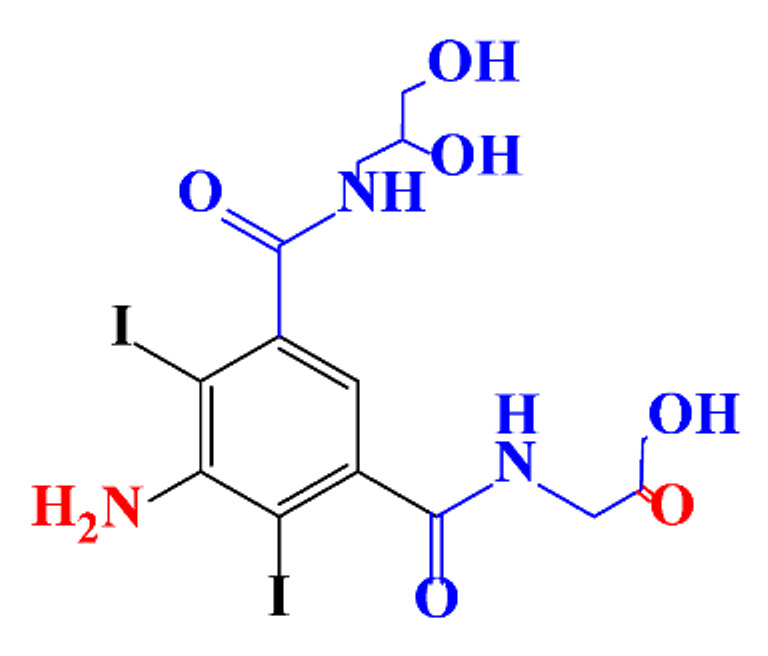	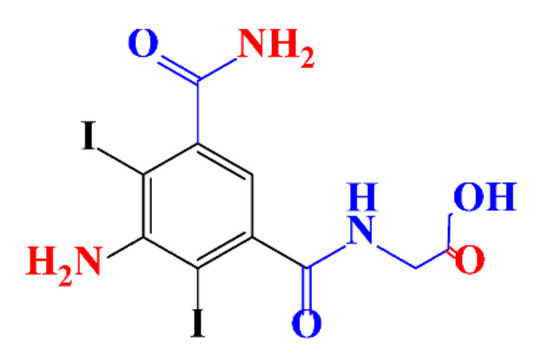	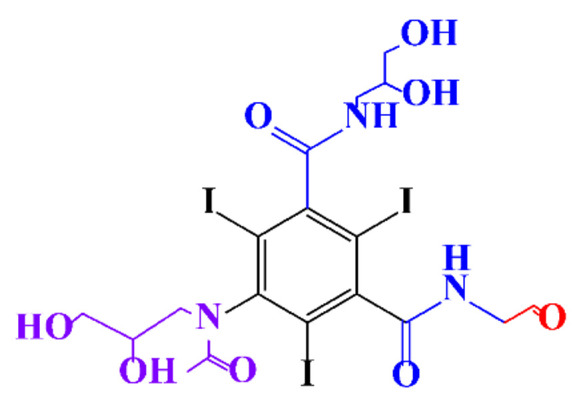	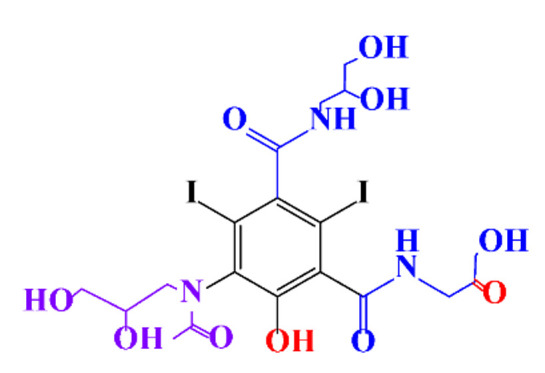	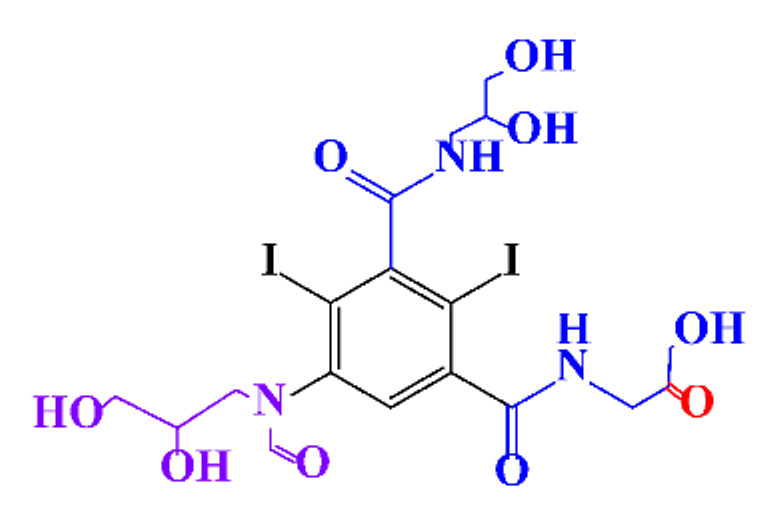	[154] [76] [79] [155]
		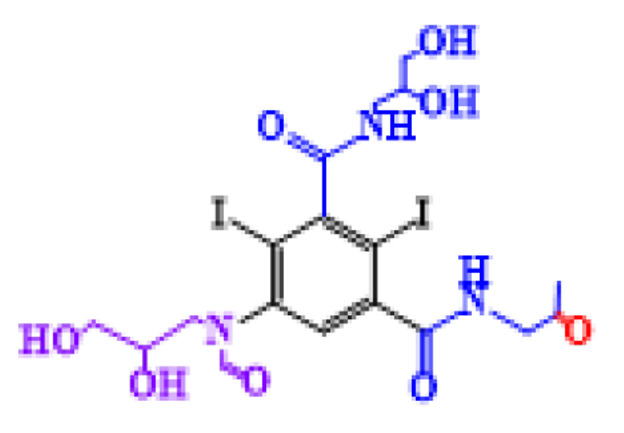	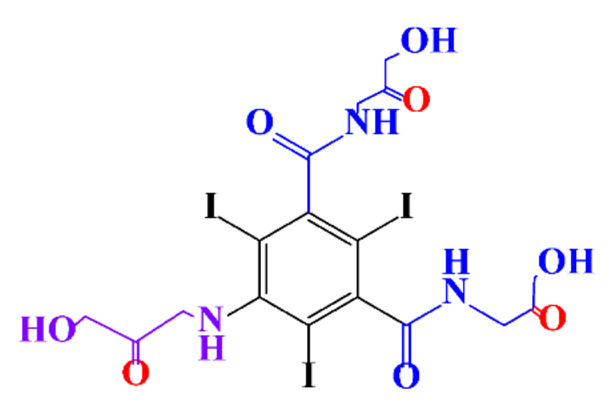	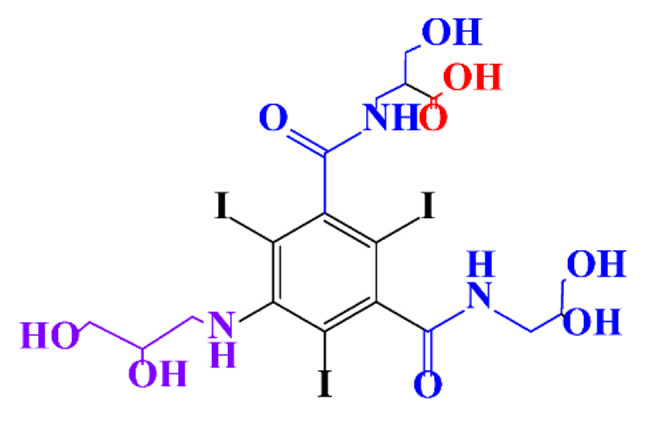	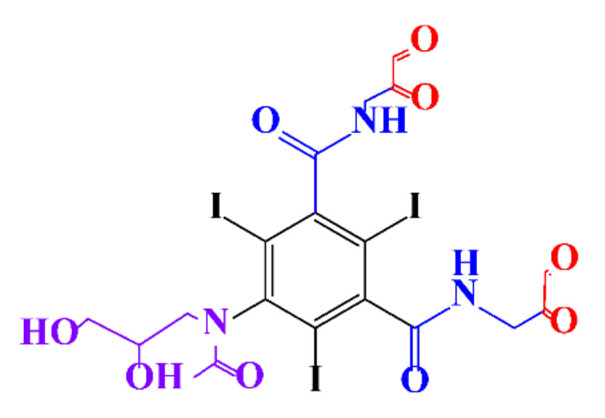	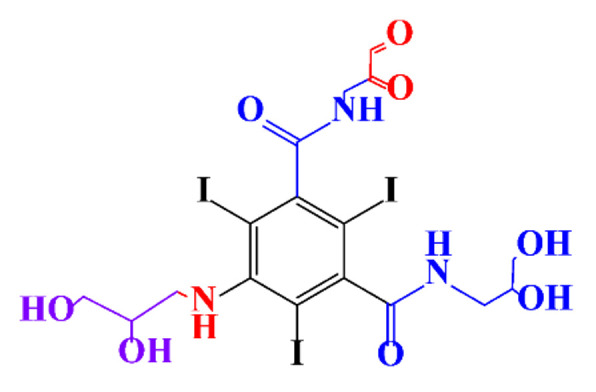			
Amino- Oxidation	Cu cathode-SO^•^_4_^−^ Co^2+^-SO^•^_4_^−^	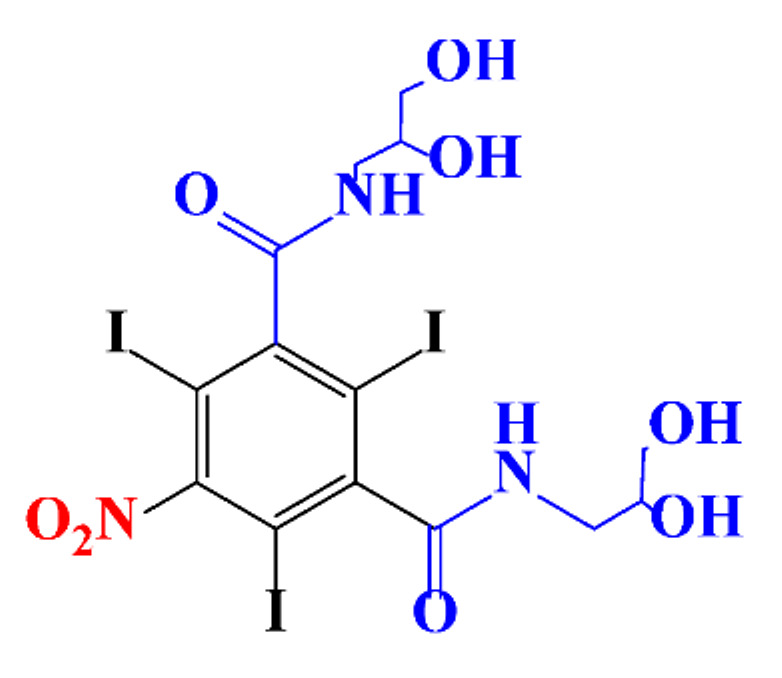	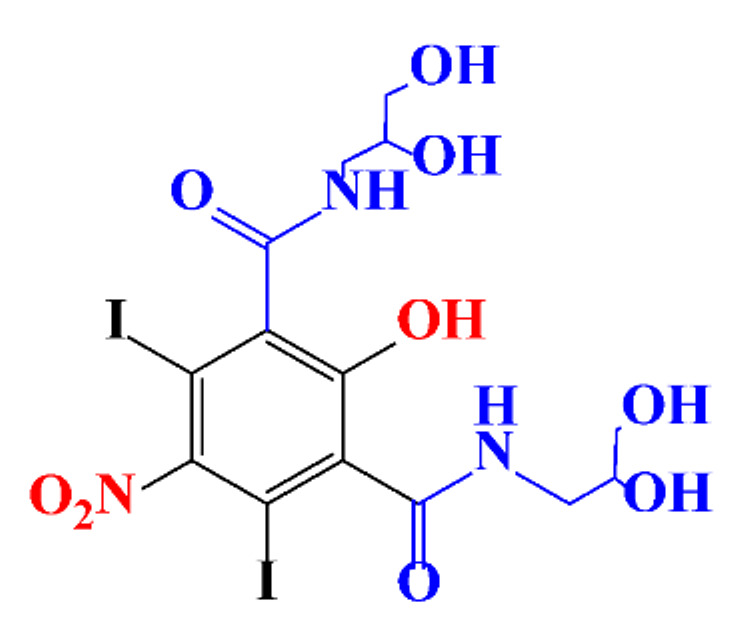	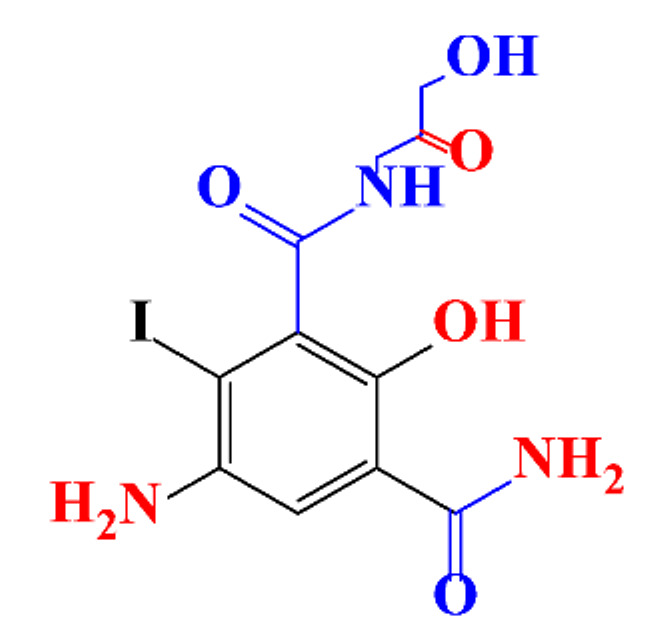	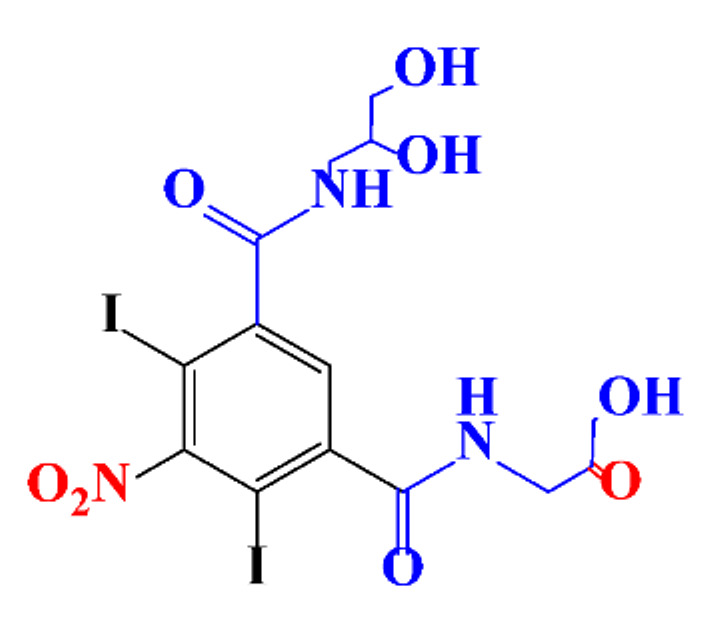	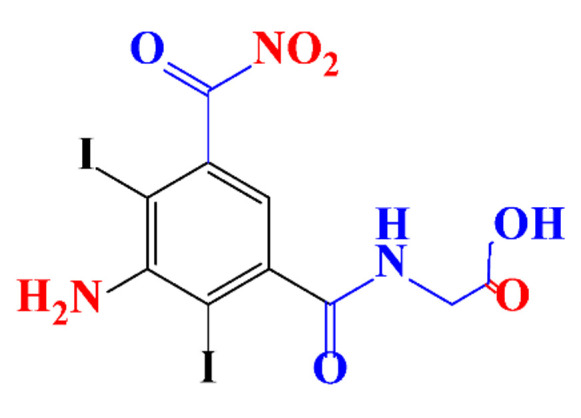	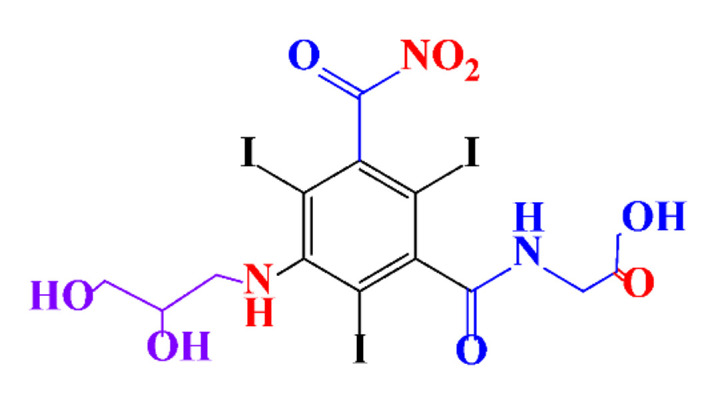		[154] [79]
•OH- Addition	Cu cathode-SO^•^_4_^−^ UV-AOPs	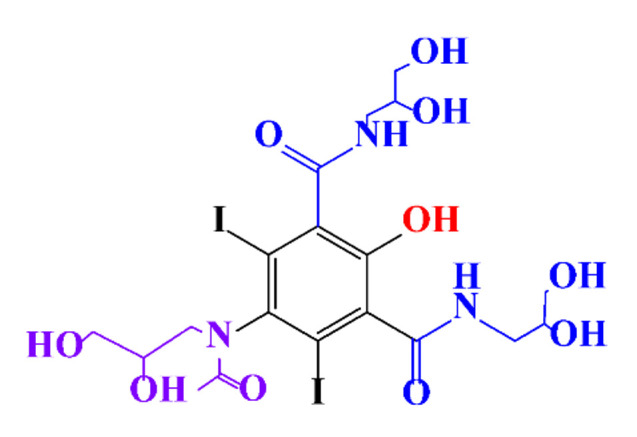	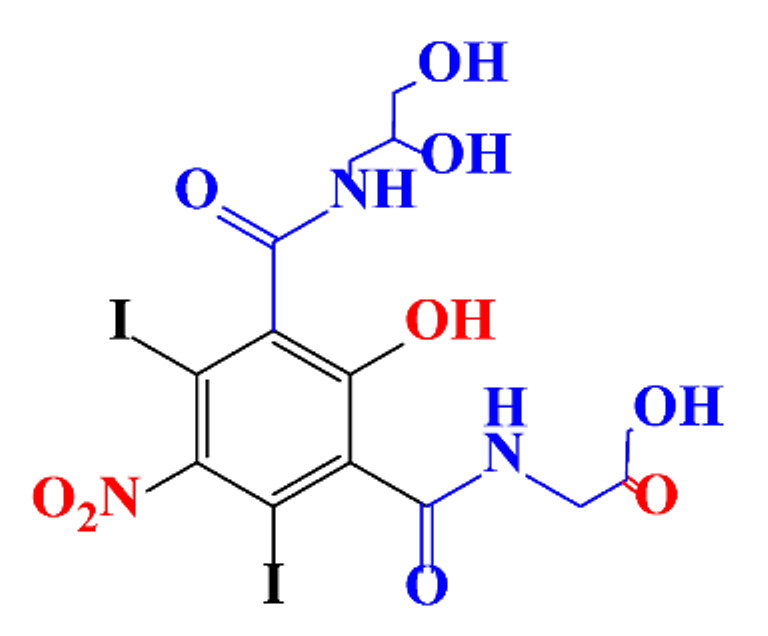	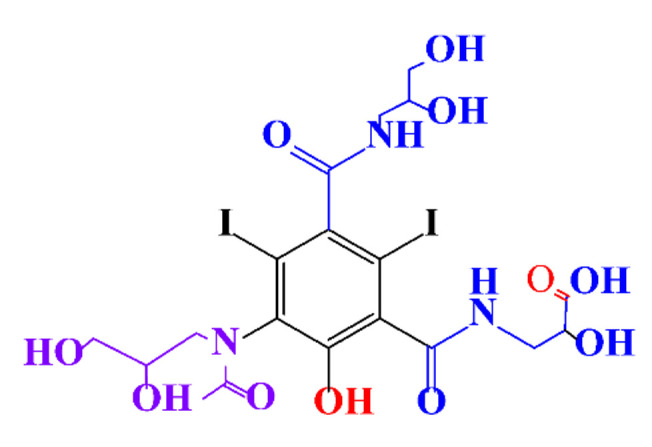					[154] [76]

**Table 7 molecules-28-00341-t007:** Identified by-products of iomeprol in recent literature.

Pathways	Methods	by-Products						Ref.
Deiodination	Electro-reduction	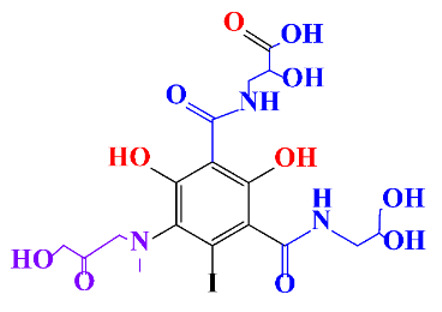	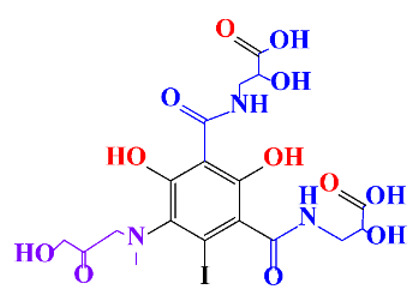					[137]
Carboxylation	Co^2+^-SO^•^_4_^−^	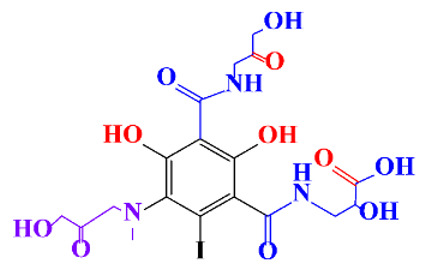						[79]
Carbamoyl	Co^2+^-SO^•^_4_^−^	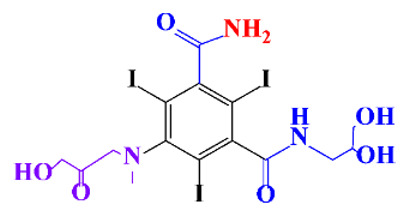	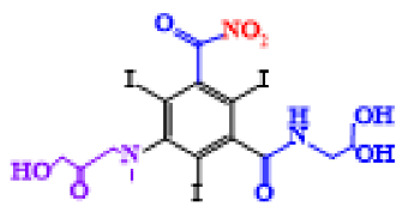	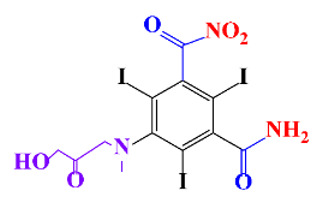				[79]
Dehydration	Co^2+^-SO^•^_4_^−^	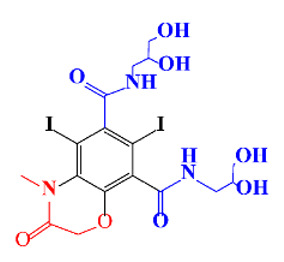						[79]
Hydrogen- Abstraction	Co^2+^-SO^•^_4_^−^	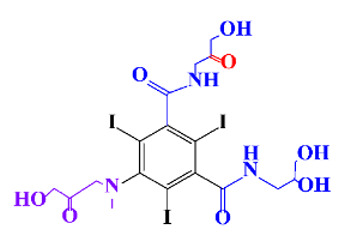						[79]
•OH- Addition	Co^2+^-SO^•^_4_^−^	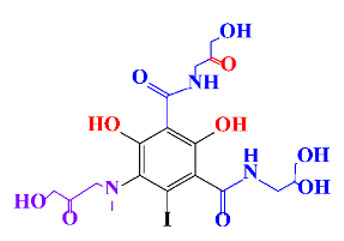	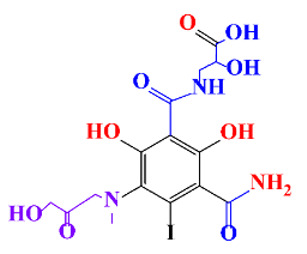					[79]

**Table 8 molecules-28-00341-t008:** Identified by-products of iopamidol in recent literature.

Pathways	Methods	By-Products						Ref.
Deiodination	UV-irradiation UV-•Cl Metal oxides-SO^•^_4_^−^ Iron-SO^•^_4_^−^ Ferrate-oxidation Catalytic-hydrodeiodination	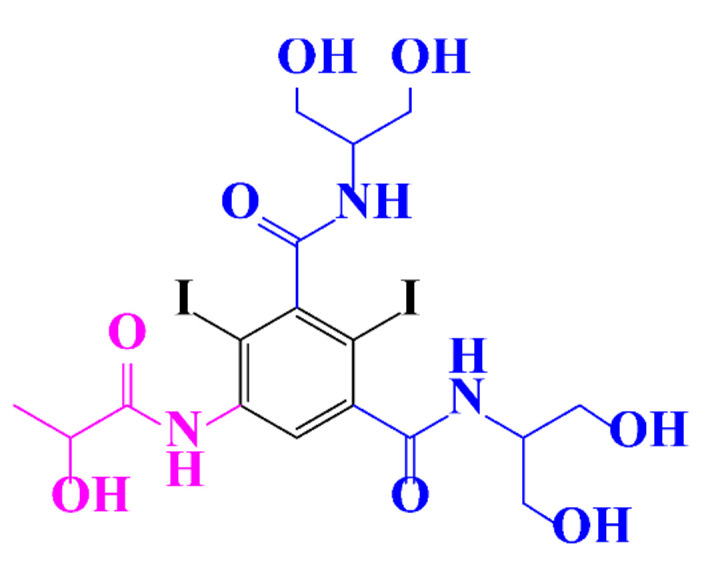	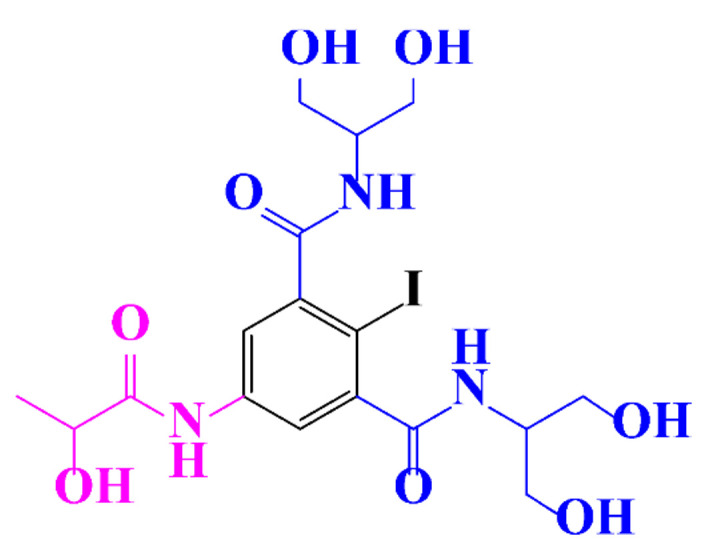	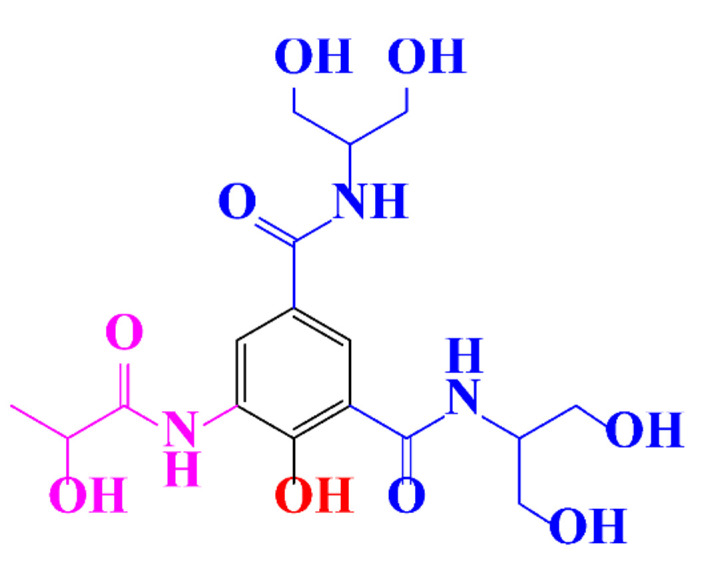	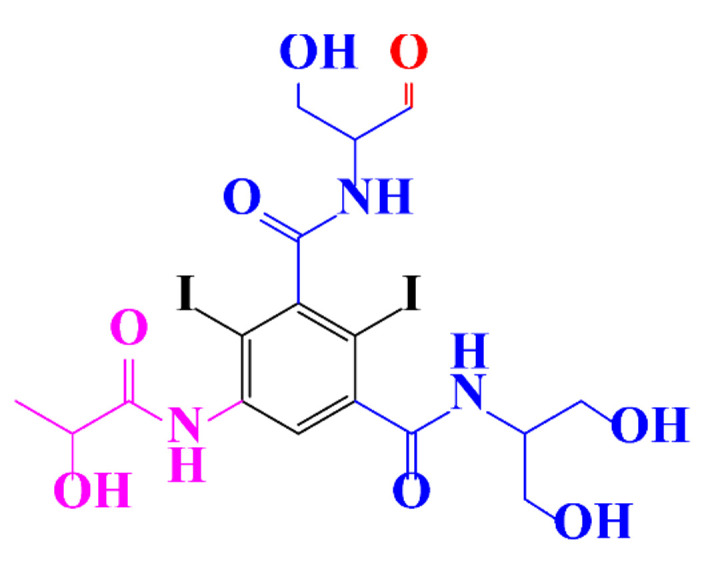	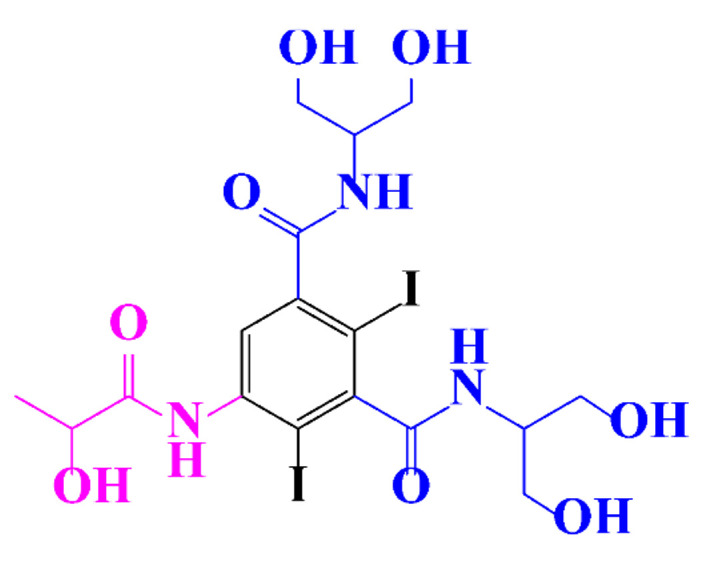	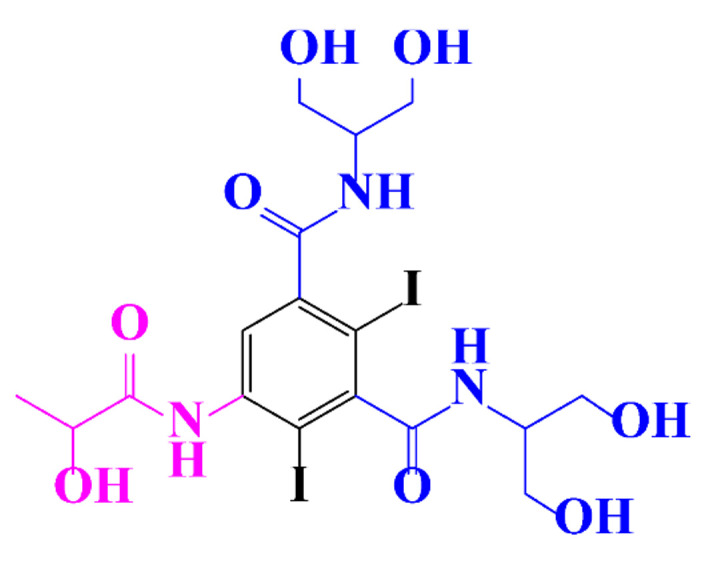	[85] [71] [120] [116] [156] [157]
		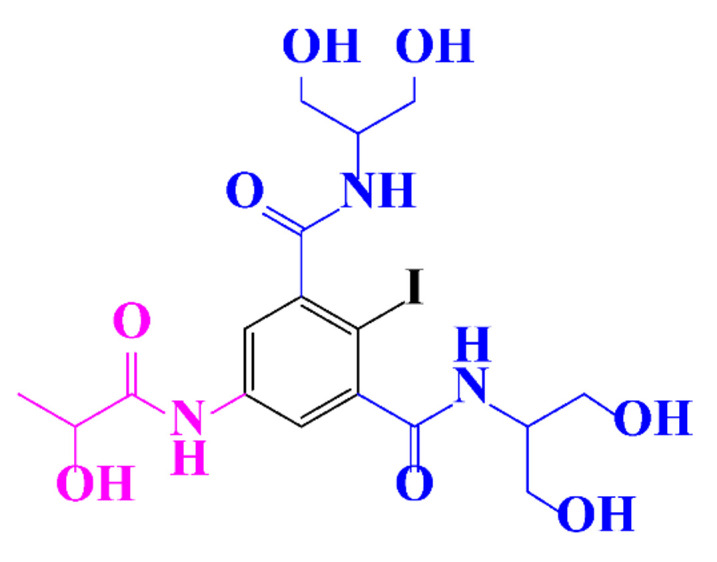	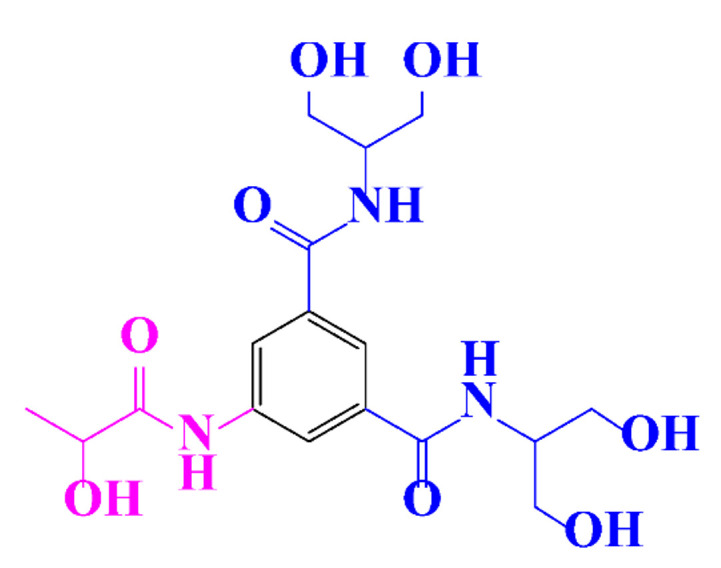	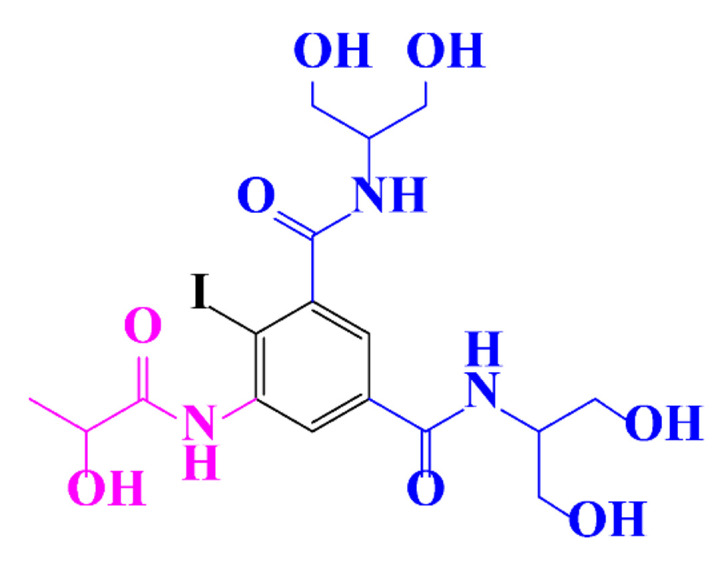				
Amine-Hydrolysis	Iron- SO^•^_4_^−^ Metal oxides- SO^•^_4_^−^ Ferrate-oxidation	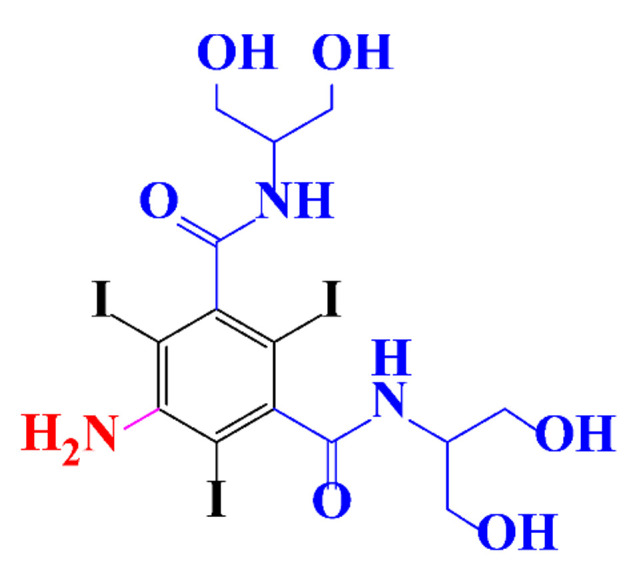	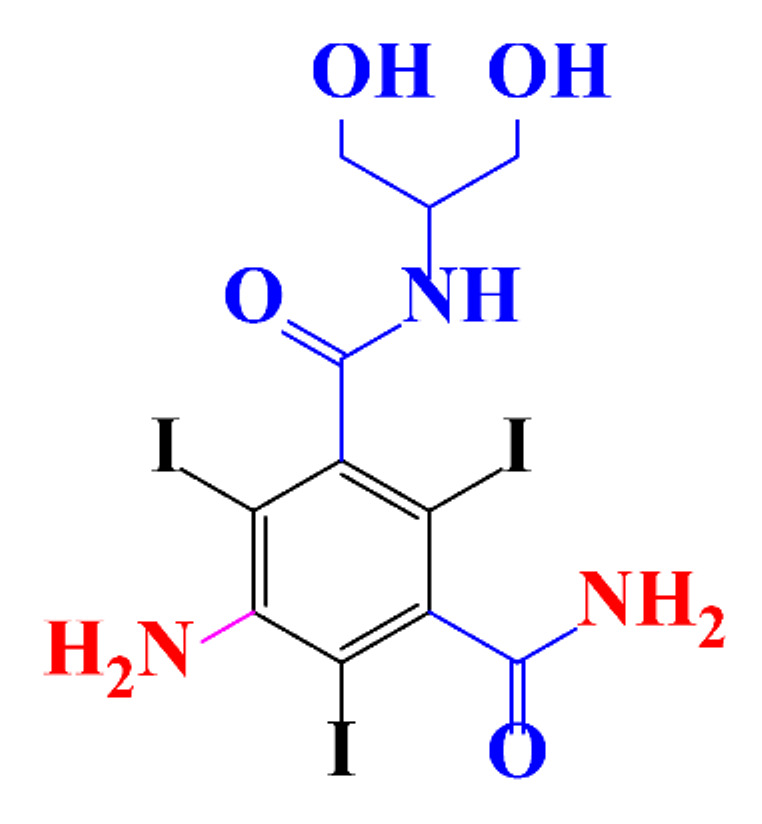	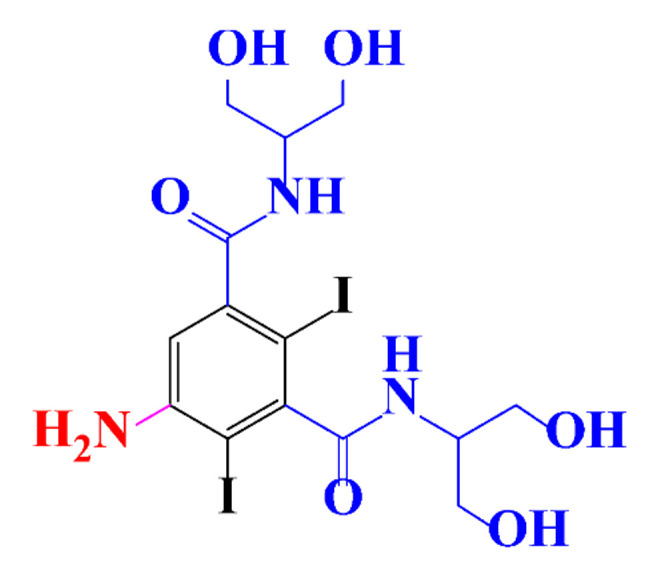	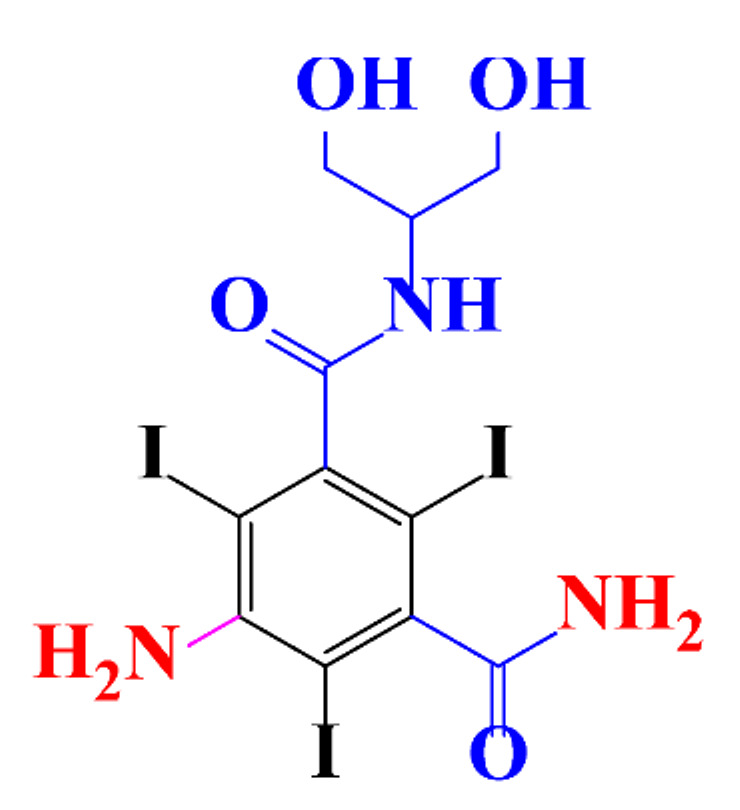	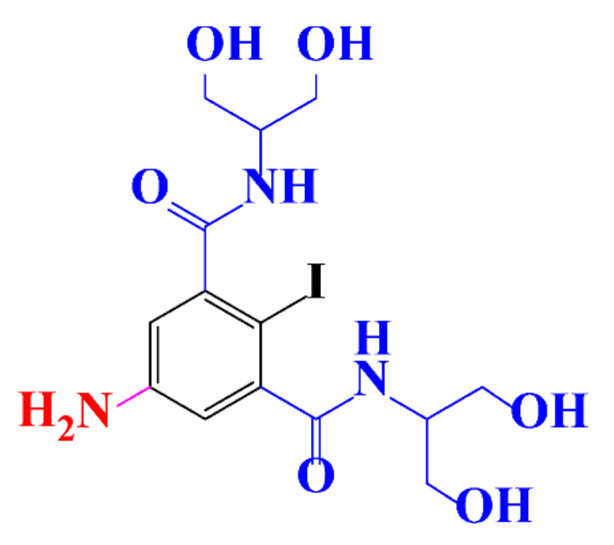	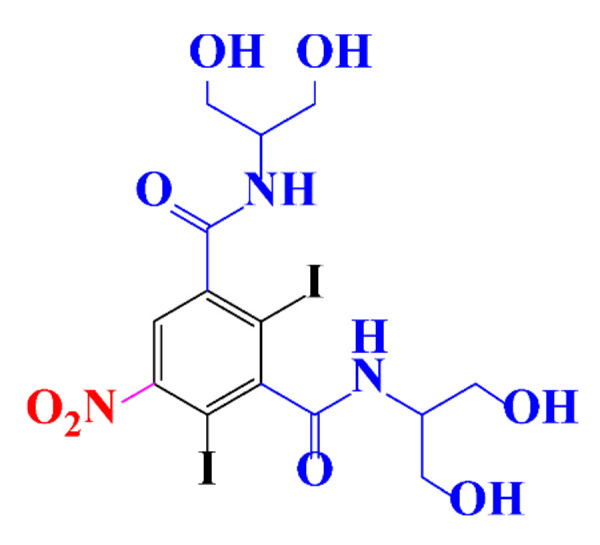	[116] [120] [157]
Amide-Oxidation	Iron-SO^•^_4_^−^ Metal oxides-SO^•^_4_^−^ Ferrate-oxidation	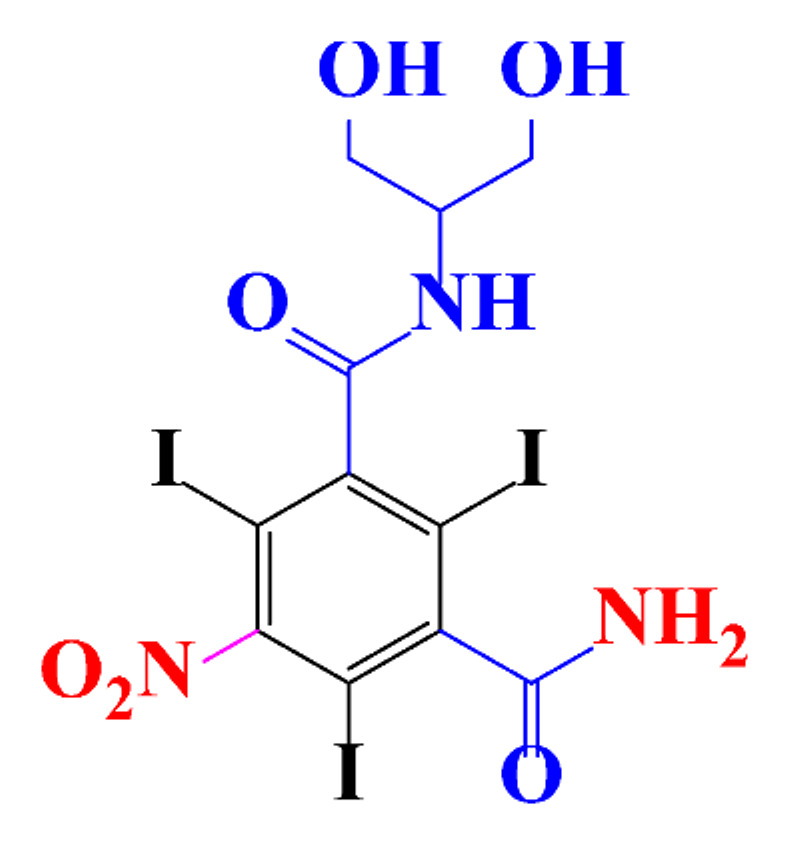	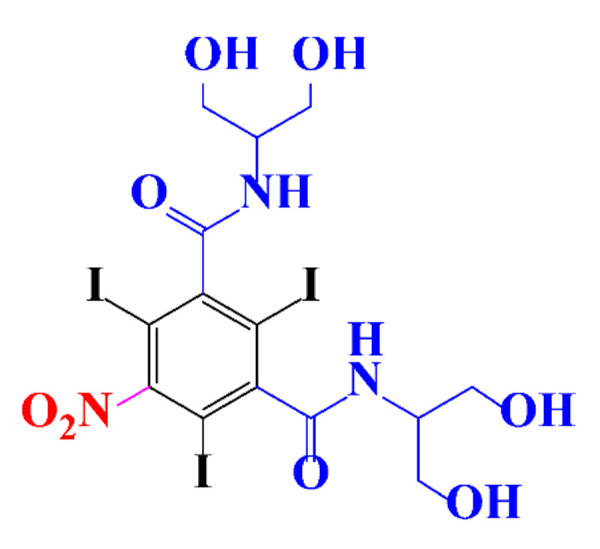	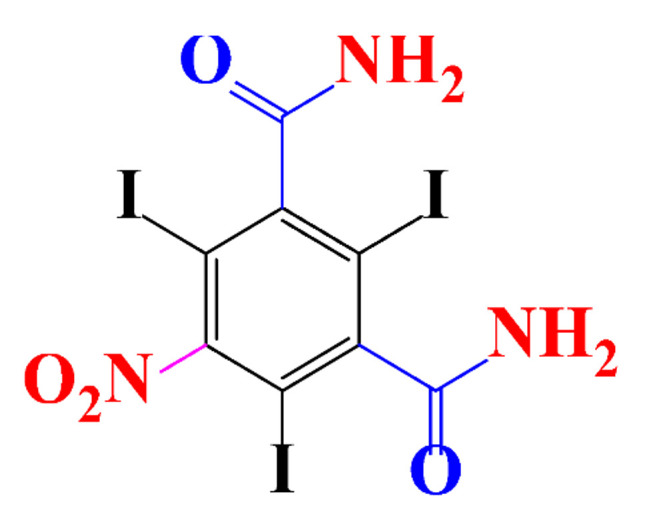	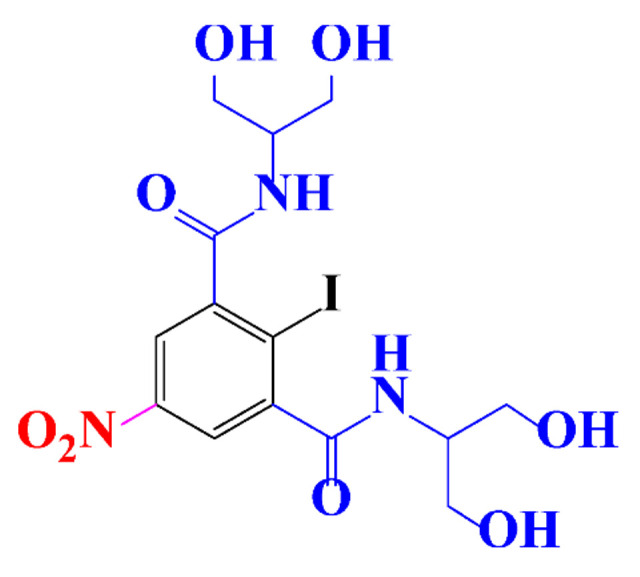			[116] [120] [157]
•Cl-Addition	UV-•Cl	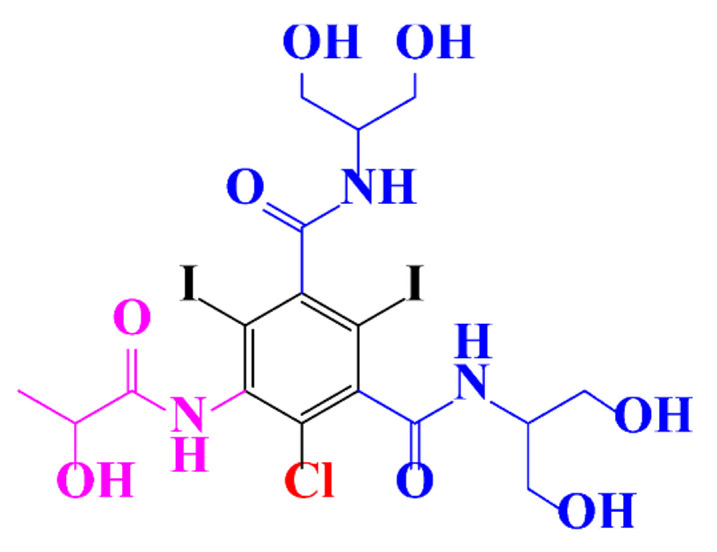	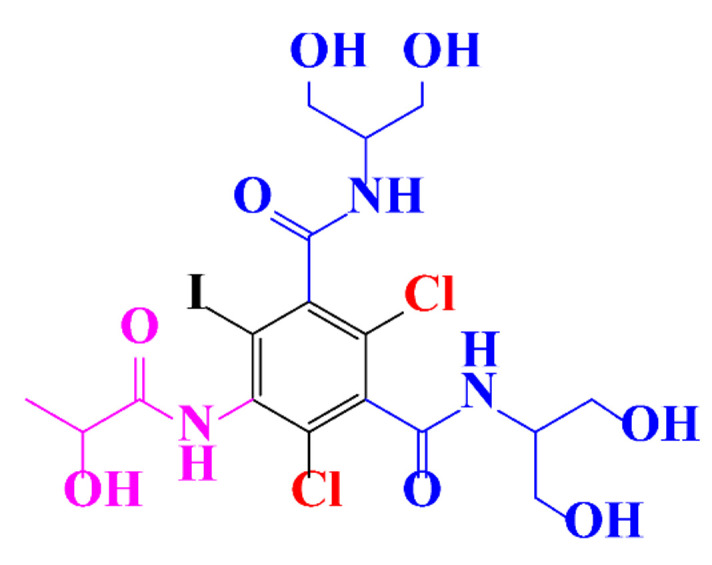	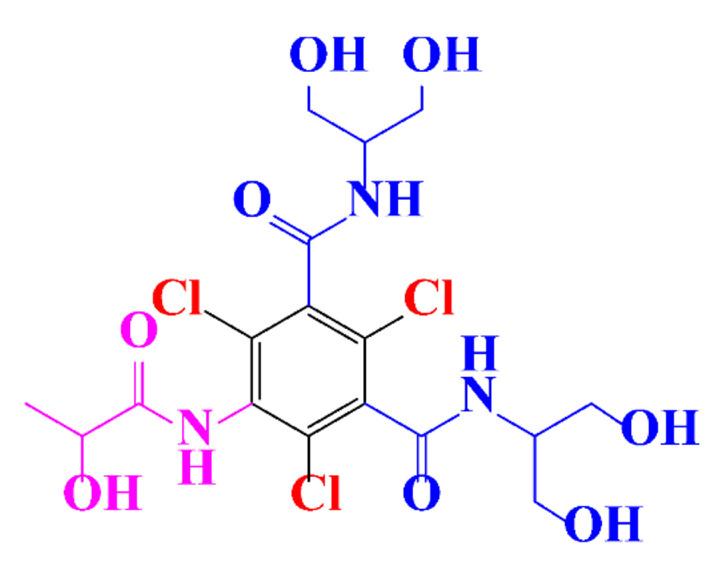				[71]
Hydrogen- Abstraction	UV-irradiation Metal oxides-SO^•^_4_^−^ Fe(III) oxalate UV-•Cl	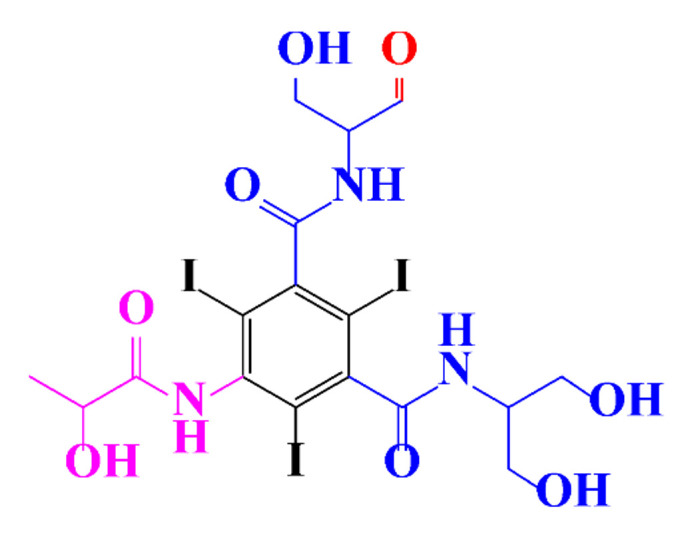	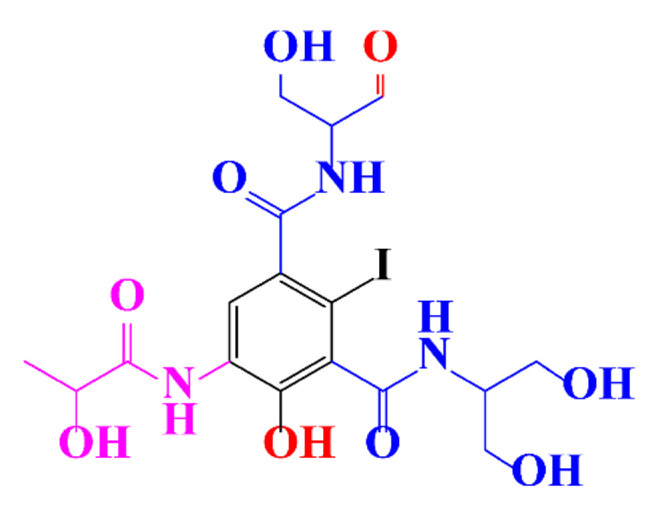	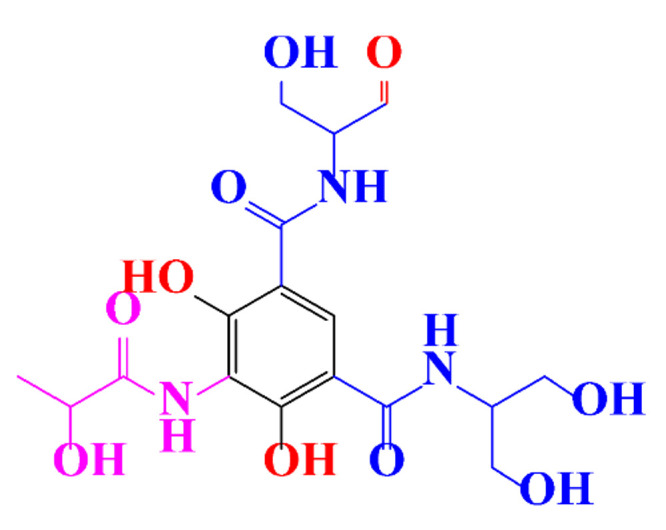	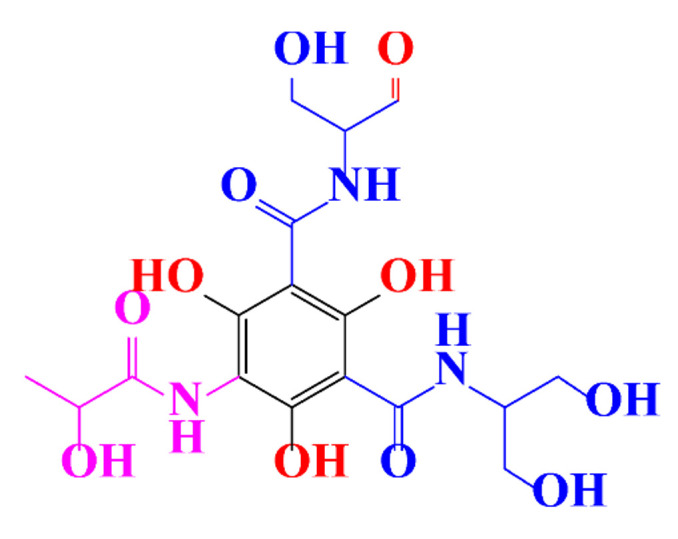	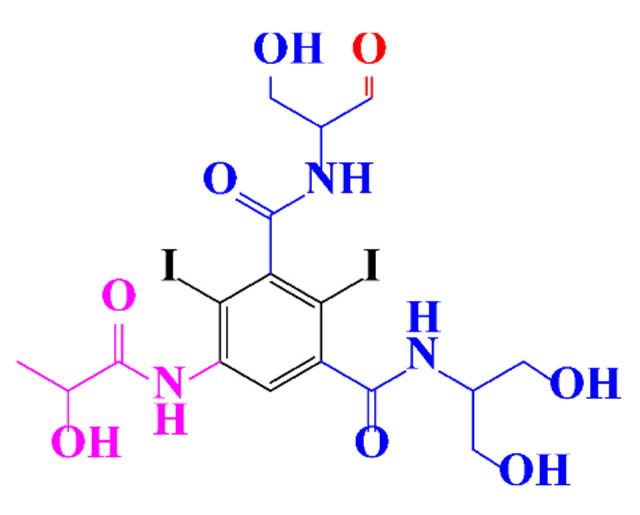	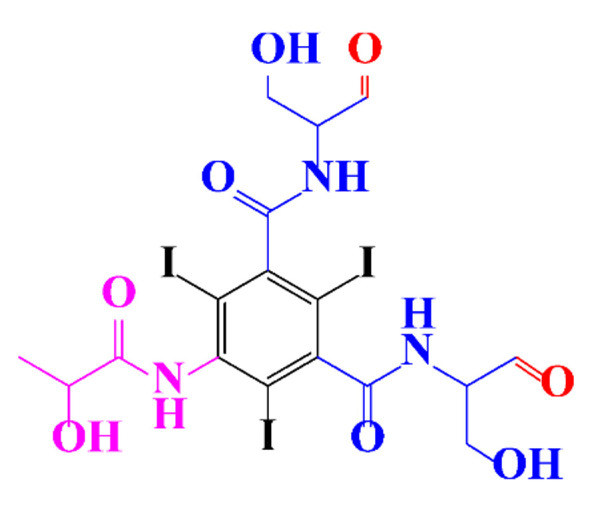	[85] [120] [158] [71]
		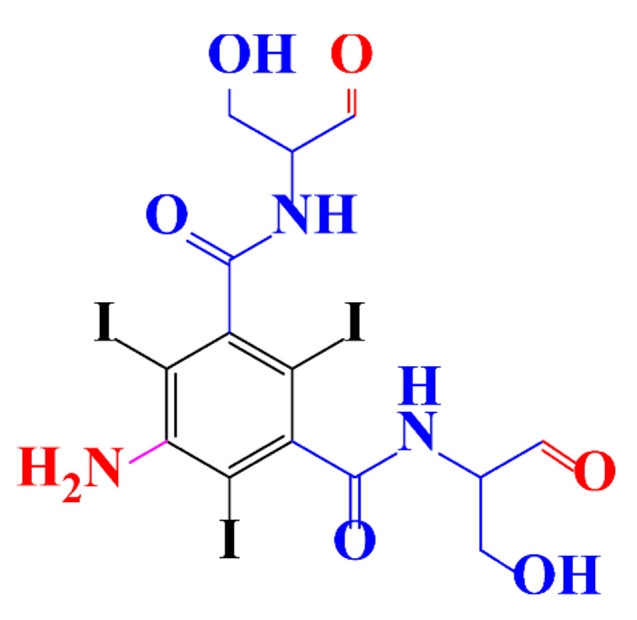	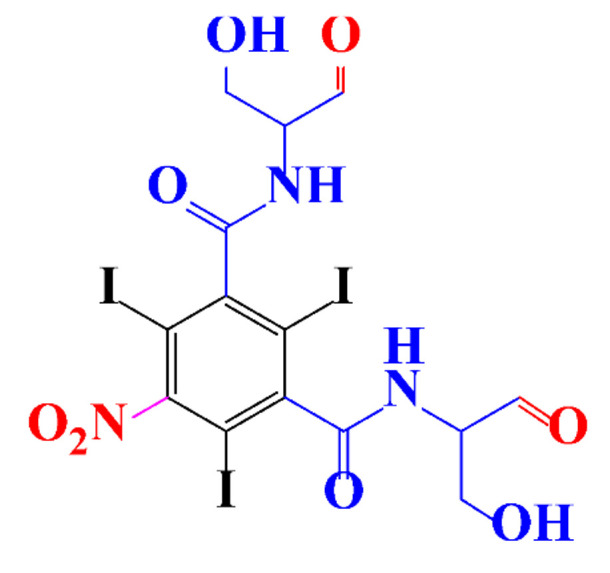					
•OH- Addition	UV-irradiation Metal oxides-SO^•^_4_^−^ Fe(III) oxalate UV-•Cl	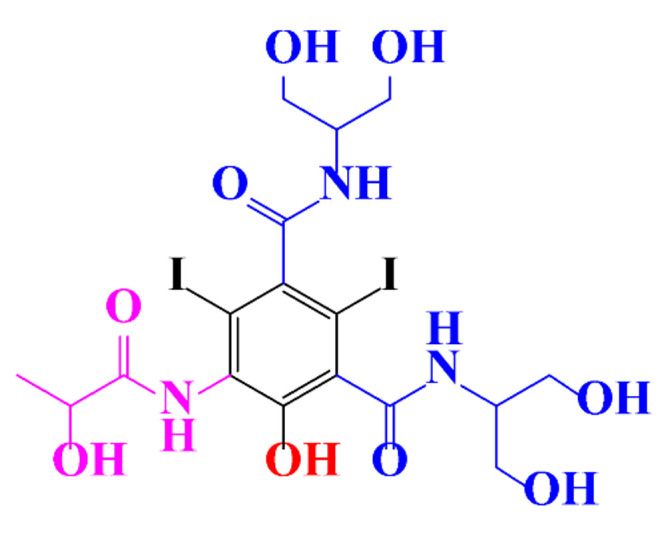	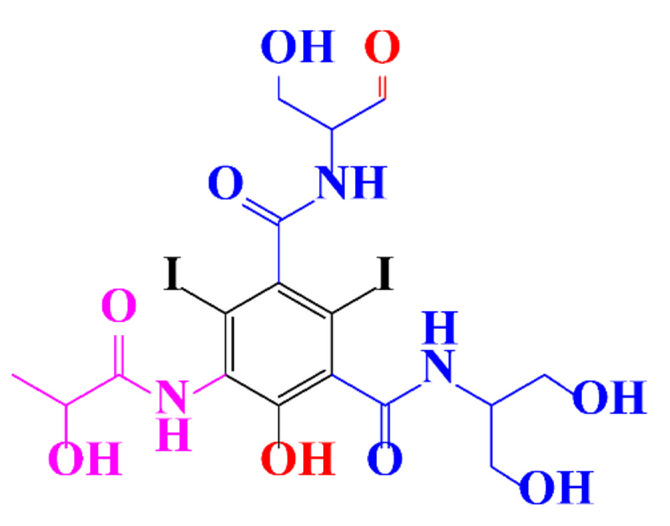	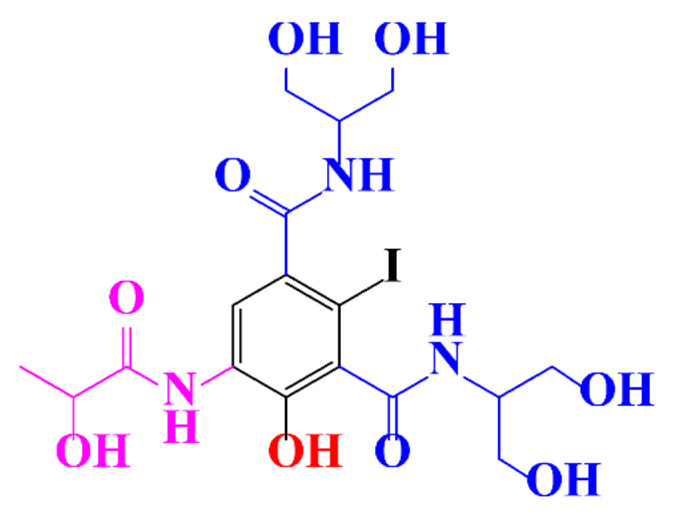	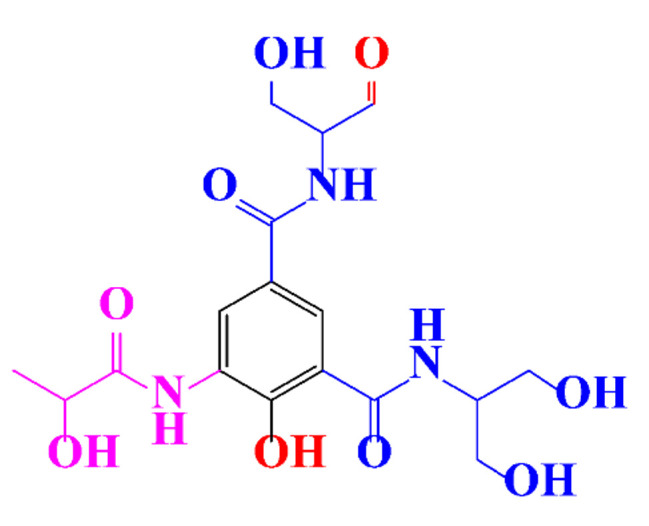	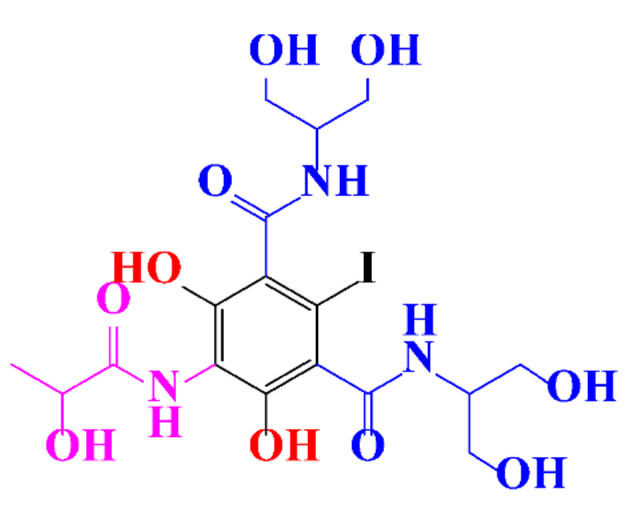	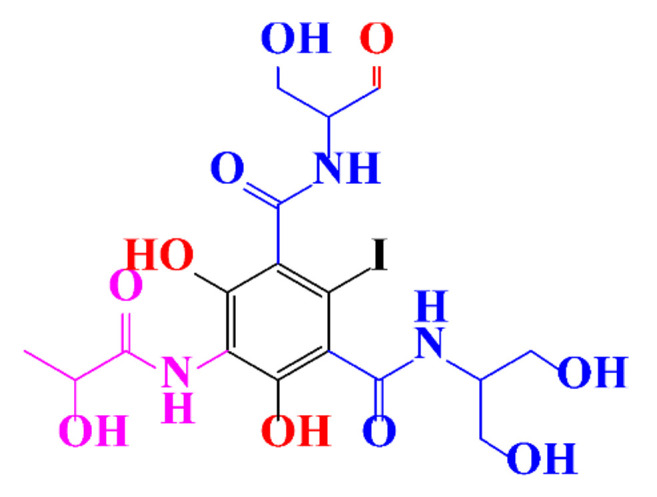	[85] [71] [120] [158]
		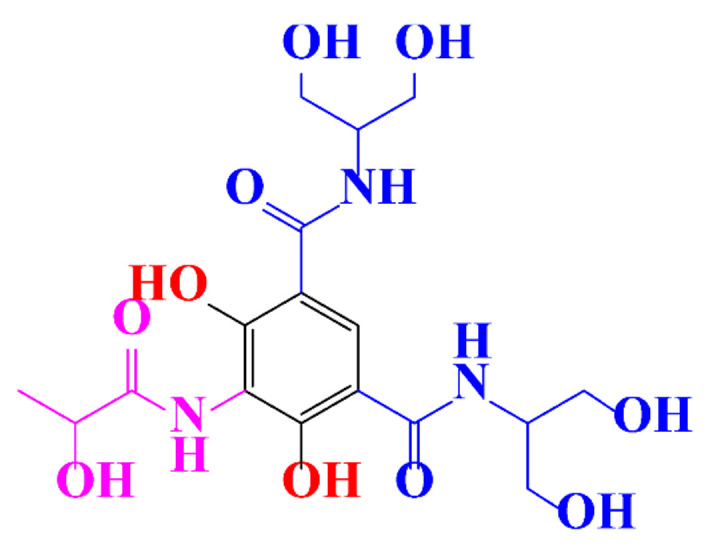	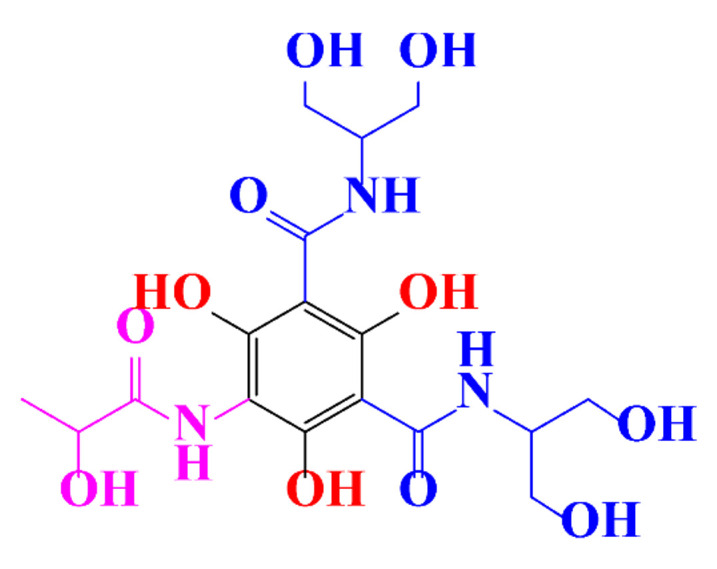	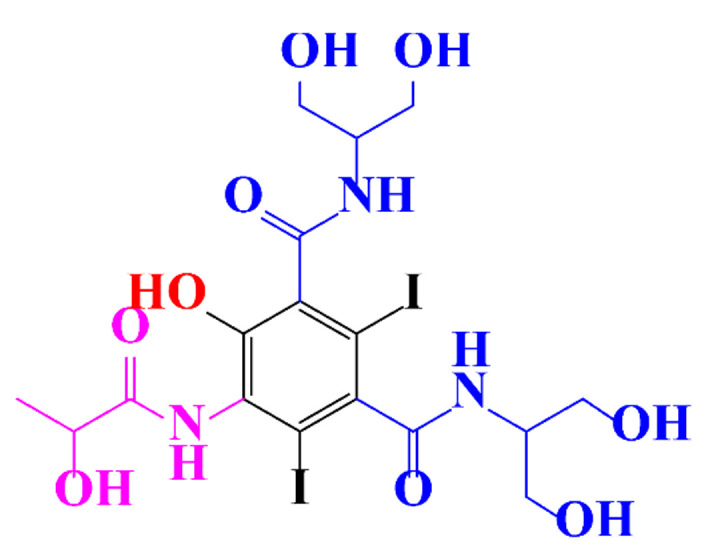	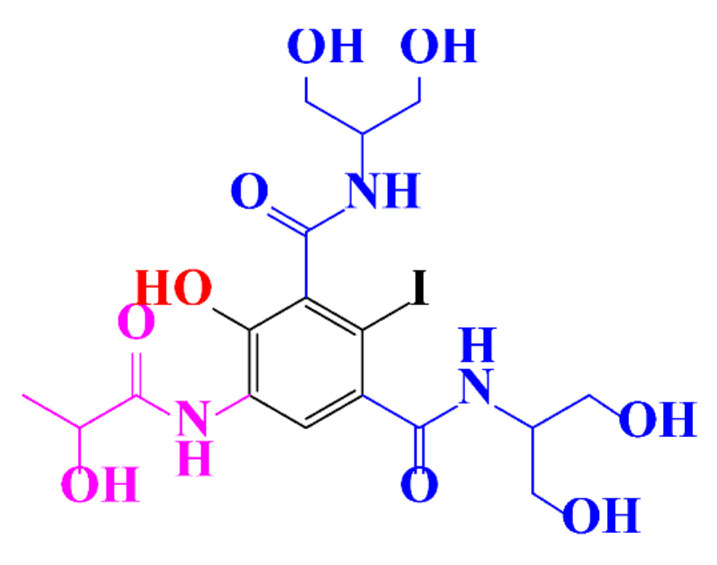	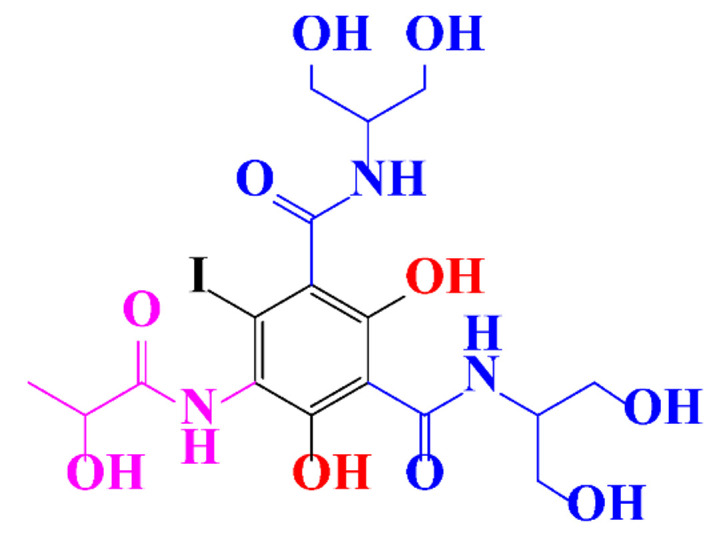		

**Table 9 molecules-28-00341-t009:** Identified by-products of iopromide in recent literature.

Pathways	Methods	by-Products					Ref.
Deiodination	Catalytic-Hydrodeiodination Gamma-Radiolysis	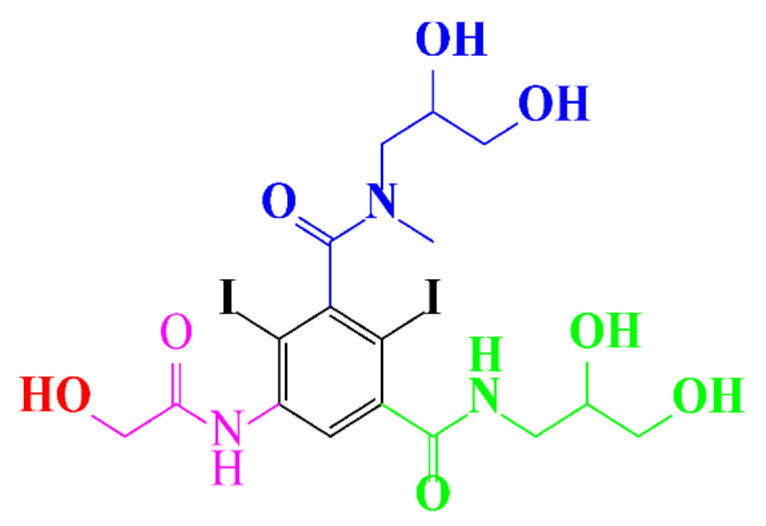	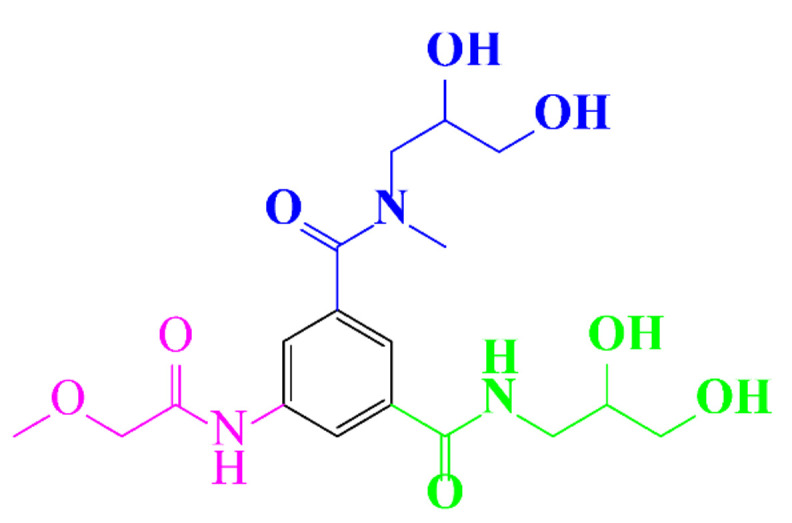	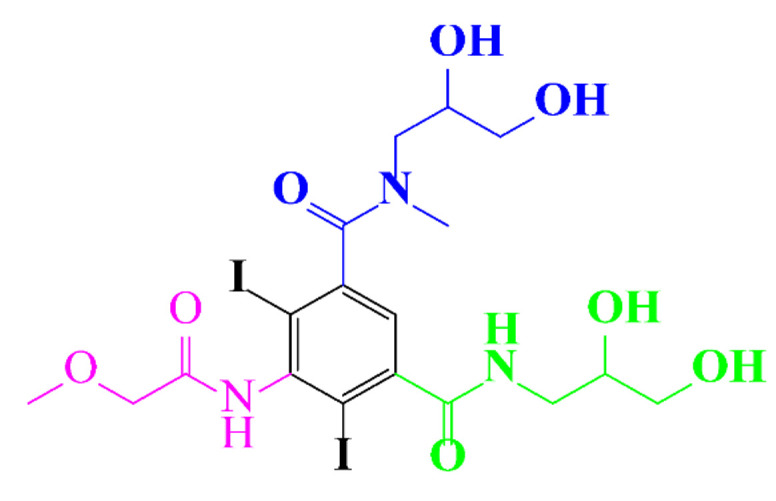	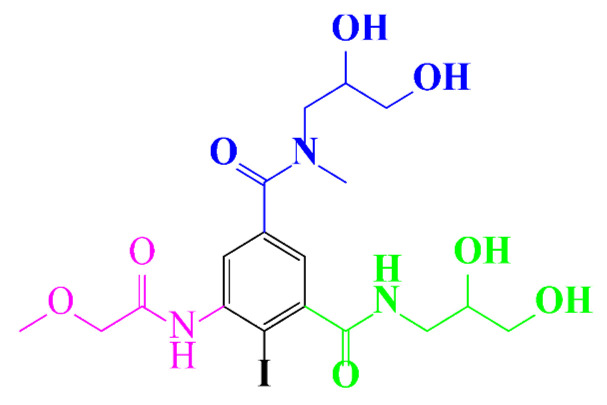	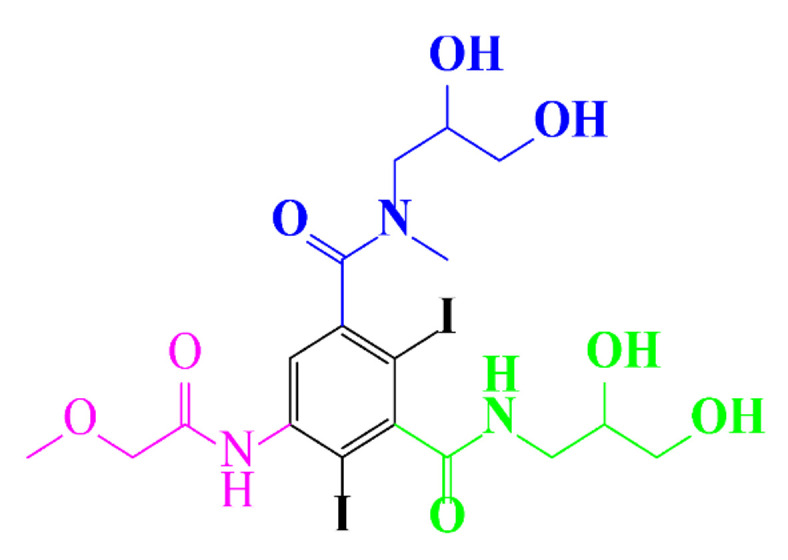	[104] [142] [159] [160]
Amide-Hydrolysis	UV-H_2_O_2_ Gamma-Radiolysis	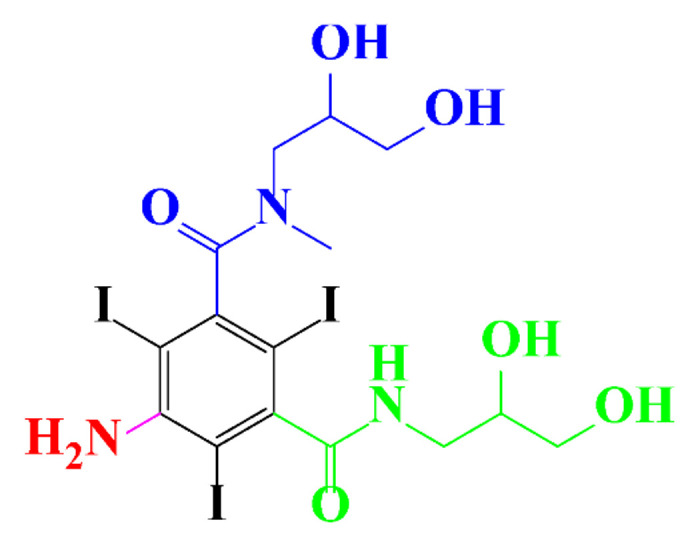	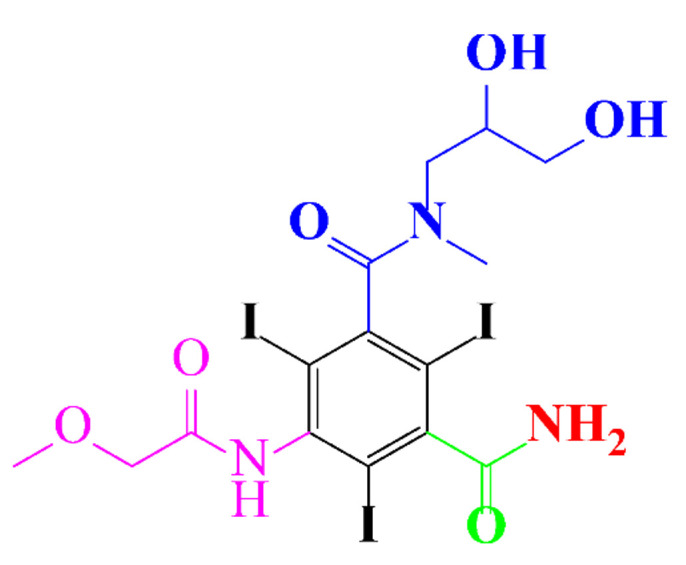	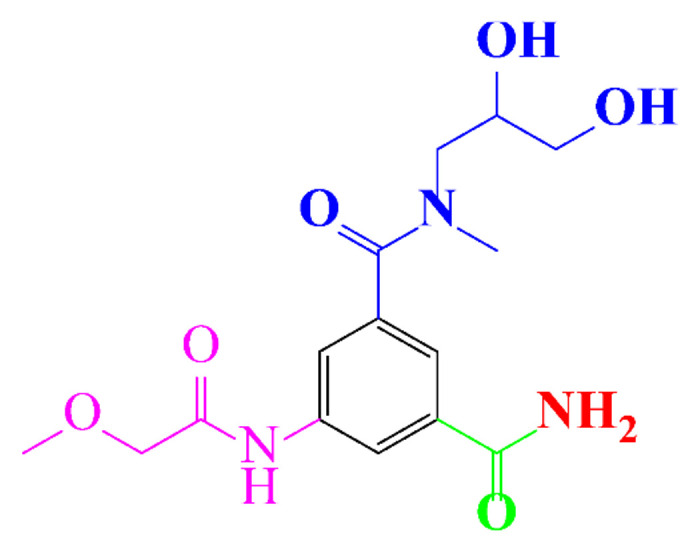	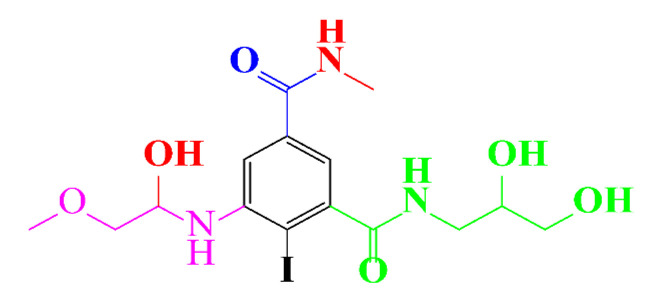		[104] [142]
Hydrogen-Abstraction	UV-SO^•^_4_^−^ Electro-oxidation UV-H_2_O_2_ Electro-Fenton Gamma-Radiolysis	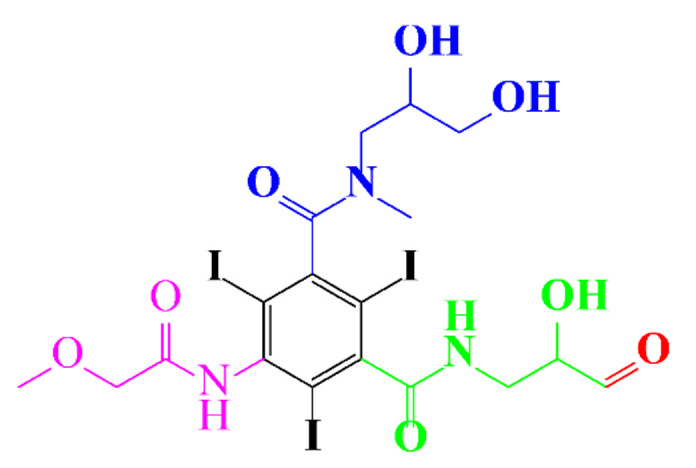	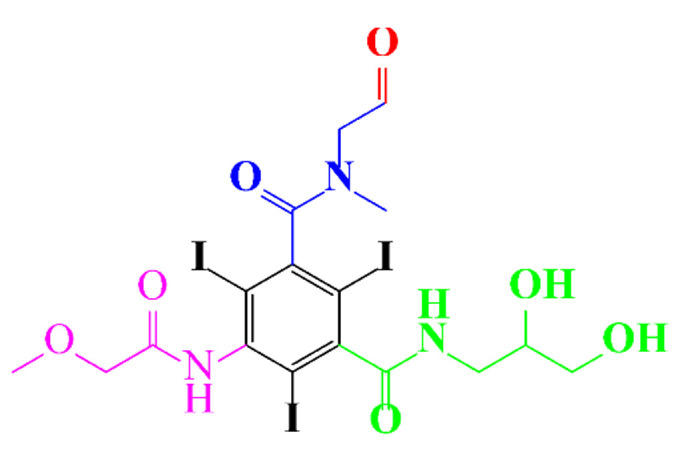	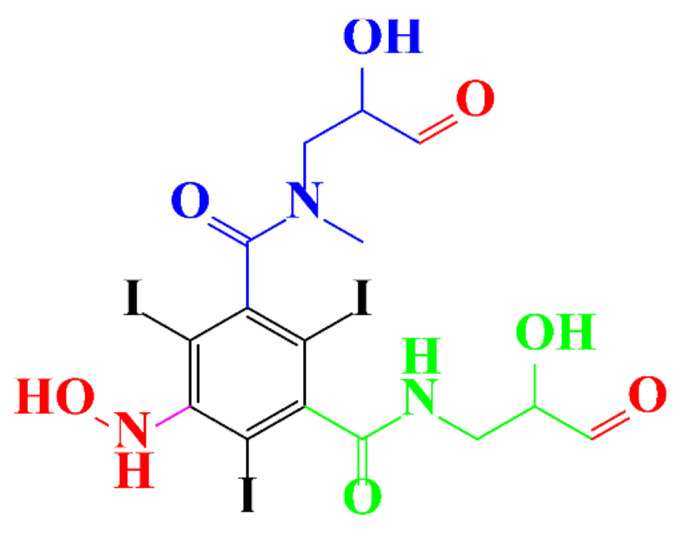	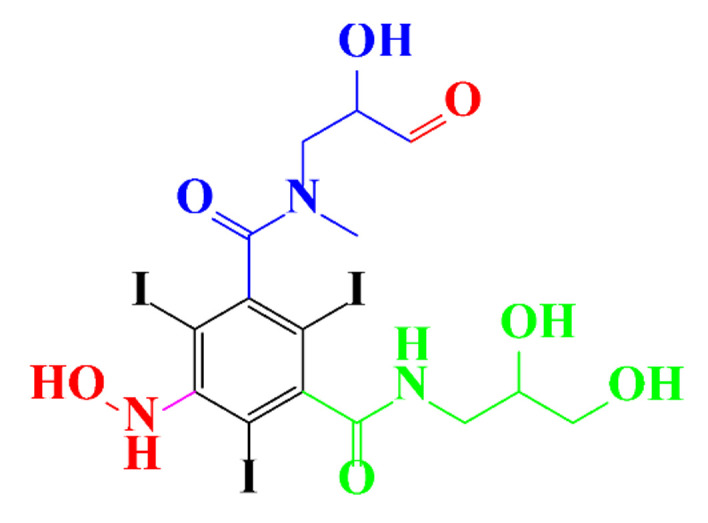	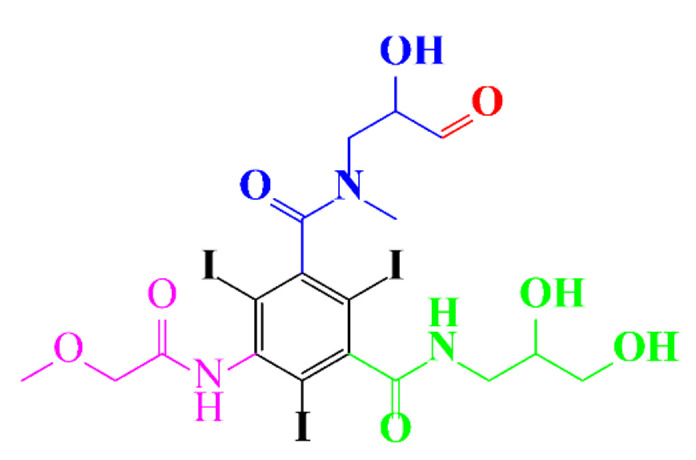	[104] [160]
•OH- Addition	UV-SO^•^_4_^−^ Electro-oxidation	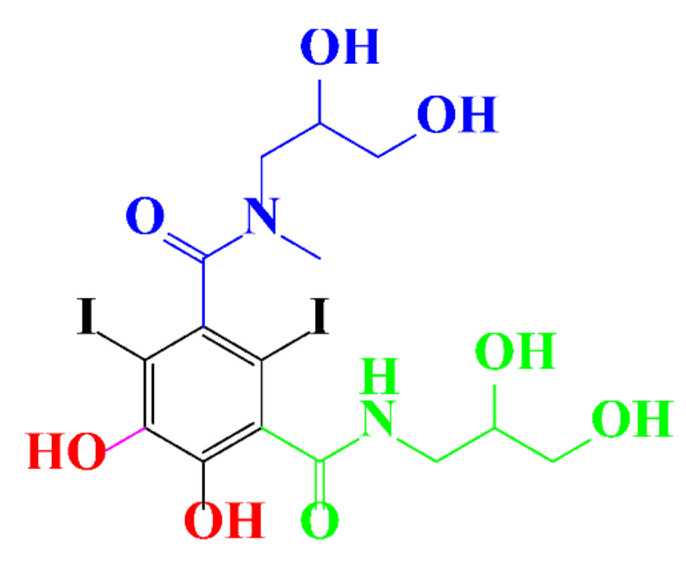	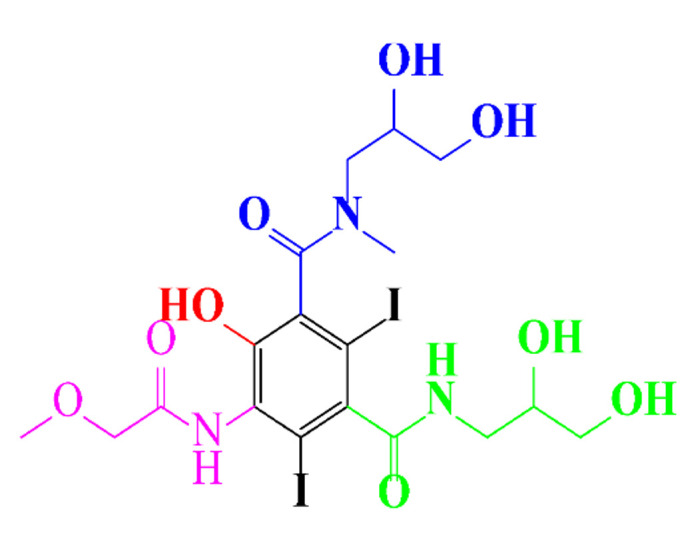				[104] [160]
Demethylation	UV-SO^•^_4_^−^ Gamma-Radiolysis	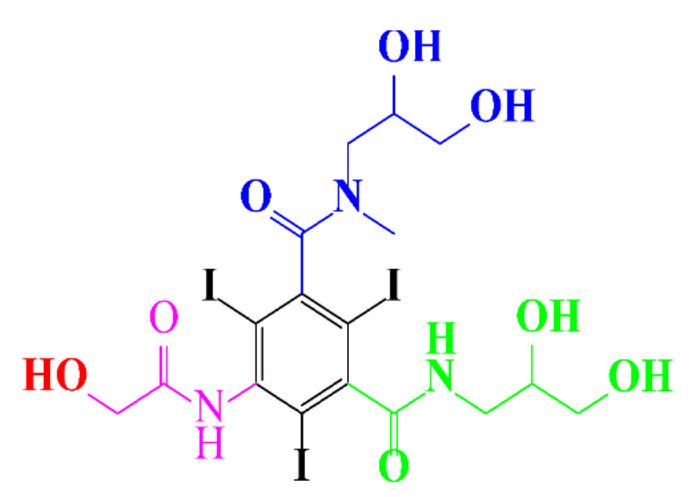	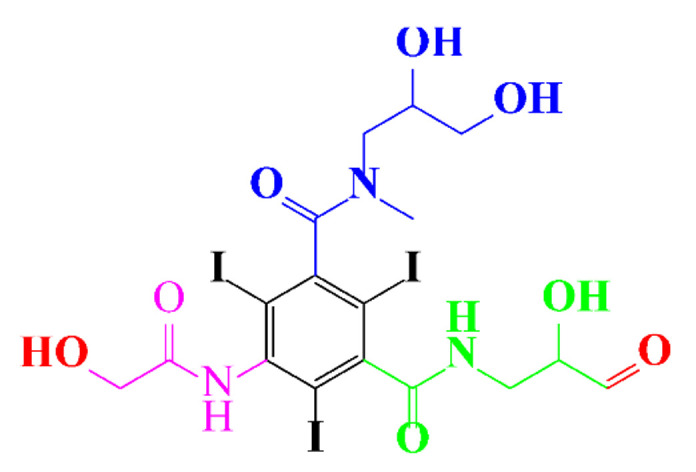				[104]

## Data Availability

Not applicable.

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
