# Peer review of "Removal of Iodine-Containing X-ray Contrast Media from Environment: The Challenge of a Total Mineralization"

_molecules, 2023, doi:10.3390/molecules28010341_

Round 1
Reviewer 1 Report
The Paper is quite up to the mark, but it lacks certain grammatical mistakes like missing some punctuation. The paper is quite structured and below said corrections can be implemented for betterment: Q1. The more latest literature review can be added in the introduction part, including their proper references. Q2. The Paper can be accepted, although some sentences can be shortened and grammatical mistakes can be removed. Q3. More references can be added to support the result and discussion part in every aspect of influencing factors.
Q4. The similarity index is high and hence needed to be relooked at.
Detailed comments are attached in a word file.

Author Response
According to the reviewer 1 recommendations, the following changes were made in the revised version of the manuscript:
- The document, when analyzed on Plagiarism’s software, i.e., Turnitin, is showing 17%. As per my view, it must be lower down up to 14% or so for an article. Self-Plagiarisms are also not accepted more than 2.5%.
We used “compilatio” to check the Plagiarism. We noticed that the titles of the article in the references are considered as plagiarism. However, we have modified some sentences in the review to avoid plagiarism.
- The tables and figures used are not clear and can be enhanced. Heading must be with sequential numbers like 1.0, 1.1, 1.2, etc.
Table 1 and 2 have been modified. Heading is already with sequential numbers. It has been checked and corrected.
- In reference section some reference is de-shaped, may be due to formatting. They are also needed to be corrected as per journal format.
We have corrected the references.
- The introduction must be reduced to one and a half pages.
The length of the introduction is only 1 page.
- The title needed significant modification.
The title has been modified.
- The numbering of content must correct.
We have checked the numbering of content and corrected it.
- The manuscript requires an extension of the literature.
The literature has been extended with recent articles and reviews.
- The manuscript does not illustrate great attention and activity in the field.
We have modified the introduction to underline the interest of the review.
- Tables also contain few references.
The review contains 9 tables with the main references.
- Please enhance the manuscript on analysis of earlier mention issues.
We thank the reviewer for the careful analysis of this review and his recommendations. We have tried to improve the review according to these recommendations.
- The figure number is distorted and can be rechecked.
The figure number has been checked.
- For the text clarity, would you refrain from using additional words, mostly meaningless filler words, which can be omitted or some archaic words see, e.g. "respectively", "thus", "hence", “therefore", "furthermore", "thereby", "basically,", "meanwhile", "wherein", "herein", "Nonetheless", "Perceivably," etc.?
We removed the words “Thus”, “Therefore”, “Furthermore”, “Meanwhile”. We kept the word “respectively” (3 times) for more clarity.
Specific Comments.
|
Introduction: |
The authors should describe the importance of their research more clearly. The references cited lack articles on said topic from last year. So, add more references (2018-2021) to support the author's points of view. Last paragraph must be an outline of the complete study showing the needed and targets assumed in the paper. Hence need minor revision. It also suggestive to add latest article in references. Please use the literature background on cleaner production/sustainability (but not self-citations, please) to broaden the manuscript foundation. Please develop a better title. This one does not state what is important to catch the prospective reader's attention. |
Eight references (2018-2021) have been added in the introduction. A paragraph related to the improvement of the ICM production process has been added with references. The title has been changed for a clearer summary of the review purpose.
|
Abstract |
It is needed to be started with small introduction and then quantitative description of the paper. It is also suggestive to shorten few unnecessary sentences in abstract. Underscore the scientific value-added to your paper in your abstract. Please look at articles we have published for models. Your abstract should clearly state the essence of the problem you are addressing, what you did and what you found and recommend. That would help a prospective reader of the abstract to decide if they wish to read the entire article. |
The abstract has been modified to underline the main objectives of the review.
|
2. ICM in the environment |
More specific details are needed to be added with use of latest reference Why such area selected must be justified either given in main text or in suppl material. Add one table depicting the need and why such study is needed to be done in a tabular for. Table 1 and Table 2 need more discussion in the text. What is significance of Page no 6 image. Why no figure name neither d discussed in text.
|
Table 1 has been modified to summarize the occurrence of ICM in the environment based on a recent review. The text has been also modified to give details that are more specific and to underline the importance of this review.
Page no 6 was a part of Figure 1. Letters a) and b) were added for a better understanding and the figure was clearly cited in the text and detailed. Table 2 has been cited several times in the text and the terms entry 1 to 6 have been added in Table 2 for a better understanding.
|
3.1.2. Anaerobic biological treatments |
Fig 2 need to be more discussed in text. Also, parameters selected must be justified and proper citation must be given to the method adopted. What is need of Table 3. Better remove it. Equation must be numbered as per journal guidelines. It is suggestive to add a table showing comparisons of all methods including combined methods about efficiency, parameters used, operating conditions etc for reader. Figures need clarity in term of pixels.
|
|
More discussion about Fig. 2 has been added in the text. Table 3 has been removed. The number of equations has been improved. A Table (Table 3) has already been added at the end of the part to summarize the UV-based AOP results to degrade ICMs. Tables 5 to 9 summarize the formed by-products. The poor quality of the figure is due to the PDF file. Each figure will be given separately to the journal.
|
|
|
3.2. Advanced oxidation processes (AOPs) for ICM removal and its sub-sections. |
Add a section specifically as discussion and clearly do not establish a strong correlation with cleaner production/sustainability/environmental concerns (however, as much as possible, avoid self-citations). In your discussion section, please link your empirical results with a broader and deeper literature review. Discussions and conclusions must go deeper, it would be more interesting if the authors focus more on the significance of their findings regarding the importance of the interrelationship between the obtained results and sustainable development/cleaner production in the sector context, and the barriers to do it, what would be the consequences, in the real world, in changing the observed situation, what would be the ways, in the real world, to change/improve the observed situation. SEM and EDS mapping must be more discussed. FTIR parameters must be shown in tabular form like theta , angles etc along with detection and describe it in more details in text. XPS spectra must be critically discussed and EDS mapping must be shown in tabular form showing bonds detected, frequencies and proper citation with discussion. What type of mechanism the CIP degradation along with key parameters has been observed , must be discussed. Mechanism of degradation of CIP by SCG/Fenton must be more critically discussed.
|
We did not understand the comments. We think that there is a mistake in this section concerning probably an article with experimental results and not the review.
|
3.3.4. Coupled processes for ICM removal |
I think it is better to discuss one para about the well-known products detected/reported so far. Please add more discussion for table 7 and Table 8. |
|
By-products identification during the degradation of ICM |
specific details are needed to be added with use of latest reference. Use of some pictorial/diagram will be more elaborative for readers. Better to write in-depth analysis of previous study. Add a table showing previous studies based on this topic.
|
We do not understand very well the recommendation of the reviewer for these last paragraphs. We have already summarized in several tables the structure of identified by-products according to the treatment.
|
4. Conclusions |
This section is needed to be free from any variables are symbols. Only main pointed like what was expected and what was achieved must be written. What signification contribution this study to the society must be mentioned in this section. Please make sure your conclusions' section underscores the scientific value-added of your paper and/or the applicability of your results. Highlight the novelty of your study. Clearly discuss what the previous studies that you are referring to are. What are the Research Gaps/Contributions? In your conclusions, please discuss the implications of your research.
|
We tried to improve the conclusion. Some sentences have been added to underline the most promising results for treatment of real samples.
Reviewer 2 Report
I recommend the publication in present form.
Author Response
We thank the reviewer for his comments.
Reviewer 3 Report
The authors studied micropollutants based on the X-ray contrast media to safeguard public health. The appropriate adsorbent and methods are always welcome in removing the toxic pollutants based on the selectivity and sensitivity parameters. Some important points are important to be addressed before going to consider the possible publication in this journal.
-The English language needs to check carefully in the revision stage because of many careless mistakes in many positions.
-The Figure’s quality needs to be improved in the revision stage.
-References: There are many references that are not adjacent to this study. The authors need to take notes in the revision stage and cite relevant references including high-impact journals to make the manuscript to a broad range of readers.
-Abstract: This section needs to be improved by presenting novel findings such as investigating the effect of the diverse pollutants, and the main experimental works in the revision stage.
-Introduction: Composite materials are growing attention for diverse pollutants removal based on their specific functionality and surface area as reported by Awual group according to ScienceDirect. The authors need to indicate such points for a broad range of readers. Moreover, the authors need to cite high-impact articles to make the manuscript high-level. The following specific articles may take be noted in the revision stage of Chemical Engineering Journal, 266 (2015) 368–375; Microchemical Journal 161 (2021) 105800.
-Scientists considered the removal technology based on selectivity, sensitivity, cost-effectiveness, and so on. The authors need to indicate such a point in the revised manuscript.
-The materials characterization is not accurate. The main morphology needs to be added instead and then compared with reported materials including Journal of Molecular Liquids 329 (2021) 115541.
-Can the present study be effective from the real waste samples? The authors need to write such sentences in the revision stage.
I would like to see the revised manuscript.
Author Response
The authors studied micropollutants based on the X-ray contrast media to safeguard public health. The appropriate adsorbent and methods are always welcome in removing the toxic pollutants based on the selectivity and sensitivity parameters. Some important points are important to be addressed before going to consider the possible publication in this journal.
According to the reviewer 2 recommendations, the following changes were made in the revised version of the manuscript:
-The English language needs to check carefully in the revision stage because of many careless mistakes in many positions.
The English language has been carefully checked by the authors.
-The Figure’s quality needs to be improved in the revision stage.
The poor quality of the figure is due to the PDF file. Each figure will be given separately to the journal.
-References: There are many references that are not adjacent to this study. The authors need to take notes in the revision stage and cite relevant references including high-impact journals to make the manuscript to a broad range of readers.
References including high-impact journals have been added.
-Abstract: This section needs to be improved by presenting novel findings such as investigating the effect of the diverse pollutants, and the main experimental works in the revision stage.
The abstract has been modified to underline the main objectives of this review.
-Introduction: Composite materials are growing attention for diverse pollutants removal based on their specific functionality and surface area as reported by Awual group according to ScienceDirect. The authors need to indicate such points for a broad range of readers. Moreover, the authors need to cite high-impact articles to make the manuscript high-level. The following specific articles may take be noted in the revision stage of Chemical Engineering Journal, 266 (2015) 368–375; Microchemical Journal 161 (2021) 105800.
We thank the reviewer for this suggestion. The part concerning the adsorption processes was in the introduction of part 3. So composite materials and references were added in this part.
-Scientists considered the removal technology based on selectivity, sensitivity, cost-effectiveness, and so on. The authors need to indicate such a point in the revised manuscript.
All along the review, the main results concerning the selectivity, sensitivity and cost-effectiveness for each process was underlined.
-The materials characterization is not accurate. The main morphology needs to be added instead and then compared with reported materials including Journal of Molecular Liquids 329 (2021) 115541.
We do not understand this comment. There is no materials characterization in the review.
-Can the present study be effective from the real waste samples? The authors need to write such sentences in the revision stage.
Some sentences have been added to underline the most promising results for treatment of real samples.
I would like to see the revised manuscript.
Round 2
Reviewer 3 Report
The authors are not serious in revision.
Reject
Author Response
We did carefully the corrections asked by the reviewer. The English langage has been corrected by a native speaker who is also an English professor at the University.
We have also added and discussed the following articles as recommended by the editor:
DOI: 10.3303/CET1332004 DOI: 10.1016/j.jece.2017.07.078 DOI: 10.3303/CET1438059